# Cyclodextrin-based host-guest complexes loaded with regorafenib for colorectal cancer treatment

Hongzhen Bai [1], Jianwei Wang[1], Chi Uyen Phan[1], Qi Chen [1], Xiurong Hu[1], Guoqiang Shao[2], Jun Zhou[1], Lihua Lai [3✉] & Guping Tang [1✉]

The malignancy of colorectal cancer (CRC) is connected with inflammation and tumor-associated macrophages (TAMs), but effective therapeutics for CRC are limited. To integrate therapeutic targeting with tumor microenvironment (TME) reprogramming, here we develop biocompatible, non-covalent channel-type nanoparticles (CNPs) that are fabricated through host-guest complexation and self-assemble of mannose-modified γ-cyclodextrin (M-γ-CD) with Regorafenib (RG), RG@M-γ-CD CNPs. In addition to its carrier role, M-γ-CD serves as a targeting device and participates in TME regulation. RG@M-γ-CD CNPs attenuate inflammation and inhibit TAM activation by targeting macrophages. They also improve RG's anti-tumor effect by potentiating kinase suppression. In vivo application shows that the channel-type formulation optimizes the pharmacokinetics and bio-distribution of RG. In colitis-associated cancer and CT26 mouse models, RG@M-γ-CD is proven to be a targeted, safe and effective anti-tumor nanomedicine that suppresses tumor cell proliferation, lesions neovascularization, and remodels TME. These findings indicate RG@M-γ-CD CNPs as a potential strategy for CRC treatment.

[1] Department of Chemistry, Zhejiang University, 310028 Hangzhou, PR China. [2] Department of Nuclear Medicine, Nanjing First Hospital, Nanjing Medical University, 210029 Nanjing, PR China. [3] Department of Pharmacology, School of Medicine, Zhejiang University, 310058 Hangzhou, PR China. ✉email: lailihua@zju.edu.cn; tangguping@zju.edu.cn

Colorectal cancer (CRC) is one of the most prevalent and malignant cancers for which colectomy combined with chemotherapy has been regarded as a pivotal treatment strategy. However, the disruptive activation of protooncogenes and tyrosine-protein kinase receptors remains the main challenge for CRC therapy, especially for the patients with unresectable CRC tumors[1–4]. The therapeutic response of CRC chemotherapy has recently been improved through the introduction of biological agents targeting vascular endothelial growth factor (VEGF)/ VEGF receptor (VEGFR) axis, such as bevacizumab, aflibercept, and regorafenib (Stivarga®, RG)[5,6]. RG is a multikinase inhibitor that targets a wide spectrum of kinases involved in angiogenesis and oncogenesis, such as VEGFR-2, platelet-derived growth factor receptor β (PDGFR-β), and the mitogen-activated protein kinases (MAPKs), and confers significant benefits in preventing CRC progression[7,8]. Furthermore, RG has been reported to potently reduce the infiltration of tumor-associated macrophages (TAMs) in CRC by blocking tyrosine kinase with immunoglobulin and epidermal growth factor homology domain 2 (TIE2) signaling, contributing to the complete metastasis inhibition[9,10]. Despite these notable progresses, RG shows poor drug properties (i.e., solubility, dissolution, permeability, circulation and distribution), which compromises therapeutic outcome and induces side effects[11,12]. Therefore, optimizing the pharmacokinetics and improving the bioavailability of RG is required to achieve effective therapeutic response against CRC.

As a hallmark of cancer, heterogeneous tumor microenvironment (TME) is another obstacle to optimal therapeutic response[13]. In the TME of CRC, abundant tumor-supportive factors, such as VEGFs, PDGFs, matrix metalloproteinases (MMPs), promote cancer cell survival, pathogenic angiogenesis, and extracellular matrix deposition. These dynamic processes support the construction of tumor niche and accelerate the invasion-metastasis cascade. Recent work has demonstrated that the tumor-supportive factors can harbor and support the survival of CRC stem cells during chemotherapy, influencing the therapeutic response[14]. On the other hand, chronic inflammation contributes to the pathogenesis and the mechanistic phases of CRC, especially colitis-associated colon cancer (CAC). Excess inflammatory cytokines within TME, including interleukin-1β (IL-1β), tumor necrosis factor-α (TNF-α) and IL-6, persistently activate some transcription factors, such as nuclear factor-κB (NF-κB), signal transducer and activator of transcription 3 (STAT3)[15,16]. These transcription activities are crucial mediators in all steps of colonic cancer, meanwhile the inflammation can also impede the therapeutic response. It has been proven that the pro-inflammatory cytokine TNF-α within the TME induces the abnormality of target expression in cancer cells, which compromises the targeted therapy[17]. Accumulating evidence has revealed that the heterogeneous TME of CRC is primarily established by the alternatively activated colonic macrophages. In colonic lamina propria, macrophages maintain pro-inflammatory phenotype, linking inflammation with polyposis, or carcinogenesis. When educated with CRC cells, the colonic macrophages acquire pro-tumor phenotype (M2), bidirectionally communicating with cancer cells to promote CRC progression[18–20]. Clinical study has suggested that the TAM population is significantly increased after conventional therapy, and preventing the recruitment of TAMs may improve the clinical responses to chemotherapy or radiotherapy. Thus, reprogramming the TME by alleviating the inflammation and regulating the TAM polarization has become an intriguing therapeutic target in CRC treatment.

Nanomedicine have provided a diverse toolbox to optimize CRC treatment, which has realized efficient delivery of standard CRC chemotherapeutics (i.e., 5-fluorouracil, oxaliplatin, and irinotecan) and targeted agents (i.e., sorafenib, regorafenib)[21–23].

Furthermore, emerging nano-medicines have also shown to reprogram the TME. Re-polarization or depletion of M2 TAMs through target delivery of kinase inhibitor or CSF-1R inhibitor has been reported to boost the chemotherapeutic response upon CRC[10,24]. Recently, nano-designs have been demonstrated to combat inflammation in colorectal environment by eliminating reactive oxygen species, releasing polyphenol molecules, and delivering anti-inflammatory antagonists[25–27]. However, strategies that integrate TAM deactivation with inflammation normalization have not been reported. Moreover, achieving the synergy of malignancy suppressing and TME reprogramming is still a significant challenge in the field of nanomedicine.

Mannose receptor (MR) is a highly effective endocytic receptor that functions to promote the cellular internalization and trafficking of molecules or nanoparticles with mannose terminus. Given that both CRC cells and colonic macrophages overexpress MR on the surface[28,29], mannose modification or mannosylation is efficient dual-targeting strategy and can be implanted into the design of CRC-specific nanomedicine.

Cyclodextrins (CDs) are biocompatible, macrocyclic amphiphiles who have hydrophilic cavity exteriors and hydrophobic cavity interiors. This property is responsible for their ability to act as host molecules encapsulating hydrophobic guests within the cavities[30]. The CD-based host-guest inclusion complexes have been widely exploited in pharmaceutical application to optimize the drug properties[31–33]. Moreover, CDs have also been employed as excellent building blocks to construct nano-structured materials, providing new opportunities in cancer treatment[34–36]. Notably, several studies have recently demonstrated that CDs initiate the anti-inflammatory mechanisms[37–39]. In atherosclerosis model, CD has been proven to increase the removal of cholesterol[40]. Furthermore, it attenuates inflammation by mediating liver X receptor (LXR) transcription in macrophages, resulting in the potent downregulation of pro-inflammatory cytokines (IL-1β, IL-6, and TNF-α) and promoting the atherosclerosis regression. The anti-inflammatory function has expanded the application of CDs and provided a new perspective in CD-based nanomedicine.

Here we formulate a chaperoned nanomedicine that is developed based on host-guest molecular recognition and non-covalent self-assembly. In our design, γ-CD and RG are used as the molecular modules to construct the artificial chaperone (shown in Fig. 1). To endow the chaperone with CRC-targeting capacity, γ-CD is modified with mannose, yielding a functionalized host molecule mannose-γ-CD (M-γ-CD). Based on the molecular recognition motifs, M-γ-CD encapsulates RG within its cavity to form the interlocked molecule RG@M-γ-CD. Notably, RG@M-γ-CD exhibits a higher order self-assembly and forms nanoparticles with channel architecture (channel-type nanoparticles, CNPs). Our study demonstrates that the mannose modification and the nanoscale, non-covalent channel construction improves the biodistribution, pharmacokinetic, and pharmaceutical properties for RG. More importantly, the anti-inflammation of CD is implanted into the nanomedicine which exerts the synergy with RG in remodeling the TME of CRC. Validated in CAC and CT26 models, RG@M-γ-CD CNPs achieve a potent therapeutic efficacy combating CRC, and integrating the carrier's chemical motif with its biological function has been confirmed to provide an efficient approach in the design of nanomedicine.

## Results

**Fabrication and characterization of RG@M-γ-CD CNPs**. The fabrication of RG@M-γ-CD host-guest complex was illustrated in Fig. 2a. Mannose was conjugated with γ-CD by using

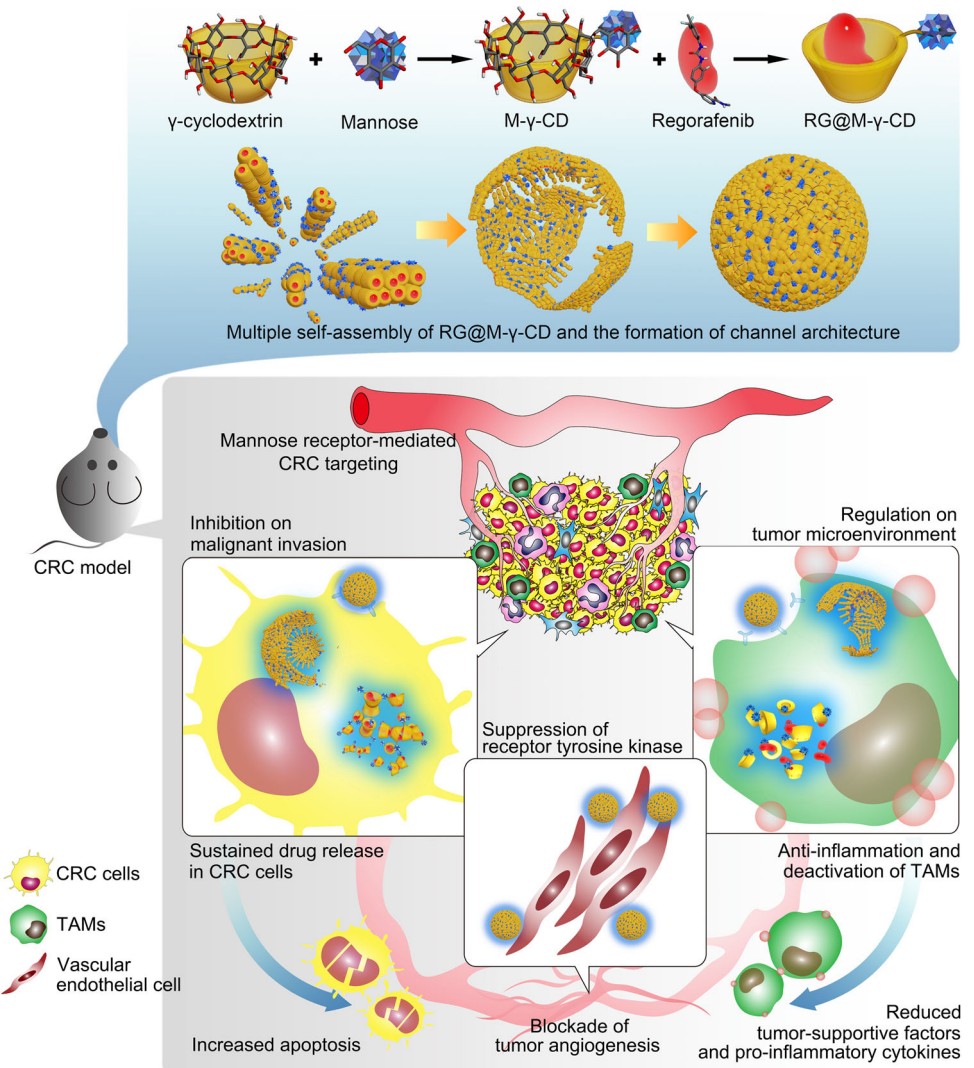

**Fig. 1 Design of regorafenib@mannose-γ-cyclodextrin channel-type nanoparticles (abbreviated as RG@M-γ-CD CNPs) and the synergetic anti-CRC mechanism.** RG@M-γ-CD CNPs inhibit tumor cell survival, lesion angiogenesis, attenuate inflammation and suppress tumor-associated macrophages activation through potentiating the kinase suppression of regorafenib and exerting the biological function of cyclodextrin.

1,1′-carbonyldiimidazole (CDI) as cross linker[41–43]. As shown in nuclear magnetic resonance (NMR) spectra (Supplementary Figs. 1–10 and Supplementary Tables 1–3), the conjugation took place between C1′ of mannose and C6 of γ-CD and the substitution rate was controlled at 1:1 (mannose: γ-CD). By comparing the chemical shifts of α-/β-anomers of mannose with the chemical shifts of M-γ-CD in $H_2O$, it was found that the α-anomer was formed in the M-γ-CD (Supplementary Table 3). The obtained M-γ-CD was used as host molecule to fabricate inclusion complex with RG by co-crystallization method (denoted as RG@M-γ-CD). Since host-guest complexation was a dynamic process of non-covalent molecular recognition, heating and stirring were conducted to potentiate the non-covalent interactions and accelerate the formation of inclusion complex[44]. After complexation, RG@M-γ-CD was characterized using NMR spectroscopy and mass spectrometry (MS) to demonstrate the structure of host-guest inclusion complex. In [1]H and [13]C NMR spectra of RG@M-γ-CD (Supplementary Figs. 11–15), peaks of M-γ-CD and RG were coexisted and full assignment was performed to confirm the integrity of RG@M-γ-CD (Supplementary Tables 4 and 5). In 2D nuclear overhauser effect (NOE) spectrum (Fig. 2d and Supplementary Fig. 16), signals of NOE correlation

were detected between (1) amide protons of RG (H17″ and H19″) and glycogen of M-γ-CD (H1 (H1′)-H4 (H4′)); (2) benzene protons of RG (H2″, H24″ and H25″) and glycogen of M-γ-CD (H3 (H3′)), reflecting the complexation between host and guest molecules. MS analysis was performed to further determine molecular weight (Mw) of RG@M-γ-CD inclusion complex. The synthesized M-γ-CD yielded a evident signals at $m/z$ 1502.4606, $m/z$ 1525.4503 and $m/z$ 1541.4243, relative to Mw [M-γ-CD], adducts of Mw $[M + Na]^+$ and Mw $[M + K]^+$, respectively, confirming the mono-substitution (Fig. 2b and Supplementary Fig. 6). There was a strong signal at $m/z$ 1965.6228 in the MS spectrum of RG@M-γ-CD, which corresponded to Mw [M-$H_2O$-H]$^-$ (Fig. 2c and Supplementary Fig. 13). All these results pointed to the fact that the host-guest complexation was achieved between M-γ-CD and RG at 1:1 of stoichiometry.

After host-guest complexation, RG@M-γ-CD was found to spontaneously undergo secondary assembly in aqueous solution, forming a high-order topology with channel architecture (CNPs). The channel-type topology could be confirmed by XRD diffractogram in which RG@M-γ-CD CNPs displayed characteristic diffraction peaks at 2θ = 7.9°, 15.4°, 16.2°, and 17.1° etc., which were distinct from those of RG or amorphous

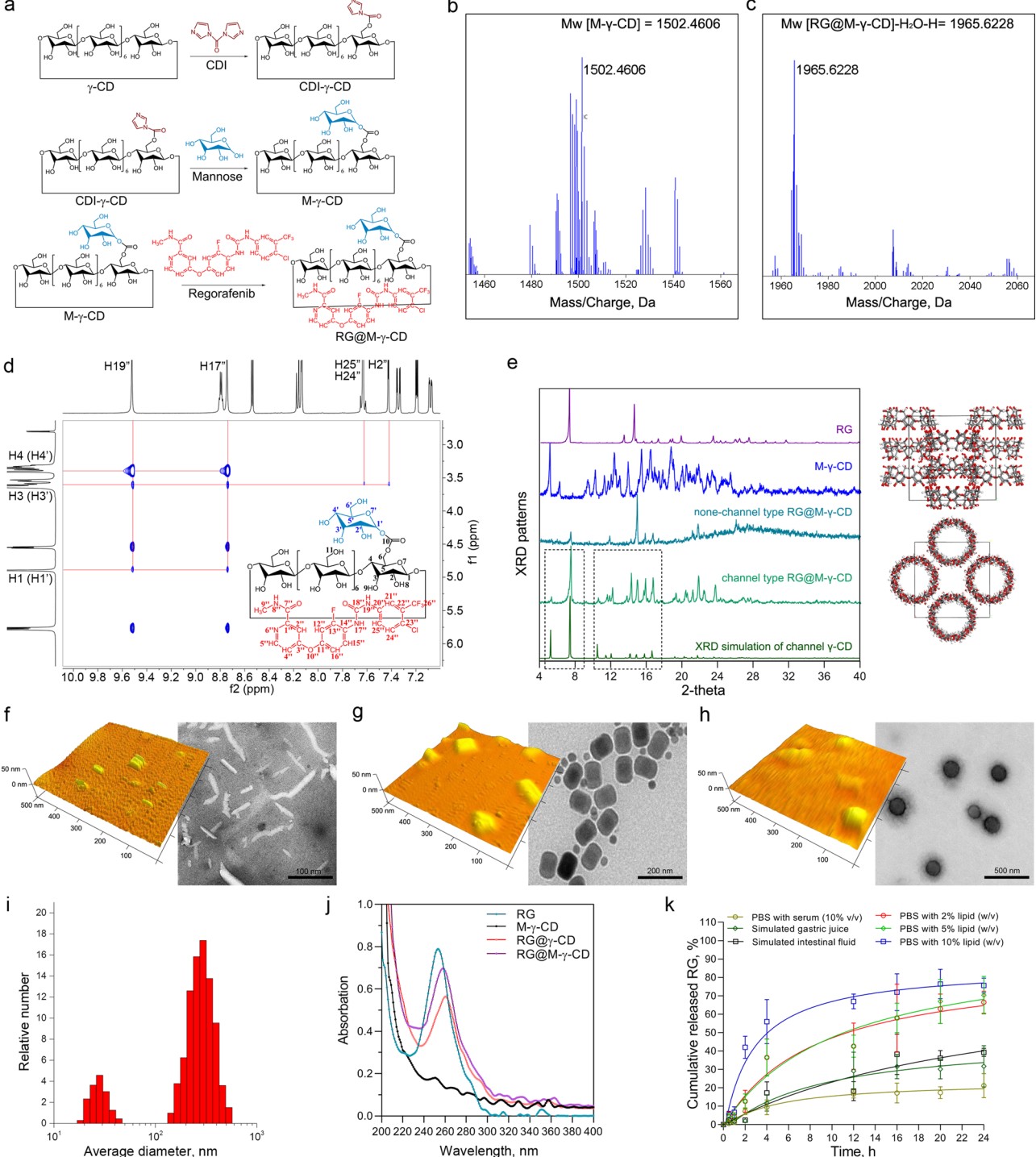

**Fig. 2 Synthesis and chemical characterization of RG@M-γ-CD. a** Synthetic procedure of RG@M-γ-CD, including activation of γ-CD by 1,1′-carbonyldiimidazole (abbreviated as CDI) (upper), functionalization of γ-CD with mannose (middle) and construction of inclusion complex, RG@M-γ-CD (lower). **b**, **c** ESI-TOF-MS analysis on M-γ-CD and RG@M-γ-CD. **d** Partial magnification of 2D NOESY spectrum of RG@M-γ-CD. **e** XRD patterns of RG, M-γ-CD, none channel-type RG@M-γ-CD, channel-type RG@M-γ-CD, and stimulated channel-type γ-CD. Schematic illustration of γ-CD stacking with channel-type architecture. Tracing observation of the assembly process by TEM and AFM. **f** Nanorods with a size of 20–50 nm; **g** nanobricks with a size of 50–100 nm, and **h** nanospheres with a size of 100–300 nm. **i** DLS results of the RG@M-γ-CD CNPs. **j** UV-vis spectra in methyl alcohol of free RG, M-γ-CD, RG@γ-CD, and RG@M-γ-CD. **k** Controlled release profiles of RG@M-γ-CD CNPs under different conditions. $N = 3$ replicates in each group. Data were expressed as means ± SD. Source data are provided as a Source Data file.

RG@M-γ-CD (Fig. 2e). These peaks were proven to reflect the channel packing in the tetragonal space group P42₁2, driven by the device motif of γ-CD[45–47]. In the channel architecture, inclusion molecules of CD were packed linearly on top of each other by hydrogen bonding, assembling into "infinite" channels including guest molecules. Thermal gravimetric and differential scanning calorimetry thermograms of RG@M-γ-CD CNPs indicated the disappearance of intrinsic endotherm of RG,

probably because of the crystalline rearrangement during complexation and assembly (Supplementary Fig. 17a, b).

We traced the assembling process using transmission electron microscopy (TEM) and atomic force microscopy (AFM) (Fig. 2f–h). RG@M-γ-CD first self-assembled into rod-like structures (20–50 nm). With stirring and ultrasonic treatment, nanorods proceed to stack in three dimensions, transforming into nanobricks (50–100 nm). They eventually formed nanospheres, primarily distributed in 100–300 nm. Dynamic light scattering (DLS) measurement was in good agreement with the morphological observation (Fig. 2i). UV-vis spectrum of RG@M-γ-CD CNPs displayed a characteristic absorbance at 260 nm with extinction coefficient of $\sim 2 \times 10^4 \, M^{-1} \, cm^{-1}$, presenting a hypochromatic shift of 10 nm as compared to equivalent RG. This discrepancy was caused by the different state of RG (free/entrapped) (Fig. 2j).

The stability of RG@M-γ-CD CNPs was assessed by DLS and TEM in various conditions (PBS with 10% serum, simulated digestive juice). After 24 h of incubation in 37 °C, the particle size distribution of CNPs kept stable in PBS with serum (pH 7.4) (Supplementary Fig. 17c). In simulated gastric (pH 1.4) and intestinal fluid (pH 7.8), the median diameter migrated to 90–110 nm owing to the partial dissociation of CNPs (Supplementary Fig. 17d). RG release in serum-containing PBS and stimulated digestive juices was next monitored over 24 h (Fig. 2k). The cumulative drug release profile confirmed the stability of RG@M-γ-CD CNPs in serum-containing PBS, with release ratios ranging from 16.2 to 21.4%. In comparison, the drug release was elevated to 24.3–35.6% in stimulated digestive juices. In the presence of α-amylase (100 U/mL), release ratio immensely increased, reaching to 76.2–85.4% (Supplementary Fig. 18). As a cyclic oligosaccharides, γ-CD could be degraded by α-amylase[48], which would accelerate CNP dissociation and drug release. Interestingly, RG@M-γ-CD CNPs exhibited a sustained release in lipophilic environment. Because of the amphiphilicity of CD, the non-covalent channel-type architecture is susceptible to disruption by lipophilic molecules, resulting in the collapse of CNPs and subsequent drug release.

**In vitro targeted drug delivery and antitumor effect.** Given the fact of CRC cells expressing MR, mannose modification of γ-CD could sever as an effective targeting strategy[28]. To examine the targeting potential, Rhodamine (Rho) was employed as an indicator and formulated into CNPs with M-γ-CD or γ-CD (Supplementary Figs. 19–30 and Supplementary Tables 6–9). Then the cell internalization of Rho@M-γ-CD and Rho@γ-CD CNPs was studied using confocal laser scanning microscopy (CLSM) and fluorescence activated cell sorting (FACS). As shown in Fig. 3a, b, both CNPs could be taken up by CRC cells (CT26 and HT29) within 2 h. In comparison, Rho@M-γ-CD group displayed a higher intracellular fluorescence accumulation, implying a more efficient internalization (Fig. 3c). When the cells were pretreated with free mannose, the fluorescence accumulation induced by internalized Rho@M-γ-CD significantly decreased (Supplementary Fig. 31), which confirmed that the CRC targeting of M-γ-CD-based CNPs was derived from the specific binding between the mannose groups of CNPs and the MR of CRC cells.

We next evaluated the in vitro antitumor activity of RG@M-γ-CD towards CRC cells (Supplementary Figs. 32 and 33). The CRC cells were treated with RG, M-γ-CD, mixture of RG and M-γ-CD (denoted as Mix), RG@γ-CD and RG@M-γ-CD at equivalent concentrations of RG. Within 4 h, these formulations showed similar lethal effects. After 12 h of treatment, the CNP formulations (RG@γ-CD and RG@M-γ-CD) were found to induce significantly higher proportion of cell death compared with free RG (Fig. 3d). Particularly, the half maximal inhibitory

concentration (IC50) of RG@M-γ-CD CNPs was remarkably decreased to 2.63–3.04 μM (Supplementary Tables 10 and 11). The incremental inhibitory effect of the CNPs was derived from their tailored nano-construction for cell uptake and sustained drug release in CRC cells. To characterize the intracellular drug release kinetics, RG@γ-CD and RG@M-γ-CD CNPs were incubated with cytoplasm extract (~20% v/v in PBS) and the released RG was monitored by HPLC (Supplementary Fig. 34). Within 12 h, 64.4 ± 11.7% and 71.5 ± 6.6% of RG was released from RG@γ-CD and RG@M-γ-CD, respectively. Coupled with the active CRC-targeting capacity, RG@M-γ-CD CNPs significantly elevated the drug availability of RG.

The inhibition mechanism of RG@M-γ-CD CNPs was next elucidated using western blot and FACS. The investigation of ERK activation indicated that RG@M-γ-CD treatment decreased ERK phosphorylation of CRC cells by 48.6 ± 7.5% compared with control, even at low concentrations (Supplementary Fig. 35), which implied a potent inhibition on the proliferation of CRC cells. Bromodeoxyuridine (BrDU) assay confirmed that RG@M-γ-CD CNP treatment markedly reduced the proliferation of CRC cells compared with other treatments (6.9 ± 3.3% and 12.8 ± 4.5% for CT26 and HT29, respectively). Annexin V/PI assays further indicated that RG@M-γ-CD CNP treatment efficiently compromised the survival of CRC cells, inducing higher proportion of apoptosis compared with other groups (47.7 ± 5.3% and 40.8 ± 7.4% for CT26 and HT29, respectively) (Fig. 3e–h). These results demonstrated that RG@M-γ-CD CNPs could enhance the kinase-inhibitory effect of RG and effectively suppress the malignancy of CRC cells.

VEGF/VEGFR signaling pathway contributes to angiogenesis and metastasis, which is a critical therapeutic target in CRC. Thus, the antiangiogenic capacity of RG@M-γ-CD was assessed by tube formation assay. Human umbilical vein endothelia cells (HUVECs) were seeded in the 3D Matrigel Matrix and treated with different formulations. Within 6 h, abundant tube-branches were observed in control and M-γ-CD groups (Fig. 3i). In comparison, treatment with RG or Mix exhibited moderate anti-angiogenesis, reducing the tube formation by 25.7 ± 8.2%. Obvious reduction of tube formation was achieved by RG@γ-CD and RG@M-γ-CD CNPs, particularly RG@M-γ-CD CNPs exerted the strongest antiangiogenic effect by decreasing the tube number by 72.5 ± 10.7%. Consistent with the angiogenesis inhibition, RG@γ-CD and RG@M-γ-CD CNPs significantly blocked the activation of receptor tyrosine kinases VEGFR-2 and PDGFR-β, while RG or Mix failed to achieve this goal (Fig. 3j). Together, these data indicated that M-γ-CD-derived CNP formulation greatly improved the drug activity of RG, and the host-guest assembled nanomedicine exerted antitumor effects by suppressing CRC cell survival and interdicting angiogenesis, bilaterally.

**Regulation of RG@M-γ-CD on macrophages in vitro.** Based on the previously reported anti-inflammatory property of CDs, we next investigated whether RG@M-γ-CD CNPs could alleviate inflammation by targeting macrophages. We isolated peritoneal macrophages (PMs) from C57BL/6 mice and the internalization by PMs was first explored using CLSM and FACS (Fig. 4a, b). PMs were first stained with green fluorescent anti-CD206 antibody to visualize the MR, then were treated with Rho@M-γ-CD and Rho@γ-CD for 2 h. As expected, the profile of PMs was portrayed by green fluorescence, confirming PMs expressing MR on their surfaces. Intracellular red fluorescence was induced by the internalized CNPs. In comparison, fluorescence accumulation of Rho@M-γ-CD group was significantly higher than that of Rho@γ-CD group. Similar tendency was obtained by FACS that

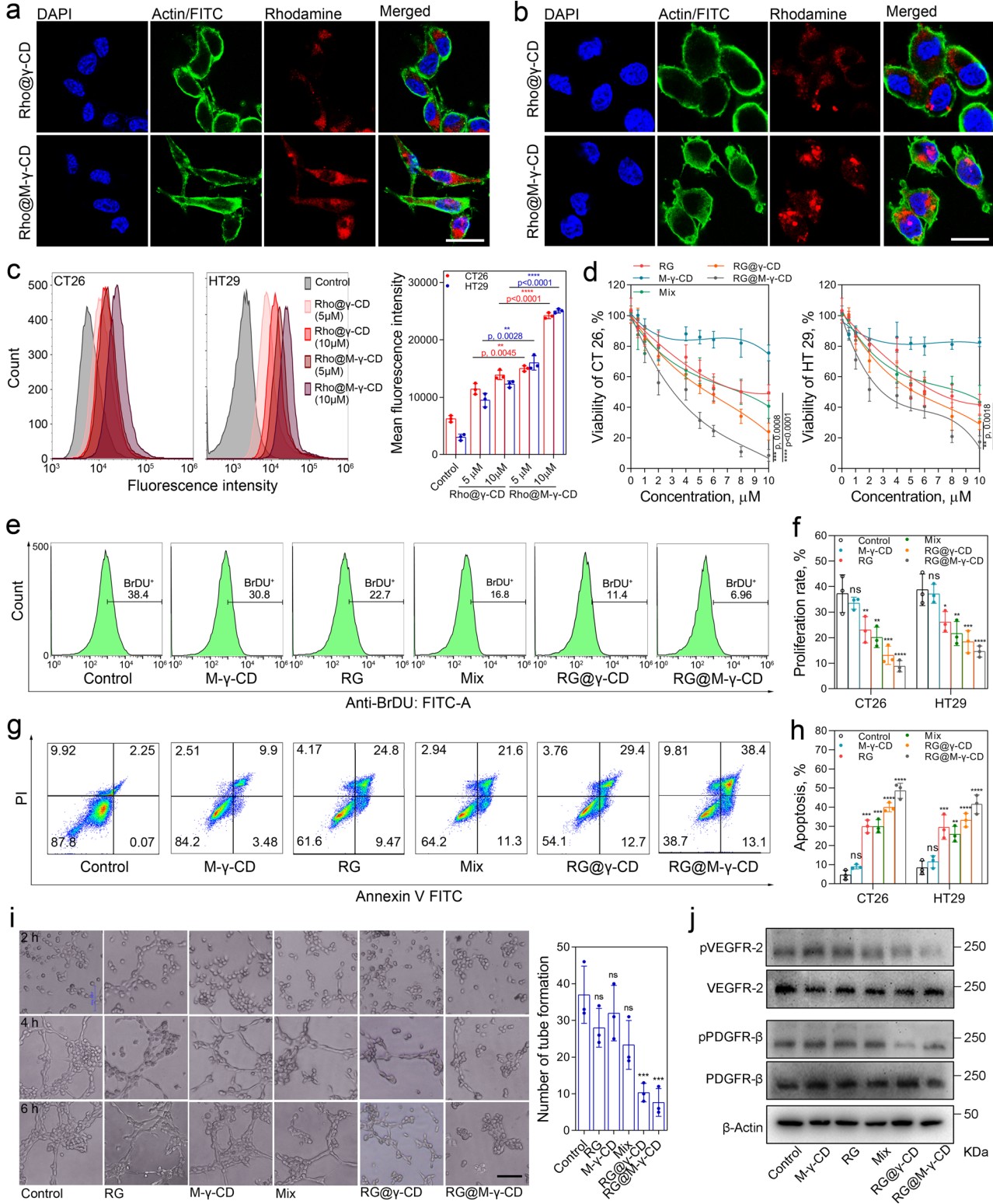

the mean fluorescence intensity of Rho@M-γ-CD increased by 88–93% compared with Rho@γ-CD group. The results indicated the MR-targeting capacity of M-γ-CD-based CNPs by which the Rho@M-γ-CD was prone to be taken up by macrophages.

Next, PMs were subjected to different treatments for 24 h (equivalent RG content, 0.5 μM), followed by stimulation with lipopolysaccharide (LPS) to evoke the secretion of inflammatory cytokines (Fig. 4c). Quantitative RT-PCR and ELISA assays were

conducted to determine the levels of typical CRC-related inflammatory cytokines[49], including IL-1β, IL-6, and TNF-α (Fig. 4d and Supplementary Fig. 36). In comparison with controls, M-γ-CD, Mix, RG@γ-CD, and RG@M-γ-CD treatments resulted in the downregulation of IL-1β, IL-6, and TNF-α expressions to varying degrees. Particularly, RG@M-γ-CD treatment displayed the significant downregulation of *IL-1β, IL-6,* and *TNF-α*, decreasing mRNA levels by 14–22-fold. Without

**Fig. 3 In vitro antitumor effect of RG@M-γ-CD CNPs.** Cell internalization of CNPs. **a**, **b** Representative CLSM images of the CT26 and HT29 cells cultured with Rho@M-γ-CD and Rho@γ-CD CNPs (equivalent Rho, 5.0 μM) for 2 h. Blue fluorescence indicated nuclear staining with DAPI; green fluorescence indicated β-actin staining with FITC-antibody; red fluorescence presented the internalization of CNPs. Scale bar, 20 μm. **c** FACS analysis and statistical result of intracellular fluorescence intensities induced by internalized Rho@M-γ-CD and Rho@γ-CD (equivalent Rho, 5.0 μM and 10.0 μM) in CT26 and HT29 cells ($2 \times 10^4$ cells/sample). $N = 3$ biological replicates in each group. Data were presented as means ± SD. Statistical significance was calculated using two-tailed unpaired $t$-test (Rho@M-γ-CD vs. Rho@γ-CD at the same Rho concentration, the exact $p$ values were indicated in the figure). **d** In vitro cytotoxicity of different formulations (equivalent RG, 0.5–10.0 μM) towards CT26 and HT29 cells with 12 h of treatment. $N = 5$ biological replicates in each group. Data were presented as means ± SD. Statistical significance was calculated using two-way ANOVA followed by Tukey's multiple comparison test (the exact $p$ values were indicated in the figure). Proliferation inhibition and apoptosis induction of different formulations (equivalent RG, 2.0 μM) was evaluated in CT26 and HT29 cells by BrdU assay and Annexin V-PI assay, respectively. **e** Representative FACS analysis and **f** statistical result of proliferation rate for treatment groups. $N = 3$ biological replicates in each group. Data were presented as means ± SD. **g** Representative FACS analysis and **h** statistical result of apoptosis rates for treatment groups. $N = 3$ biological replicates in each group. Data were presented as means ± SD. **i** Representative images of in vitro tube formation by HUVECs treated with different formulations for 6 h (equivalent RG, 2.0 μM). Scale bar, 50 μm. $N = 3$ biological replicates in each group. Data were expressed as means ± SD. Statistical significance for **f**, **h**, and **i** was calculated using one-way ANOVA followed by Dunnett's multiple comparison test (*$p \leq 0.05$, **$p \leq 0.01$, ***$p \leq 0.001$, ****$p \leq 0.0001$, treatment group vs. control, the exact $p$ values were indicated in the Source Data file). **j** Western blot analysis of phosphorylation of VEGFR-2 and PDGFR-β in HUVECs treated with different formulations (equivalent RG, 2.0 μM, 6 h). Images were representative of three independent experiments. Source data are provided as a Source Data file.

mannose modification, RG@γ-CD CNPs showed a relatively weaker inhibition as compared to RG@M-γ-CD CNPs. Conversely, free RG had little inhibitory effect on these cytokines, highlighting the key role of M-γ-CD in anti-inflammation. To further characterize the regulatory effects of RG@M-γ-CD upon PMs, the activation of NF-κB, a crucial transcription factor that regulates inflammation, was evaluated. The phosphorylation of p65 (p-p65) was determined by western blot. As shown in Fig. 4e, M-γ-CD significantly inhibited p-p65 compared with control. Moreover, the anti-inflammatory effect of M-γ-CD was implanted into RG@M-γ-CD CNPs which also showed a potent inhibition on p-p65. Consistent with the concept that CDs exerted the anti-inflammation functions by mediating LXR transcription[39], M-γ-CD and RG@M-γ-CD was also found to augment the LXR target gene Abca1 in PMs. Compared with RG@M-γ-CD CNPs, RG@γ-CD CNPs exhibited a suboptimal mediation on p65 phosphorylation, likely owing to the lack of MR-targeting capacity. Together, these data demonstrated that nano-engineering of RG@M-γ-CD CNPs maintained and further amplified the anti-inflammation properties of CDs, showing potentials to alleviate inflammation for CRC treatment.

Once activated by CRC cells, TAMs primarily present M2 phenotype and devote to establish a tumor-supportive TME by producing a series of cytokines, such as VEGFs, PDGFs, and MMPs[46,50]. Since RG was reported to reprogram the TME by suppressing M2 activation, we hypothesized that the channel-type nano-formulation could enhance this effect. To examine this possibility, PMs were subjected to different treatments for 24 h, subsequently incubated with IL-4 or supernate from CT26 cells to trigger M2 polarization. Then the expressions of M2-related markers (arginase-1 [Arg-1], IL-10, and CD206) were quantified to evaluate the inhibition on M2 polarization. RG decreased the mRNA levels of M2-related markers by only 1.4-fold compared with control. As a CNP formulation, RG@γ-CD showed a higher inhibition of M2 polarization (1.6-fold). As an upgraded CNP system, RG@M-γ-CD CNPs significantly decreased the M2 markers by 2.8–3.2-fold (Fig. 4f). We also investigated whether RG@M-γ-CD could deactivate M2 cells by reducing their production of tumor-supportive cytokines. PMs were first subjected to different formulations and then activated to secrete the tumor-supportive cytokines (i.e., VEGF-B, PDGF-α, and MMP-9). RT-PCR and ELISA quantification revealed that RG@M-γ-CD treatment remarkably downregulated these cytokines compared with control. In comparison, RG, Mix and RG@γ-CD treatments showed relatively weaker reduction, while M-γ-CD had little effect (Fig. 4g and Supplementary Fig. 36). Overall, our results demonstrated that RG@M-γ-CD CNPs

exhibited a dual regulation towards macrophages by reducing inflammatory cytokines and suppressing M2 activation.

**Therapeutic efficacy and safety of RG@M-γ-CD in CAC mouse model.** CAC is a subtype of CRC and characterized by chronic inflammation and accelerated neoplasia, thus anti-inflammation therapy combined with conventional therapies can be used to prevent CAC[51]. Encouraged by the promising in vitro performance of RG@M-γ-CD CNPs, we then evaluated the in vivo therapeutic efficacy on colon tumorigenesis in the CAC mouse model. We first assessed the intestinal retention of CNPs via oral gavage using free Rho, Rho@γ-CD and Rho@M-γ-CD as indicators. Compared with Rho@γ-CD and Rho@M-γ-CD CNPs, free Rho exhibited a low intestinal accumulation and quick attenuation within 24 h post administration. For both CNPs, Rho fluorescence gradually accumulated in intestines, reaching the highest intensity at ~12 h post administration. After 24 h, considerable fluorescence could still be detected, indicating that CD-based nano-formulation facilitated intestinal retention (Fig. 5a). Further ex vivo fluorescence imaging displayed that CNPs were mainly distributed in intestines with relatively low distribution in stomach. Based on the previously demonstrated stability in simulated gastric juice, this observed suggested that the CNP formulation could protect the drug from acid hydrolysis in stomach and was prone to accumulate in intestines. Besides, weak fluorescence intensity in liver and kidney might suggest a reduction of hepatic first-pass metabolism (Fig. 5b and Supplementary Fig. 37). Taken together, these results indicated that the CNP formulation could be an ideal candidate for colonic drug delivery.

We next examined the in vivo antitumor efficacy of RG@M-γ-CD CNPs in the CAC mouse model via oral administration. A chemically induced colon tumor model was established in C57BL/6 mice using azoxymethane (AOM) and dextran sodium sulfate (DSS) (Fig. 5c)[52]. During tumor development, the mice presented reduced body weights and colitis pathology of hematochezia, hypothermia, and shortened colons, with disease activity index (DAI) increasing to 10 (Fig. 5d, e). Treatment with M-γ-CD, Mix, RG@γ-CD, and RG@M-γ-CD alleviated hematochezia, preserved colonic length, and lowered DAI when compared with untreated mice, indicating mitigation of colitis. Particularly, RG@M-γ-CD group displayed the lowest DAI (5.8 ± 0.5) and comparable length to normal colons. By contrast, RG treatment had little effect on colitis, DAI and colon length were similar to that of untreated mice (Fig. 5f). Tumor loadings in colons were then counted and measured (Fig. 5g, j). M-γ-CD treatment displayed little inhibition of tumor progression, although it remitted colitis. RG

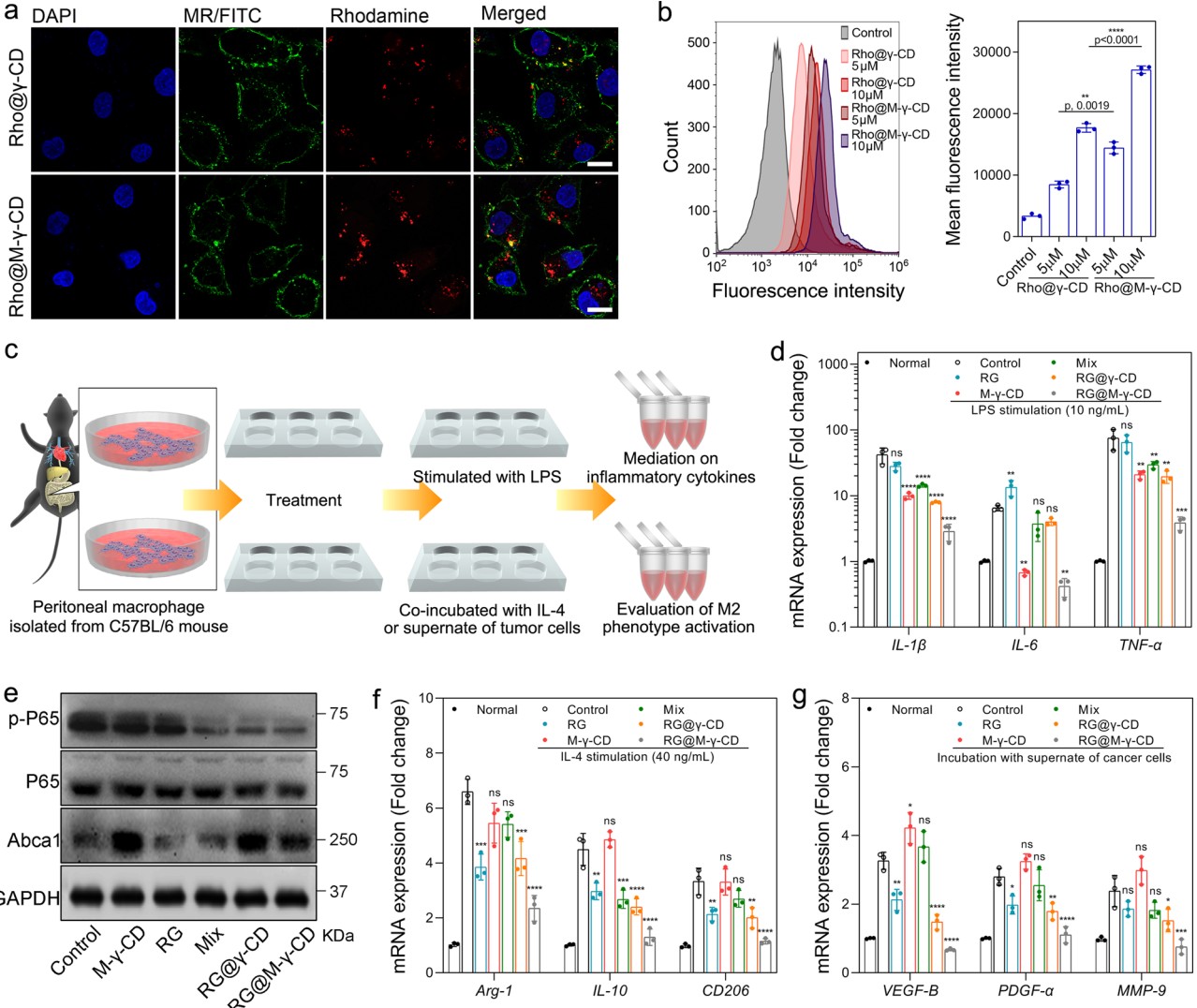

**Fig. 4 In vitro regulatory effect of RG@M-γ-CD upon macrophage.** Cell internalization of CNPs by macrophages. **a** Representative CLSM images of the peritoneal macrophages cultured with Rho@M-γ-CD and Rho@γ-CD CNPs (equivalent Rho, 5.0 μM) for 2 h. Blue fluorescence indicated nuclear staining with DAPI; green fluorescence indicated mannose receptors staining with FITC-antibody; red fluorescence presented the internalization of CNPs. Scale bar, 20 μm. **b** FACS analysis and statistical result of intracellular fluorescence intensities induced by internalized Rho@M-γ-CD and Rho@γ-CD (equivalent Rho, 5.0 and 10.0 μM) in peritoneal macrophages ($2 \times 10^4$ cells/sample). $N = 3$ biological replicates in each group. Data were presented as means ± SD. Statistical significance was calculated using two-tailed unpaired $t$-test (Rho@M-γ-CD vs. Rho@γ-CD at the same Rho concentration, the exact $p$ values were indicated in the figure). **c** Schematic illustration of peritoneal macrophages isolation, stimulation, and phenotype evaluation. Peritoneal macrophages were treated with different formulations for 24 h, followed by incubation with LPS, IL-4, or tumor supernate for 2 h. Then the cells and culture media were collected for phenotype evaluation. **d** mRNA expression levels of pro-inflammatory cytokines by peritoneal macrophages treated with different formulations (equivalent RG, 0.5 μM). $N = 3$ biological replicates in each group. Data were presented as means ± SD. **e** Western blot analysis of p-p65, p65, and abca1 expressions in macrophages under different treatments. Images were representative of three independent experiments. **f** mRNA expression levels of M2-associated markers by macrophages under different treatments (equivalent RG, 0.5 μM). $N = 3$ biological replicates in each group. Data were presented as means ± SD. **g** mRNA expression levels of VEGF-B, PDGF-α, and MMP-9 by macrophages under different treatments (equivalent RG, 0.5 μM). $N = 3$ biological replicates in each group. Data were presented as means ± SD. Statistical significance for **d**, **f**, and **g** was calculated using one-way ANOVA followed by Dunnett's multiple comparison test (*$p \leq 0.05$, **$p \leq 0.01$, ***$p \leq 0.001$, ****$p \leq 0.0001$, treatment group vs. control, the exact $p$ values were indicated in the Source Data file). Source data are provided as a Source Data file.

and Mix groups showed a reduction of tumor number; however, the tumors were mainly of large size (8–13 mm³). This result was attributed to the biopharmaceutical limitations of RG which obstructed the antitumor efficacy. RG@γ-CD CNPs improved the antitumor effect of RG and the average volume of tumors in RG@γ-CD group was reduced below 10 mm³. In comparison, RG@M-γ-CD treatment further decreased tumor number and volume (below 6 mm³) owing to the targeting capacity to colon tumor cells, showing a significant therapeutic efficacy against

malignancy. Also, RG@M-γ-CD treatment significantly extended the survival time for the model mice as compared to other treatments (Supplementary Fig. 38). These results suggested that RG@M-γ-CD CNPs could be regarded as an ideal platform for the treatment of CAC mouse model. The non-covalent channel architecture offered the system stability and advantages in colon-specific drug delivery. Furthermore, this binary system also exerted a synergistic effect by alleviating inflammation and inhibiting tumor progression.

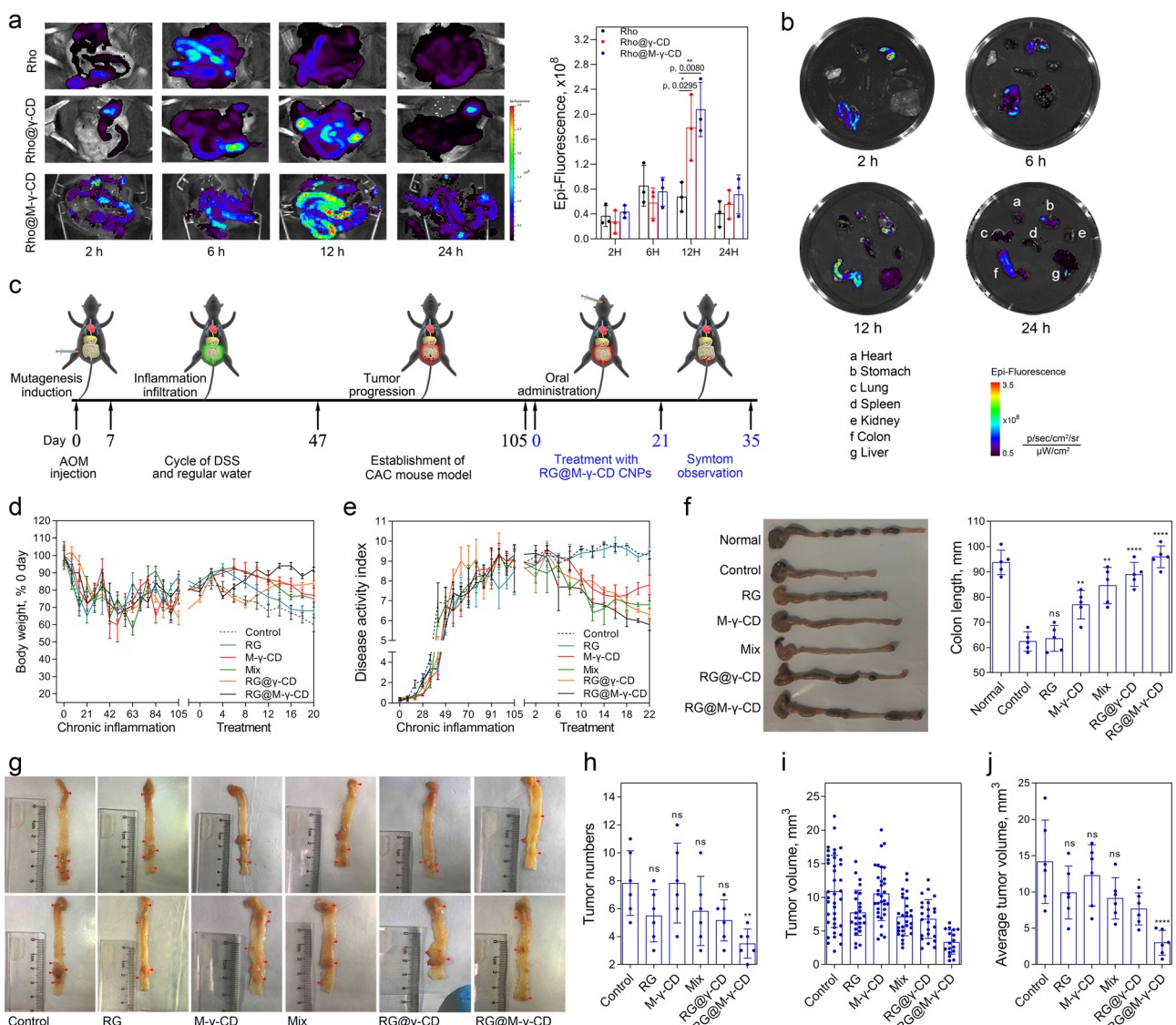

**Fig. 5 Therapeutic efficacy of RG@M-γ-CD in CAC mouse model.** Biodistribution of free Rho, Rho@γ-CD, and Rho@M-γ-CD CNPs was evaluated with ex vivo imaging in C57 mice via oral administration. **a** Representative ex vivo fluorescence images and mean fluorescence intensities within intestinal tissues at different time points post administration (equivalent Rho, 10 μg/g) Scale bar, 2.0 cm. N = 3 biologically independent mice in each group at each time point. Data were presented as means ± SD. **b** Representative ex vivo fluorescence imaging of main organs at 2 h, 6 h, 12 h, and 24 h post administration of Rho@M-γ-CD CNPs. Images were representative of three biologically independent mice. **c** Schematic illustration of CAC mouse model establishment based on AOM-DSS method. Changes of **d** body weight and **e** disease activity index for different treatment groups during modeling and antitumor experiment (equivalent RG, 10 μg/g). N = 10 biologically independent mice in each group. Data were presented as means ± SD. **f** Representative images of colons, and colon lengths from different treatment groups. N = 5 biologically independent mice in each group. Data were presented as means ± SD. **g** Representative images of colon tissues with tumor loading from different treatment groups. The red arrows indicated tumor areas. **h** Numbers of tumor loading, **i** tumor volumes, and **j** average tumor sizes for treatment groups. **g–j** N = 6 biologically independent mice in each group. Data were presented as means ± SD. Statistical significance for **a**, **f**, **h–j** was calculated using one-way ANOVA followed by Dunnett's multiple comparison test (*$p \leq$ 0.05, **$p \leq$ 0.01, ***$p \leq$ 0.001, ****$p \leq$ 0.0001, treatment group vs. control, the exact $p$ values were indicated in the Source Data file). Source data are provided as a Source Data file.

We further examined the safety profile of RG@M-γ-CD CNPs administered via oral route. Healthy mice were orally given with various formulations at a drug dosage of 10 μg/g three times. At 21 day post administration, RG@M-γ-CD CNPs-treated mice showed comparable levels of biochemical markers associated with liver and kidney functions compared with untreated mice. By contrast, administration with RG or Mix resulted in changes of some parameters, including alanine aminotransferase and aspartate aminotransferase, implying abnormal liver functions (Supplementary Fig. 39a). Histological analysis was performed to further evaluate the safety profile. H&E sections of liver of mice treated with RG or Mix revealed some vessels congested by erythrocyte, as well as edematous areas, indicating a certain degree of hepatic injury. By contrast, liver sections from RG@M-γ-CD group presented regular vessels and appearance, showing no abnormal variations (Supplementary Fig. 39b). These data suggested that M-γ-CD-based CNP system could reduce RG-induced hepatic toxicity, offering a safer strategy for oral drug administration.

**Antitumor mechanism of RG@M-γ-CD in CAC mouse model.** To further understand the mechanisms underlying the therapeutic activity of RG@M-γ-CD CNPs for CAC mouse model, we

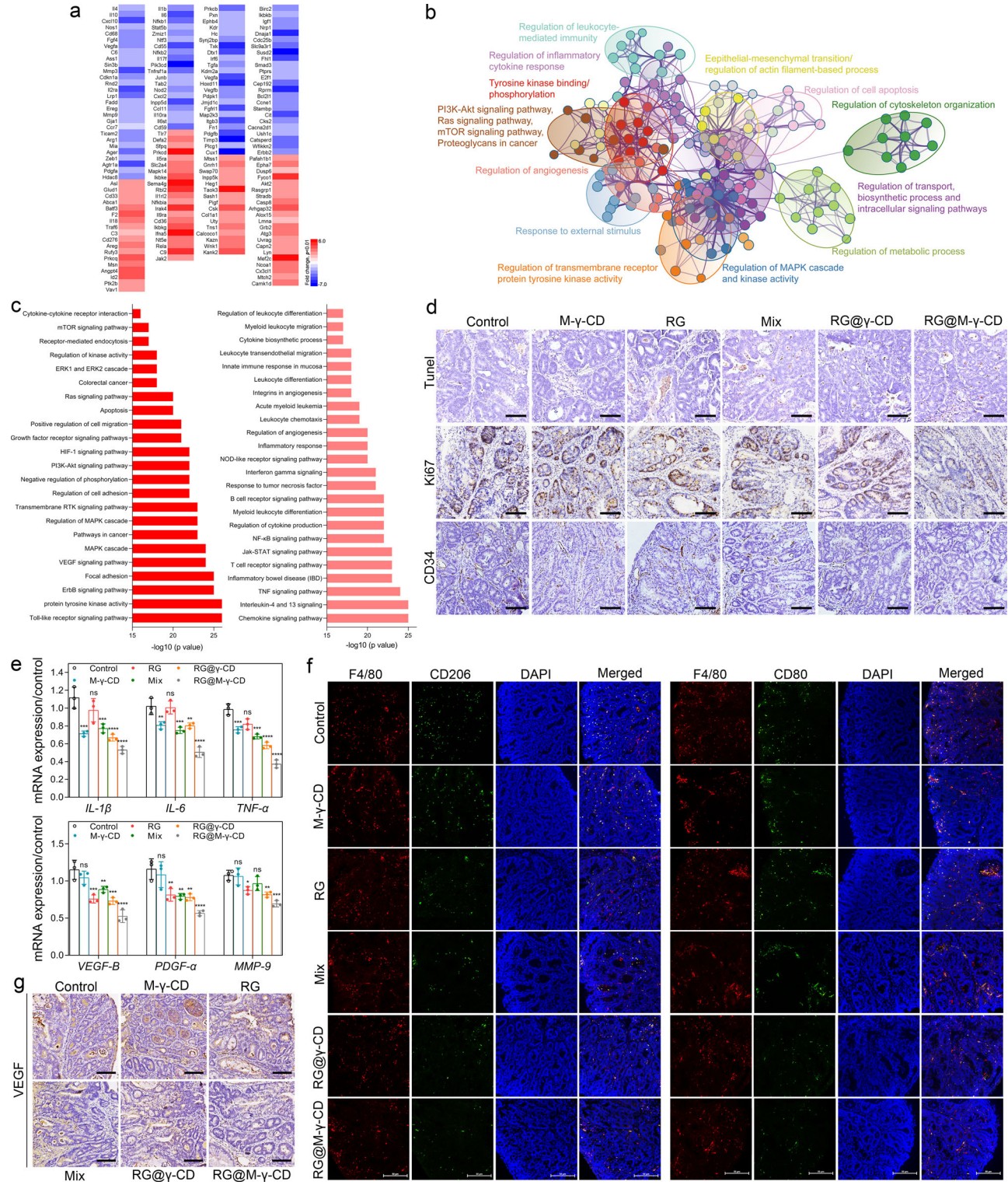

investigated the whole-genetic variations in CAC tumors from RG@M-γ-CD and PBS groups. A total of 587 differentially expressed genes were identified ($p < 0.01$, fold change cutoff > 1.5, hypergeometric test), including 202 upregulated genes and 385 downregulated genes (Fig. 6a and Supplementary Fig. 40). Differentially expressed genes were then analyzed by Gene Ontology enrichment analysis (GOEA) based on Biological Process and KEGG Pathway (Fig. 6b, c). Genes involved in transportation, metabolism, and cellular response to stimuli were upregulated

upon RG@M-γ-CD treatment, represented by GO terms "response to external stimulus", "receptor-mediated endocytosis", "regulation of vesicle-mediated transport", "glucose metabolic process", "response to organic cyclic compound", and "drug metabolic process", et al. These GO clusters profiled the mechanism of RG@M-γ-CD CNPs in internalization, transformation, and metabolism. Further interrogation of GOEA revealed that RG@M-γ-CD treatment negatively regulated many key genes related to GO terms of "protein tyrosine kinase activity",

**Fig. 6 In vivo antitumor mechanism of RG@M-γ-CD CNPs in CAC mouse model. a** Heat map of differential expression (DE) genes involved in TME mediation and tumor inhibition (RG@M-γ-CD vs. control). **b** GOEA of differential expression DE genes visualized as GO network, where nodes with different colors indicated the clusters based on connected GO terms. The clusters were annotated manually by a shared general term. **c** Histograms of DE genes associated with tumor inhibition (left part) and TME (right part) mediation based on GO annotation (−log10 (p value), 27–15). **a–c** N = 3 biologically independent mice in each group. Statistical significance was calculated using unpaired two-tailed t-test. Fold change ≥ 2.0 with p value ≤ 0.05 was used as threshold set for the analysis of DE genes. **d** Representative images of Tunel, Ki67, and CD34 staining sections of CAC tumor tissues from different treatment groups. Scale bar, 50 μm. Images were representative of three biologically independent mice in each group. **e** mRNA expression levels of inflammatory cytokines (IL-1β, IL-6, and TNF-α) and M2-related tumor-supportive cytokines (VEGF-B, PDGF-α, and MMP-9) in CAC tumor tissues from different treatment groups. N = 3 biologically independent mice in each group. Data were presented as means ± SD. Statistical significance was calculated using one-way ANOVA followed by Dunnett's multiple comparison test (*p ≤ 0.05, **p ≤ 0.01, ***p ≤ 0.001, ****p ≤ 0.0001, treatment group vs. control, the exact p values were indicated in in the Source Data file). **f** Representative immunofluorescence images of CD11b, CD80, and CD206 staining sections of CAC tumor tissues from different treatment groups. Scale bar, 50 μm. Images were representative of three biologically independent mice in each group. **g** Representative images of VEGF staining sections of CAC tumor tissues from different treatment groups. Scale bar, 50 μm. Images were representative of three biologically independent mice in each group. Source data are provided as a Source Data file.

"transmembrane receptor protein tyrosine kinase activity", "growth factor receptor signaling pathways" and "MAPK signaling pathway", among others. This negative regulation on tyrosine activity might be associated with the impaired angiogenesis, as well as the inhibition of cancer cells (such as "apoptosis" and "negative regulation on cell cycle"). In line with the GOEA result, immunohistochemical staining confirmed the systematic antitumor effect of RG@M-γ-CD CNPs (Fig. 6d). TUNEL staining revealed considerable cell apoptosis in the colon tumor sections from RG@M-γ-CD group. In comparison, RG, Mix, and RG@γ-CD groups showed sparse apoptotic cells, indicating limited suppression on malignancy. Similarly, Ki67 staining displayed that RG@M-γ-CD treatment resulted in a significant reduction on cell proliferation, while other treatments failed to achieve this goal. We also used CD34 staining to investigate the neovascularization in different groups. Vessel number in RG@M-γ-CD group was obviously reduced compared with other groups, indicating that RG@M-γ-CD treatment suppressed tumor angiogenesis.

GOEA also elucidated regulatory effects of RG@M-γ-CD CNPs on TME of CAC tumor. RG@M-γ-CD treatment significantly downregulated the gene production of inflammatory mediators, including Il1β, Il6, Il17, Tnfrsf, Nfκb1, as well as some toll-like receptors (TLRs) and TLR adapters. These genes play crucial roles in linking inflammation and cancer, as represented by terms, such as, "Toll-like receptor signaling pathway", "Jak-STAT signaling pathway", and "NF-κB signaling pathway". RT-PCR quantification showed that the expression of inflammatory cytokines in tumor loading colons was suppressed with RG@M-γ-CD treatment, confirming the remission of colon inflammation. It should be emphasized that M-γ-CD treatment also significantly reduced the expression level of inflammatory cytokines, suggesting that M-γ-CD could be applied as a drug carrier with anti-inflammatory property (Fig. 6e). Furthermore, genes associated with immune cell differentiation, including Ager, Cxcl1, CD59, Arg1, Il4, Mmp8, Ym2, and Csf1 et al., were regulated after RG@M-γ-CD treatment. Regulation on differentiation, represented by terms, such as, "Interleukin-4 and 13 signaling", "Myeloid leukocyte differentiation", and "Interferon gamma signaling", might impact TAM polarization. To evaluate the TAM profile, immunofluorescence analysis was performed on colon tumor sections from different treatment groups using anti-F4/80, anti-CD206, and anti-CD80 antibodies. As shown in Fig. 6f and Supplementary Fig. 41, there was a significant decrease in M2 TAMs (F4/80+ CD206+ cells) in the colon tumors from the CNP groups (RG@γ-CD and RG@M-γ-CD) as compared to the colon tumor from control group. Meanwhile, reduced infiltration of M1 TAMs (F4/80+ CD80+ cells) was observed in RG@γ-CD and RG@M-γ-CD groups. FACS analysis also

indicated that the expression of CD206 in colon lamina propria macrophages (F4/80+ CD11b+ subset) of tumor regions was significantly reduced (Supplementary Fig. 42). This illustrated that RG@γ-CD and RG@M-γ-CD CNPs regulated the TME of CAC tumor by deactivating TAMs, and the RG@M-γ-CD CNPs showed the most potent deactivation effect.

Consistently, the expression levels of M2-related tumor-supportive factors, VEGF-B, PDGF-α, and MMP-9, were determined to be significantly downregulated after RG@M-γ-CD treatment (Fig. 6e). Since M2-related cytokines were reported to be associated with drug resistance and connect with poor prognosis, efficient M2 deactivation of RG@M-γ-CD CNPs might have great significance in improving therapeutic response. A reduction of VEGF-B protein was also observed in tumor sections from RG@M-γ-CD group, the positive areas in extracellular depots were dramatically decreased compared with other groups (Fig. 6g). Collectively, these findings help establish the mechanism by which RG@M-γ-CD synergistically targets malignant cells and heterogeneous TAMs combating CAC.

**Therapeutic efficacy of RG@M-γ-CD in CT26 mouse model.** We next evaluated the antitumor efficacy of RG@M-γ-CD CNPs in CT26 tumor model (Fig. 7a). Pharmacokinetics and biodistribution were first investigated by monitoring plasma drug concentration and quantifying drug accumulation in tissues. CT26 tumor-bearing mice were injected with RG, RG@γ-CD, and RG@M-γ-CD via I.V. injection at RG dosage of 10 μg/g. Plasma concentration profiles demonstrated that CNP nano-medicines (RG@γ-CD and RG@M-γ-CD) exhibited 2.38–3.45-fold higher AUC₀-₂₄ compared with RG, confirming that prolonged blood circulation and lower elimination was achieved by channel-type nano-formulation (Fig. 7b). Biodistribution analysis indicated that RG@γ-CD and RG@M-γ-CD CNPs rapidly accumulated at tumors at 2 h post injection, RG concentration in the tumor tissue was 1.8–2.7-fold higher than that of free drug. After 24 h post injection, free RG was eliminated and showed negligible tumor accumulation. In comparison, RG@γ-CD and RG@M-γ-CD CNPs still maintained a relatively higher drug concentration (1.4–2.3 μg/g) at tumor site, owing to the enhanced permeability and retention effect of CNPs. Particularly, coupled with the active CRC-targeting effect, RG@M-γ-CD CNPs exhibited the highest tumor accumulation at 24 h post injection. Deceased exposure of RG@M-γ-CD to liver and kidney was also detected, which might reduce systemic toxicity of kinase inhibitor in these tissues (Fig. 7c). The results indicated that the M-γ-CD-derived non-covalent nano-formulation effectively optimized the pharmacokinetics and biodistribution of RG. The appropriated particle size and stable channel construction of RG@M-γ-CD was favorable for systemic circulation and tumor penetration[53]. Moreover,

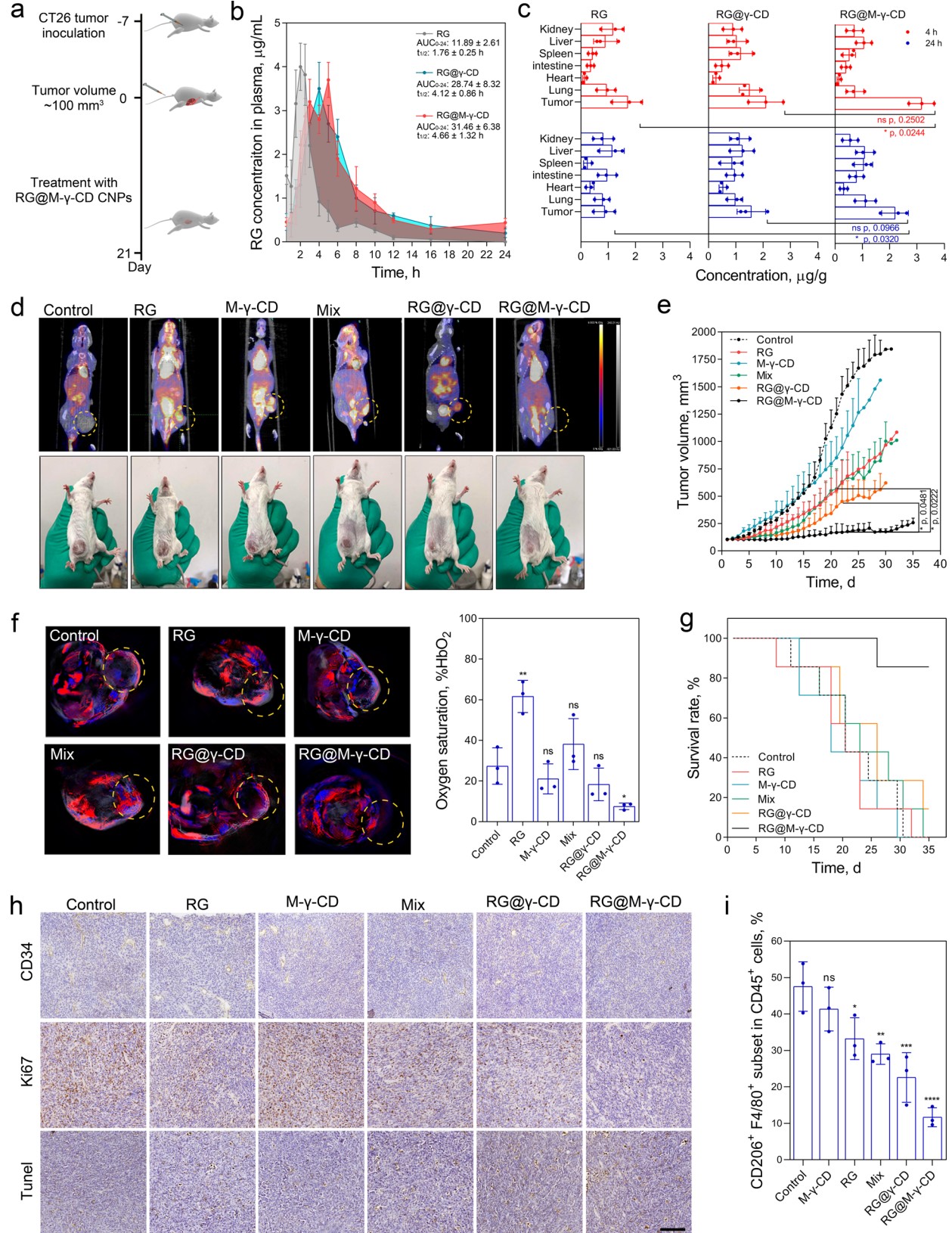

mannose-mediated targeting strategy further potentiated tumor accumulation of RG@M-γ-CD CNPs.

Based on the pharmacokinetic and targeting advantages, we implemented the strategy into treatment of CT26 tumor (~100 mm³). CT26 tumor-bearing mice were randomly divided into six groups (n = 7/group) and treated with PBS, RG, M-γ-CD, Mix, RG@γ-CD and RG@M-γ-CD every 2 days for 21 days. Positron emission tomography-computed tomography imaging was conducted to visualize the tumor progression (Fig. 7d and Supplementary Fig. 43). Multispectral optoacoustic tomography

**Fig. 7 Therapeutic efficacy of RG@M-γ-CD in CT26 CRC model. a** Schematic illustration of CT26 subcutaneous tumor model and treatment via tail intravenous injection (equivalent RG, 10 μg/g). **b** The pharmacokinetics and **c** biodistribution of RG, RG@γ-CD, and RG@M-γ-CD CNPs determined by HPLC at 4 h and 24 h post injection (equivalent RG, 10 μg/g). $N = 3$ biologically independent mice in each group at each time point. Data were presented as means ± SD. **d** Representative PET/CT imaging results and photos of mice CT26 tumor-bearing mice in different treatment groups at 22 day of treatment. **e** The tumor volume changes of CT26 tumor-bearing mice in different treatment groups. $N = 7$ biologically independent mice in each group. Tumor growth data were presented as means ± SD. **f** Representative photoacoustic imaging of the mice in different treatment groups and the degree of blood oxygen saturation (HbO$_2$) within tumors at 22 day of treatment. $N = 3$ biologically independent mice in each group. Data were presented as means ± SD. **g** Kaplan–Meier survival curves for treatment groups ($n = 7$). **h** Representative images of CD34, Ki67, and tunel staining sections of tumor tissues from different groups. Scale bar, 50 μm. Images were representative of three biologically independent mice in each group. **i** Flow cytometry analysis of TAM population (CD206$^+$ F4/80$^+$ subset gated on CD45$^+$ cell set) in different treatment groups. $N = 3$ biologically independent mice in each group. Data were presented as means ± SD. Statistical significance for **c**, **e**, **i** was calculated using one-way ANOVA followed by Dunnett's multiple comparison test (*$p \leq$ 0.05, **$p \leq$ 0.01, ***$p \leq$ 0.001, ****$p \leq$ 0.0001, treatment group vs. control, the exact $p$ values were indicated in the figure or in the Source Data file). Source data are provided as a Source Data file.

(MSOT) imaging technology was also implied to monitor the degree of tumor vascularization. Enlarged tumor burdens (1500–1800 mm$^3$ by 21 days) and rim enhancement of vascular pattern were observed in PBS and M-γ-CD group, indicating little therapeutic effects. RG and Mix treatments had a moderate inhibition on tumor growth (750–1000 mm$^3$) but failed to suppress the vascularization, and the tumor areas showed high blood oxygen saturation levels, resulting in rapid recurrence and poor prognosis (Fig. 7e, g). In comparison, treatment with RG@γ-CD and RG@M-γ-CD CNPs evidently suppressed tumor growth and vascularization (<600 mm$^3$). As an optimized nanomedicine, RG@M-γ-CD CNPs achieved the most effective malignancy inhibition and the tumor volume was restricted to ~200 mm$^3$.

Immunohistochemical staining was performed to investigate the inhibitory effect of RG@M-γ-CD on malignancy (Fig. 7h). TUNEL and Ki67 staining showed that RG@M-γ-CD treatment caused the highest apoptotic level and the lowest proliferating rate compared with other groups. In line with the MOST imaging, CD34 staining demonstrated that treatment with RG@M-γ-CD CNPs efficiently blocked tumor angiogenesis. To characterize the regulatory effect of RG@M-γ-CD CNPs on TAMs, FACS analysis of CD206$^+$ F4/80$^+$ subset gated on CD45$^+$ set within tumor tissues was performed. Tumors from different groups were excised, mechanically and enzymatically processed, then analyzed by flow cytometry. Compared with control, treatment with RG, Mix, RG@γ-CD or RG@M-γ-CD significantly decreased the TAM population in tumor tissue (CD206$^+$ F4/80$^+$ subset gated on CD45$^+$ set). Particularly, a ~5.0-fold decrease of CD206$^+$ F4/80$^+$ subpopulation was observed in RG@M-γ-CD group (Fig. 7i and Supplementary Fig. 44). Immunofluorescence analysis demonstrated that there was a significant decrease of infiltration of F4/80$^+$ CD206$^+$ cells within tumor from the mice treated with RG@M-γ-CD as compared to the tumor from the untreated mice (Supplementary Figs. 45–47). This confirmed the effect of RG@M-γ-CD CNPs in deactivating the TAMs in CT26 CRC model. Since TAM infiltration in tumor contributes to protumorigenic microenvironment and has been identified as an poor prognostic factor in a broad range of cancers[54], potent TAM deactivation of RG@M-γ-CD CNPs has significant implication in improving therapeutic outcome.

## Discussion
Host-guest assembly offers a bottom-up approach to fabricate functional materials and systems by organizing molecular building blocks in a non-covalent fashion. The application of CD-derived host-guest chaperone complexes and assemblies has been extensively developed in cancer theranostics[30,55,56]. We previously constructed a series of host-guest artificial chaperones using CDs and tyrosine kinase inhibitors (TKIs) with the aim of improving bioavailability and antitumor efficiency. However,

difficulties in manipulating the construction and morphology of CD chaperones resulted in suboptimal systemic circulation and undesirable drug leakage for in vivo use. Furthermore, lack of active targeting capacity compromised drug biodistribution. These led us to pursue innovation. Here we synthesized a mannose-modified γ-CD (M-γ-CD) as a CRC-targeting carrier. Theoretically, increasing mannose substitution on γ-CD would potentiate the targeting capacity, we found that high degree substitution could impair the complexation of γ-CD. Thus, the substitution was controlled at 1:1 to keep the balance between targeting capacity and inclusion ability. Hydrophobic RG was successfully encapsulated by M-γ-CD and the resultant chaperone complex showed an intriguing motif to assemble into CNPs. Structurally, RG@M-γ-CD CNPs were maintained and stabilized by non-covalent interactions of neighboring γ-CDs and the encapsulated RG. PXRD pattern confirmed the channel-type stacking mode of RG@M-γ-CD with P42$_1$2 space group (unit cell dimensions of $a = 23.759$ Å, $c = 23.069$ Å). Functionally, RG@M-γ-CD CNPs could be regarded as a non-covalent pro-drug that exerted synergistic effects combating CRC. With tailored morphology and CRC-targeting capacity, the CNPs improved drug delivery efficiency to targeted cells and therefore amplified the suppression of TKI on malignancy. Meanwhile, the non-covalent binding would not lesion the bio-function of CD and the CNPs were confirmed to initiate anti-inflammation mechanism through targeting macrophages. In addition to alleviating inflammation, M-γ-CD-derived CNP system elevated TKI's regulation on TAMs, manifesting enhanced efficiency in suppressing phenotypic polarization and reducing tumor-supportive factors. Such integrated regulation corroborated the superiority of host-guest system in which the functions of the whole were greater than the sum of its parts.

For in vivo application, RG@M-γ-CD CNPs inherited and developed the pharmaceutical advantages of CDs, not only improving RG's pharmacokinetics but also showing specialities for CRC drug delivery, such as protection in gastric area, retention in colorectum, and dual targeting to CRC cells and colonic macrophages. It was noteworthy that CNP system could protect the entrapped RG from extravasation during circulation via both oral and intravenous routes, and therefore reduced drug retention in liver and kidney as compared to free drug. This property of CNPs had important significance in improving bioavailability and safety for TKIs[57,58]. In the lipophilic colonic or intracellular environment, RG@M-γ-CD CNPs were gradually dissociated, concomitant with sustained release of loaded RG. By suppressing proliferation and blocking angiogenesis, the released RG efficiently abolished the communication between aberrant stroma and malignant cells. In addition, in vivo study on CAC mouse model revealed that RG@M-γ-CD CNPs could effectively relieved inflammation in colon, which was derived from the nanocarrier

M-γ-CD. Although M-γ-CD or Mix treatment attenuated manifestations relevant to DSS-induced colitis (including weight loss, DAI, colon length and hematochezia), it failed to suppress the malignancy. In most cases, anti-inflammation was not cytocidal or tumor-suppressive on its own and needed to be combined with other therapeutics that inhibited malignant processes[59]. CNP system optimized molecular targeted therapy and further integrated it with anti-inflammatory module, thereby providing superior therapeutic effects on CAC tumor.

Verifying in both CAC and CT26 mouse models, our study revealed that RG@M-γ-CD CNPs could also reverse the heterogeneous microenvironment of CRC by decreasing TAM infiltration. Although re-polarizing M2 into M1 or increasing M1 cells had therapeutic utility for some types of cancer, M1 cells played important roles in CRC-associated inflammation[19]. CAC is directly associated with chronic inflammation which initiates genomic instability and mutations in oncogenes and tumor suppressor genes, leading to tumorigenesis. Despite the pathogenesis of non-inflammatory CRC differs from that of CAC, clinical studies have indicated that non-inflammatory CRC tumors also display obvious inflammatory infiltration and increased pro-inflammatory cytokines within the localized microenvironment, which can promote accumulation of additional mutations and epigenetic changes during CRC development[49,60]. Therefore, intervention strategies for targeting the CRC-related inflammation have been being evaluated as adjuvant therapy for CRC treatment[61].

Colonic macrophages are one of the most abundant inflammatory cells in the colon, whose polarization contributes to homeostasis, inflammation, and CRC. During CRC progression, macrophages within TME can be activated into M1 and M2 TAMs. The activated TAMs represent the extremes of a continuum spectrum of different phenotypic features, which are determined by the variation of the physiological/pathological conditions[20]. For non-inflammatory CRC progression, M1 TAMs begin to accumulate within the adenoma. Following the malignant transformation that results in cancer, M2 cells become the predominant TAMs[62]. For CAC, M1 macrophages are highly expressed in the lamina propria adjacent to neoplasm, contributing to pro-inflammatory microenvironment. Within adenomas or tumors, M2 phenotype predominates to exert the protumorigenic role[63]. Despite that the M1 cells can antagonize the growth of established tumors, they contribute to CRC-related inflammation by producing cytokines including TNF-α, IL-6, and IL-1β. These cytokines can mediate the activity of transcription factors NF-κB and STAT3, which has important roles in dysplastic lesion, neoplasia, angiogenesis and extracellular matrix synthesis during CRC progression[16,64]. Thus, strategies that were directed to dampen the recruitment and activation of TAMs would find an important place within the therapeutic arsenal against CRC.

In line with this opinion, RG@M-γ-CD CNPs exert unique deactivation mechanisms on macrophages to decrease the infiltration and diversity of TAMs, by which the nanomedicine attenuates the production of pro-inflammatory cytokines and suppresses the polarization of protumorigenic M2 within the CRC microenvironment. In combination with its suppression on malignancy, RG@M-γ-CD nanomedicine achieves customized, synergetic therapeutic effects combating CRC. Given the complicated pathogenesis of CRC, we believe that prior to treatment, the genotypic subtype should be firstly characterized for patients to determine the valid targets[65]. Since RG@M-γ-CD nanomedicine inherits the broad-spectrum of kinase inhibition of RG, RG@M-γ-CD can be used as monotherapy for the CRC patients with specific molecular and genetic targets. Moreover, considering the aforesaid role of inflammation in CRC, we believe that the anti-inflammation function of RG@M-γ-CD has practical significance for the CRC patients with clinically detectable colorectal inflammation, which is consistent with the opinion that anti-inflammatory reagents can be used as adjuvant strategy for treatment of CRC.

In summary, we here develop a host-guest strategy that enables the integration of TME reprogramming and targeting therapy for CRC treatment. By exploiting the chemical motif and biological potential of CD, we endow RG@M-γ-CD CNPs with pragmatic channel structure for drug delivery and synergetic function against CRC. Together with facile fabrication and high biocompatibility, this host-guest assembling design represents a paradigm for nanomedicine innovation, and our findings indicate a promising prospect of RG@M-γ-CD CNPs in exploration for clinical applications.

## Methods

**Synthesis of M-γ-CD.** γ-CD (0.65 g, 0.50 mmol) and CDI (0.70 g, 4.30 mmol) were dissolved in DMSO (10.0 mL). The mixture was stirred at room temperature for 3 h and was then precipitated with cold diethyl ether. The obtained CDI-activated γ-CD was filtered, and stored at 4 °C for further use. Next, mannose (0.60 g, 3.30 mmol) was dissolved in 10.0 mL DMSO. The CDI-activated γ-CD in 5.0 mL DMSO and 0.3 mL triethylamine were added drop wise to the mannose solution over 2.5 h with continuous stirring, followed by reaction for 5 h. After reaction, the mixture was dialyzed in double distilled water for 24 h and freeze-dried to produce mannose-γ-CD (M-γ-CD).

**Preparation of RG@M-γ-CD NPs.** Co-crystallization method was adapted to prepare the RG@M-γ-CD inclusion complex in ethyl alcohol-water mix solution. RG (0.50 g, 1.0 mmol) and M-γ-CD (1.50 g, 1.0 mmol) were dissolved in 30.0 mL alcohol solvent (70%, v/v) in a round bottom flask. The mixture solution was stirred at 75 °C with reflux until the solids dissolving completely. The solution was kept for 75 °C for 12 h with continuous stirring to allow the host-guest complexation. After the formation of RG@M-γ-CD inclusion complexes, the solution was cooled slowly to room temperature, leaving 4–6 h for nano-suspension of RG@M-γ-CD. Afterwards, the solution was ultrasonically stirred at low speed for 2–4 h, the temperature and speed were carefully controlled to allow multilevel assembly of RG@M-γ-CD and avoid flocculation. RG@M-γ-CD CNPs was collected by filtration.

**Characterization of RG@M-γ-CD.** The NMR spectra including [1]H, [13]C, HSQC, HMBC, and NOESY were recorded on a Bruker Avance DMX-500 spectrometer (Billerica). M-γ-CD (15 mg), RG@γ-CD (18 mg), RG@M-γ-CD (20 mg), Rho@γ-CD (18 mg), and Rho@M-γ-CD (20 mg) were separately dissolved in 600 μL of DMSO-d6 and samples were examined in room temperature. The NMR spectra were analyzed with MestReNova 11.0. ESI-ToF-MS detection was conducted using AB Triple TOF 5600[plus] MS system (AB SCIEX, Framingham) with syringe pump (flow rate, 20 L/min) in positive ion mode. Source voltage was +5.0 kV and source temperature was normal atmospheric temperature. Pressure of Gas 1 (air) and Gas 2 (air) were set to 25 psi, and the pressure of Curtain Gas (N₂) was set to 25 psi. Declustering potential and collision energy (CE) was 60 and 10 V, respectively. Period Cycle Time was 275 ms and the scan range ranged from 200 $m/z$ to 3000 $m/z$. Exact mass calibration was performed automatically before each analysis employing the Automated Calibration Delivery System. Data acquisition was performed by Peak View software 1.2.

RG, RG@γ-CD, RG@M-γ-CD (equivalent RG, 10 mg), and M-γ-CD were separately solubilized in 25 mL methanol. Then 1 mL of the prepared solutions was diluted to 25 mL with distilled water. After stand for 1 h, the solutions were added into the quartz cell for scanning. The path length of the quartz cell was 1 cm. The absorbance spectra of the samples were recorded by TU-1901 dual-beam UV-vis spectrophotometer (Jingpu) between 200 and 400 nm. X-Ray diffraction patterns for RG (5 mg), unordered M-γ-CD (10 mg), none channel-type RG@M-γ-CD (15 mg), and RG@M-γ-CD CNPs (15 mg) were separately measured on Rigaku D/Max-2550PC diffractometer using a rotating-anode Cu-target X-ray (λ = 1.5406 Å) generator operated at 40 kV and 250 mA. Range of scanning was from 3.0° to 40.0° (2θ) with an increasing step size of 0.02° and count time of 0.5–2 s. RG (2 mg), M-γ-CD (5 mg), RG@M-γ-CD CNPs (8 mg) were separately placed in aluminum pan, the TG-DSC analysis was conducted on a TA DSC Q100 differential scanning calorimeter (TA Instruments). The heating was carried out at rate of 10 °C/min under a N₂ flow of 50 cm³/min, and the scanning was ranged from 30 to 380 °C.

RG@M-γ-CD-containing solutions were sampled at various time points and were diluted with water. Then the TEM samples (10 μL) were prepared by dropping diluted aqueous solutions onto 400-mesh carbon-coated copper grids with the excess solvent immediately evaporated. The morphologies of CNPs were observed using TEM 7066 system (Hitachi) operating at an accelerating voltage of

120 kV without any staining. A Veeco MultimodeV AFM (Veeco) was used to observe the 3-D morphology of CNPs. The aqueous solutions were sampled at various time points and were diluted with water. After dispersion by using ultrasonic, the solution (about 10 μL) was adsorbed onto a piece of freshly cleaved mica sheet and dried at 25 °C for 12 h before imaging. The imaging was conducted in a tapping mode with a scan rate of 0.5 Hz at room temperature. Data acquisition was performed by Veeco NanoScope 7.30. The DLS measurement was conducted on Zetasizer Nano ZS series (Malvern Instruments) at 25 °C. The measurements were carried out with scattering angle of 173°, and apparent hydrodynamic diameter ($D_h$) was presented as intensity. Data acquisition was performed by Zetasizer 7.0.

**High performance liquid chromatography (HPLC).** High performance liquid chromatography (HPLC) measurement of RG was performed on an HPLC system equipped with a 6200A Intelligent pump, auto-sampler, thermostat, and UV-Vis detector (Wanyi). Chromatographic separation was carried out on a Agilent ZORBAX-SB-C18 column (4.6 × 150 mm, 5 μm) maintained at 37 °C. The mobile phase consisted of acetonitrile and water (80:20, v/v) at flow rate of 0.5 mL min$^{-1}$ over a 15 min run time. The injection volume was 40 μL, and the UV detection was selected at wavelength of 265 nm.

**Drug release.** Individual 20 mg RG@M-γ-CD NPs were dispersed in PBS solutions (pH 7.0, pH 6.4, containing α-amylase or lipid) and then incubated in dark at 37 °C on a shaker plate at 250 rpm. At each time point, 200 μL of the release medium was replaced and diluted to a constant volume of 1 mL with acetonitrile. The released RG was measured by HPLC using a linear standard curve of RG (correlation coefficient > 99.9%). All measurements were carried out in triplicate.

**Cell cultures.** CT26, HT29, SW480, and RKO cells were purchased from American Type Culture Collection. HUVECs were purchased from Thermo Fisher Scientific. The quality of these cells was verified by the conventional tests of cell line quality control methods (e.g., morphology, isoenzymes, and mycoplasma). CT26 cells were maintained in RPMI-1640 medium with 10% fetal bovine serum (FBS) and 2% penicillin-streptomycin. HT29 cells were maintained in McCoy's 5a medium with 10% FBS and 2% penicillin-streptomycin. SW480 and RKO cells were maintained in Dulbecco's modified Eagle's medium with 10% FBS and 2% penicillin-streptomycin. HUVECs were maintained in vascular cell basal medium with 10% FBS and 2% penicillin-streptomycin. Cells were cultured and allowed to grow as a monolayer in 5% $CO_2$ and 37 °C, then collected with trypsin solution (0.5% w/v in PBS). Early-passage cells were used for in vitro assays.

**Cytotoxicity evaluation.** The cytotoxicities of free RG, M-γ-CD, Mix, RG@γ-CD, and RG@M-γ-CD against target cells were evaluated by MTT assay. Cells were seeded in 96-well plates with 100 μL medium (1 × 10$^4$ cells/well) and incubated for 16 h for attachment. Afterwards, the cells were treated with the formulations at various concentrations. At pre-set time points, the treatments were removed and the cells were washed with PBS. Then 100 μL MTT solution (0.5 mg/mL) was added into each well and the cells were incubated in MTT solution for 4 h. Next, the MTT solution was removed and 100 μL DMSO was added to solubilize formazan crystals. The absorbance of formazan solution at 570 nm was measured with a micro-plate reader (Thermo Fisher Scientific) to determine the cell viability.

**Endothelial cell tube formation.** In vitro HUVEC tube formation assay was conducted in a tumor cell environment by co-incubation and 3D Matrigel Matrix (Corning). The co-incubation was carried out in a transwell system (12-well plate). The CT26 cells were cultured on the upper insert, and the HUVEC were seeded in the plate with Matrigel Matrix (1%, w/v). Then the RPMI-1640 medium (FBS 10%) with free RG, M-γ-CD, Mix, RG@γ-CD, and RG@M-γ-CD were added, and the cells were co-incubated in 37 °C and 5% $CO_2$ for 6 h. The endothelial cell tube formation was observed by fluorescence microscope (Carl Zeiss) in bright field. In addition, HUVEC were seeded with Matrigel Matrix (0.5%, w/v) in a six-well plate, then were treated as above. The treated HUVECs were washed and cultured in serum medium containing Matrigel Matrix for 2 h. Then the cells were collected and treated with RIPA lysate (Beyotime) for phosphorylation examination.

**Western blot analysis.** Cells were washed and collected in PBS, lysed in RIPA buffer supplemented with 2% SDS, protease inhibitors and phosphatase inhibitors, followed by 5 min water sonication and 20,000 × g centrifugation for 20 min. The protein contents were analyzed using BCA protein assay kit (keygentec). Then equal amounts of protein were subjected to SDS-PAGE and transferred to PVDF membranes. Then the membranes were blocked in 5% (w/v) BSA in Tris-buffered saline with Tween-20, incubated overnight with primary antibody at 4 °C, followed by corresponding HRP-linked secondary antibody at 1:4000 dilution. Blots were exposed to autoradiographic films and scanned with Dorun scanner. The used primary antibodies and dilution rate were indicated in Supplementary Table 13.

**Quantification of inflammatory cytokines and M2-associated markers.** C57BL/6 mice were treated with thioglycollate broth via intraperitoneal injection. After

4 days post treatment, PMs were surgically isolated from the treated mice. PMs were seeded into six-well plates, then were treated with free RG, M-γ-CD, Mix, RG@γ-CD, and RG@M-γ-CD for 24 h. Subsequently, the treated PMs were stimulated with LPS (10 ng/mL), IL-4 (40 ng/mL), or supernatant of tumor cells. Then the cells and culture media were collected for quantitative RT-PCR and ELISA assay, respectively. For culture media, the inflammatory cytokines (IL-6 and TNF-α) and M2-associated markers (VEGF-B and MMP-9) were quantified by mouse ELISA kit (Neo bioscience) according to the manufacturer's instructions.

**RNA isolation and quantitative RT-PCR.** Total RNA of the collected PMs was isolated with RNAiso Plus (Takara) and single-strand cDNA was synthesized by reverse transcription kit (Tyobo). Quantitative real-time PCR was performed on CFX-Touch PCR system (Bio-Rad) using SYBR Green reagent (Roche). The mRNA level for each target gene was normalized to the corresponding GAPDH level, which was obtained by $2^{-\Delta\Delta CT}$ method. The primers were listed in Supplementary Table 12.

**Cell apoptosis detection.** Cell apoptosis was detected by annexin V-FITC/PI double staining assay kit (keygentec). The colon tumor cells cultured in 12-well plates were treated with free RG, M-γ-CD, Mix, RG@γ-CD, and RG@M-γ-CD at 2.0 μM of equivalent RG for 6 h. Then treatments were removed and the treated cells were incubated for 8 h. Afterwards, the cells were harvested and washed with PBS buffer, then re-suspended in 500 μL of binding buffer and stained with 5 μL of annexin V-FITC and 5 μL of PI for 15 min in dark. The fluorescence intensity of annexin V-FITC was measured by NovoCyte flow cytometer (ACEA).

**Bromodeoxyuridine incorporation.** The proliferation of the treated tumor cells was evaluated by BrdU incorporation assay. The colon tumor cells were seeded in 12-well plates and treated with free RG, M-γ-CD, Mix, RG@γ-CD, and RG@M-γ-CD at 2.0 μM of equivalent RG for 6 h. Then the treated cells were added with BrdU (Thermo Fisher Scientific) solution at a concentration of 10 μM for 24 h allowing the BrdU to incorporate into proliferating cells. Afterwards, the cells were fixed and incubated with anti-BrdU monoclonal detector antibody for 1 h, then the labeled cells were quantified by NovoCyte flow cytometer (ACEA).

**Animals.** Balb/c mice (male, 6–8 weeks) and C57BL/6 mice (male, 6–8 weeks) were purchased from the Zhejiang Academy of Medical Sciences institution (Hangzhou, China). All mice were maintained in the SPF care facility with sterilized food pellets and distilled water under a 12 h light/dark cycle. The temperature range for the housing room was 22–26 °C (average, ~24 °C) and the humidity range was 30–50% (average, ~50%). Animal experiment procedures were conducted in accordance with the protocols approved by the Zhejiang University Institutional Animal Care and Use Committee (ZJU20190040).

**In vivo CT26 tumor model.** Balb/c mice (~20 g body weight) were injected subcutaneously in the right groin region with 200 μL of cell suspension containing 2 × 10$^6$ CT26 cells. After 7–10 days post injection, the tumors reached a volume of ~100 mm$^3$. Then the tumor-bearing mice were randomly divided into six groups, intravenously injected with PBS, free RG (10 μg/g), M-γ-CD (20 μg/g), Mix (10 μg/g RG + 20 μg/g M-γ-CD), RG@γ-CD (27 μg/g), and RG@M-γ-CD (30 μg/g, equivalent RG of 10 μg/g), respectively. During treatment, tumor volume was measured every 2 days by using a caliper and calculated as [(tumor length) × (tumor width)$^2$/2]. Body weight change and survival rates were also recorded. After 21-day treatment followed by 14-day symptom observation, the mice were sacrificed and the tumors were excised for further analysis.

**In vivo CAC mouse model.** In vivo CAC mouse model was established on C57BL/6 mice according to the AOM and DSS method. Typically, the C57BL/6 mice (~20 g body weight) were intraperitoneally injected with AOM (10 μg/g, Sigma). After 7 days post injection, the mice were given with 2% (w/v) DSS in drinking water for 5 days, followed by treatment with ordinary water for 5 days. For the next 50 days, the model mice were fostered to allow the development of CAC. In the meantime, the body weight change, pathological phenomena and survival rate were recorded, the DAI was calculated to monitor the physical condition of the mice. At day 105 post-AOM injection, three mice were randomly selected and sacrificed. Their colons were excised and analyzed to confirm the establishment of tumor model. The model mice were randomly divided into six groups, orally administrated with PBS, free RG (10 μg/g), M-γ-CD (20 μg/g), Mix (10 μg/g RG + 20 μg/g M-γ-CD), RG@γ-CD (27 μg/g), and RG@M-γ-CD (30 μg/g, equivalent RG of 10 μg/g), respectively. During treatment, the changes of body weight, survival rate and DAI were recorded every 2 days. After 21-day treatment followed by 14-day symptom observation, the treated mice were sacrificed and the colons were surgically excised. After flushed with PBS, the CAC phenotypes of the colon tissues were analyzed, including colon length, tumor loading, and tumor volume. Then the colon tissues were stored in −80 °C for further analysis.

**Pharmacokinetics and biodistribution.** The study of pharmacokinetics and biodistribution was performed in tumor-bearing Balb/c mice by intravenous

injection. The tumor-bearing Balb/c mice received free RG, RG@γ-CD or RG@M-γ-CD at RG equivalent dose of 10 μg/g by tail vein injection. At the pre-determined time points, the mice were anesthetized and orbit collected 0.3 mL blood for the quantification of plasma drug concentration. The plasma was separated from the cell fraction by centrifugation. Then the supernatant was diluted with acidified acetonitrile to constant volume and centrifuged. RG concentration in the supernatant was quantified by HPLC. Pharmacokinetic parameters, $AUC_{(0-24)}$, $C_{max}$ and $T_{1/2}$, were calculated from plasma drug concentrations using a non-compartmental analysis. The tumors and main organs (heart, liver, lung, spleen, intestines and kidney) were excised and washed with PBS. Approximately 100 mg of tumors and organs were separated as samples for the assessment of biodistribution. The samples were weighed and cut into small pieces, added with acidified acetonitrile. After smashed using tissue grinder, the samples were homogenized and transferred to microtubes. The homogenates were then diluted with acidified acetonitrile to constant volume. The RG contents in tumors and organs were quantified by HPLC. All measurements were carried out in triplicate.

**In vivo imaging**. The MSOT Imaging was conducted with an In Vision 256-TF small animal scanner (iThera Medical). The tumor-bearing mice were anesthetized with isoflurane (1.0–2.0%, v/v) and then were transferred to the animal holder. Ultrasound gel was applied at tumor region for signal coupling and MSOT imaging were performed around tumor region to acquire transaxial sections. Optical parametric oscillator with seven arms of a fiber bundle was used to generate ring-shaped illumination with 8.0 mm width (excitation pulse, 9 ns of duration, 680–980 nm of wavelength). Ultrasound transducers (center frequency, 5.0 MHz) at a concave array was used for ultrasound signal detection. Data collection, reconstruction, analysis, and visualization were conducted with View MSOT 4.0 (iThera Medical). The raw signals were reconstructed with the largest cross-section in the middle of tumor region using a model-based inversion algorithm. Micro-positron emission tomography/computed tomography (micro-PET/CT) imaging was conducted with a U-SPECT-II/CT scanner (MILabs) using [18]F-FDG as indicator. Mice were kept under limosis at least 8 h, then injected intravenously with [18]F-FDG diluted in 100 μL saline solution (37 MBq/kg). After 10 min post injection, the mice were anesthetized with isoflurane to reduce [18]F-FDG uptake by skeletal muscles and adipose tissues. After 1 h post injection, the mice were placed in a holding device and PET/CT imaging was conducted using micro-PET/CT apparatus. PET scanning data were acquired for 15 min at normal count rate. CT scanning had the same localization as PET scanning for the correction of PET imaging. The raw data were reconstructed using a 3-D iterative algorithm, and the PET/CT images were analyzed with Xeleris 3.0.

**Ex vivo imaging**. C57BL/6 mice were orally given with Rho, Rho@γ-CD, or Rho@M-γ-CD at Rho equivalent dose of 10 μg/g. At different time point post administration, the mice from treatment groups were sacrificed and the main tissues (heart, stomach, lung, spleen, kidney, colon and liver) were surgically isolated. Then the tissues were washed with PBS and placed on a clean dish. Fluorescence intensities were measured by IVIS Lumina XRMS in vivo imaging system (PerkinElmer). Data acquisition was performed by Living Image 4.5.

**Histology and immunohistochemistry**. The tissues harvested from treatment groups were fixed in 4% paraformaldehyde neutral buffer and embedded in paraffin. Then the paraffin-embedded tissues were cut into slices (thickness, 6 μm) for H&E staining. For immunohistochemical studies, tumor tissues were harvested and fixed, then embedded in paraffin and prepared into serial sections. The sections were processed for immunohistochemical examination of tumor blood vessels (CD34 staining), apoptosis (TUNEL staining) and proliferation (Ki67 staining) according to the manufacturer's instructions. The antibodies used and dilution rate were indicated in Supplementary Table 13.

**Flow cytometry**. The colons were cut into pieces, followed by EDTA incubation. Then the incubation was digested by collagenase IV (300 U/mL, Worthington) in 5% FBS PBS solution at 37 °C for 1 h. After digestion, the cell suspension was filtered to get single-cell suspension containing lamina propria cells. Single-cell suspensions were then stained with antibodies and subsequently analyzed with NovoCyte flow cytometer (ACEA). The antibodies used and dilution rate were indicated in Supplementary Table 13. Data acquisition was performed by NovoExpress 1.4. Flow cytometry analysis was performed with FlowJo 10.0.

**Gene expression analysis**. Total mRNAs of colon tissues with tumors were quantified by Nano Drop ND-2000 (Thermo Fisher Scientific) and the mRNA integrity was verified using Agilent Bioanalyzer 2100 (Agilent Technologies). Sample labeling, microarray hybridization and washing procedures were carried out according to the manufacturer's protocols. The total mRNAs were transcribed into double strand cDNAs, then the cDNA was labeled with Cyanine-3-CTP. The labeled cRNAs were hybridized onto Agilent Sure Print G3 Mouse GE V2.0 microarray (8 × 60 K, Design ID:074809). After washing, the microarray was scanned with Agilent Scanner G2505C (Agilent Technologies). Data acquisition and array image analysis were performed using Feature Extraction software (version 10.7.1.1, Agilent Technologies) and Gene spring (version 13.1, Agilent Technologies) to quantify the mRNA expression levels. Differentially expressed mRNAs were then identified through fold changes and the corresponding statistical significances were calculated using unpaired two-tailed $t$-test. Fold change ≥ 2.0 with $p$ value ≤ 0.05 was used as threshold set to determine the up- and down-regulations. Afterwards, GO analysis and KEGG analysis were applied to analyze the roles of differential expression. Finally, Hierarchical Clustering was performed to display the distinguishable mRNA expression patterns.

**Statistical analysis**. Data are presented as mean values ± SD (standard deviation), calculated using GraphPad Prism 8 version 8.4.2. $p$ value < 0.05 was considered the threshold for statistical significance. $p$ value significance intervals (*) are provided within each figure legend. Unpaired two-tailed $t$-test statistical test is performed for the comparison of two groups and one-way (or two-way) ANOVA followed by multiple comparison test is performed for the comparison of multiple groups. N values are indicated within figure legends and refer to biological replicates. Derived statistics correspond to analysis of averaged values across biological replicates.

**Reporting summary**. Further information on research design is available in the Nature Research Reporting Summary linked to this article.

## Data availability
The mRNA microarray data generated in this study have been deposited in the NCBI database under accession code GSE157993. The source data underlying Figs. 2–7 and Supplementary Figs. 17, 18, 31–46 are provided as a Source Data file. Exact $p$ values are also included within the Source Data file. The remaining data are available within the Article, Supplementary Information or available from the authors upon request. Source data are provided with this paper.

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

## Acknowledgements

We gratefully acknowledge the National Natural Science Foundation of China (51573161, 51873185, and 81771699) for the financial support of this work. We thank Mr. Hua Wang from Bioultrastructure analysis lab of analysis Center, Agrobiology and environmental science, Zhejiang University, Dr. Qiaohong He, Dr. Linshen Chen, Ms. Fang Chen, Dr. Yaqin Liu, and Ms. Lin He from Analysis and test platform, Department of chemistry, Zhejiang University, for their technical assistance with characterization. We would also like to thank Ms. Ting Pan and Dr. Jia Xiong from School of medicine, Zhejiang University, and Ms. Xiaoqin Jin from Department of pathology academy of Chinese medical sciences, Zhejiang Chinese Medical University, for assistance with animal experiments.

## Author contributions

H.B., L.L., X.H., and G.T. conceived the project and designed experiments. H.B., J.W., and C.P. performed the nanomedicine formulation, characterization, and in vitro study. G.S., H.B., Q.C., and J.Z. performed the animal imaging and in vivo antitumor study. H.B. and L.L. analyzed data and wrote manuscript with the guidance and support of G.T.

## Competing interests

The authors declare no competing interests.
