## [Peer Review File · Nature Communications]

Reviewers' comments:

Reviewer #1 (Remarks to the Author): expertise in cancer immunology and mouse models of colorectal cancer

In this manuscript the authors describe a method to produce mannose-modified nanoparticles which carry the tyrosine kinase inhibitor Regorafenib (RG). Nanoparticles with RG showed better internalization in the two CRC cancer cell lines CT26 and HT29 than control nanoparticles, higher apoptosis rates and reduced proliferation rates.

In isolated peritoneal macrophages treatment with RG-containing nanoparticles lead to a reduced expression of pro-inflammatory cytokines.

In the AOM-DSS model mice treated with RG-containing nanoparticles showed the fewest tumor numbers and the longest colon length compared with controls.

In the CT26 CRC model mice treated with RG-containing nanoparticles showed dramatically reduced tumor volumes and an enhanced survival rate.

This is an interesting manuscript on a very important topic. Delivering therapeutics directly to tumors and metastasis will improve therapeutic options dramatically.

Although the study is informative and will be of great interest to researchers in the CRC field, I do suggest a few experiments and some controls.

Major points:

1. In Fig. 2a, 2b and 2c the authors show that Rhodamine-containing nanoparticles got incorporated into CT26 and HT29 cells although to a lesser extent than Rhodamine-containing M- γ -CD particles. In the functional assays with Regorafenib there are no RG@- γ -CD nanoparticles but a mixture, presumably RG plus M- γ -CD (I could not find any description of the mixture throughout the whole manuscript!!). It would be of some importance to show that RG-containing particles without mannose can not reduce eg. viability to the same extent as particles with mannose.

2. In the text the authors claim that within 12h of incubation tube formation by HUVEC cells was significantly blocked by RG@- γ -CD but I could not find the 12h images in Fig.2g. The authors should complement the figure or correct the text.

As endothelial cells express mannose receptors it would be of some interest to show that RG-containing nanoparticles without mannose do not block tube formation.

With the exception of Fig.2g, throughout the whole Figure 2 there are no time points stated. And I could not find any in the figure legends. This makes the figure very difficult to read and understand.

3. In Fig. 3 the authors show the effects of nanoparticles on isolated peritoneal macrophages.

In Fig.3a it would be interesting to show that macrophages possess mannose receptors. In Fig.3f I do not understand why the authors measured CD206 in IL-4-treated macrophages. Is the mannose receptor CD206 in peritoneal macrophages only expressed after IL-4 stimulation? Is the mannose receptor on M1 macrophages missing?

The qPCR analysis are a good hint towards anti-inflammatory function of RG@M- γ -CD but the authors should show some protein data like FACS analysis, ELISA or immunofluorescence to validate the PCR data.

4. In Fig. 4 the authors switch to the AOM/DSS mouse model of CRC. In the in vitro experiments the authors used Rho@- γ -CD and Rho@M- γ -CD nanoparticle but in Fig. 4a only free Rho and no Rho@- γ -CD. Why is only free Rhodamine used?

In Fig. 4d and Fig.4e it seems that mice treated with Mix (whatever this may be) were doing best of all mice. They had the highest body weight and the lowest disease activity index. And in Fig. 4f, although the Mix in the in vitro assays is not able to reduce inflammatory cytokines efficiently or to polarize macrophages towards an anti-tumoral phenotype, mice treated with Mix had the same colon length as mice treated with RG@M- γ -CD.

At the same time mice fed with Mix had the same tumor numbers and the same tumor volumes compared with mice fed with RG. Obviously the Mix dampens inflammation in this tumor model. But reduced inflammation is not leading to reduced tumorigenesis. This is a discrepancy the authors should explain and discuss.

5. In Fig. 5 the authors performed gene expression analysis in tumors from RG@M- γ -CD-treated

mice and PBS-treated mice as controls. What is the expression status of the microenvironment? What is the expression status of treated tumor cells? The authors should isolate the cells, sort them and then perform gene expression analysis on the different cell types of the TEM and isolated carcinoma cells.

In Fig. 5d the images are not very good. Especially the quality of the Ki67 and the CD31 stainings are very poor and need to be improved. And a small image in high magnification for every image would help the reader to see the differences between the groups more clearly.

In Fig. 5f the FACS analysis of the TAMs is not very convincing. Treating mice with RG@M- γ -CD nanoparticles lead to a decrease of the mannose receptor in TAMs compared with control macrophages. In the same group CD80-positive activated macrophages decreased from 3.92% (control) to 3.24%. In the text the authors claim that treatment with RG@M- γ -CD nanoparticles lead to a potent deactivation on TAMs. This is not evidenced by the data. In Fig. 5i the MMP9 expression of all groups are more or less the same and I doubt that there are differences in the mRNA levels of PDGF-A between the different groups.

In Fig. 5j the quality of the images need to be improved. Although the authors say that "the positive areas in extracellular depots were dramatically decreased compared with other groups" I could not see any difference between the Mix group and the RG@M- γ -CD.

Although the authors claim that the mechanism by which RG@M- γ -CD targets malignant processes are the macrophages in the TME, the results they present are largely phenomenological. To really show the mechanism via macrophages the authors should perform therapeutic experiments with mice lacking macrophages e.g. knockout mice.

6. In Figure 6 the authors switch to a transplantation model. This means that the therapeutic agent RG is no longer fed but applied systemically. In Fig. 6d it looks as if the PBS-treated control mouse shows the least uptake of RGD of all five groups. Are the authors sure that CT26 colon carcinoma cells absorb the PET tracer? Please provide better PET images or remove the images from the manuscript.

Please use higher magnification or better images in Fig. 6h. The stainings are really hard to see. The authors claim that "the potent TAM deactivation of RG@M- γ -CD CNP has significant implication". This conclusion is not supported by the data the authors present.

To really show that macrophages are the main target of the treatment the authors should perform in vivo treatment in mice deficient of macrophages.

7. Throughout the whole manuscript the authors claim that RG@M- γ -CD reprograms the TME but the authors never show the composition of the TME in neither of the two tumor models they use. It would be of great interest to show the composition of the TME in both models.

Minor points

I know there is a limit of characters for this manuscript, but I think the discussion is too short. Half of the discussion deals with the physical properties of the nanoparticles and only very few words are said to the functional results of the in vivo experiments.

The authors use a Mix they never specify (or at least I could not find a word about the composition of the Mix).

In the figures and in the figure legends there is no word about the time points used. This makes the figures hard to read.

I know there is a limit of characters for the figure legends but the legends used are too short. They do not help to understand the figures.

Reviewer #2 (Remarks to the Author): expertise in cyclodextrine-based nanomaterials and drug delivery

The manuscript reports the synthesis of a new gamma-CD monofunctionalized with mannose as a carrier of regorafenib. The manuscript is in the field of cyclodextrin inclusion complexes and reports several experiments in vitro and in vivo.

The main problem of the manuscript is the very lacking characterization of the cyclodextrin

derivative M- γ -CD and some doubt on the identity of the proposed nanosystem. Chemical-physical characterization and full analysis of data are the first essential step in this kind of study. I think that RG@M- γ -CD in solution is not a nanosystem but is the inclusion complexes of RG and M- γ -CD. Regardless, this aspect may not reduce interest in the results. The title should be changed.

Main points

Synthesis: The authors should demonstrate the identity of the new product, C-4 functionalization of Man and C-6 functionalization of CD in the derivative. The authors should add references to the synthetic approach.

Characterization in solution M- γ -CD

A full characterization with NMR of the new derivative M- γ -CD is needed. The few NMR spectra (^1H NMR and ^{13}C NMR) reported were not assigned. Man and CD signals overlap but 2D spectra (COSY; TOCSY, HSQC, HMBC etc) can identify all the protons and the carbons. In fig 1sa small signals between 3.5-2.0 ppm are also evident. What are they? Spectra should be fully assigned, integrated and used to support the proposed structure of the derivative. Furthermore, Mannose forms two anomers and this should be evident in the spectra.

In the SI, NMR spectra in D₂O are reported while in the manuscript the authors wrote "d6 DMSO". The spectra in SI are not well readable.

A careful analysis of MS spectra is needed. The reported analysis of MS spectra is very approximate. The MS analysis show a main peak at 1511 m/z assigned to M- γ -CD+K (line129). The expected isotopic molecular mass of the molecule shown in Figure 1 is 1502 and so we can expect in MS spectrum peaks at 1502+H, 1502+Na, 1502+K m/z (not 1511). Furthermore, the peak at 758.22 cannot be the double charge of M- γ -CD (line 170).

M(2)- γ -CD cited and used in SI was not described and characterized in the manuscript A full characterization should be added.

RG@M- γ -CD

Line 135 Authors: "The co-existence of peaks related to the protons of M- γ -CD and RG in ^1H NMR spectrum of RG@M- γ -CD revealed the formation of interlocked molecule"

The coexistence of peaks is not evidence of inclusion or interaction, it is the natural sum of the proton resonances of the two molecules. Other experiments should be carried out to investigate the interaction. Spectrum should be fully assigned and integrated. In the spectra of fig 3s (pictures have low quality and are unreadable), there are peaks in the aliphatic region that are not present in the spectra in fig 1sa and are not due to RG protons. What are they? Is the spectrum in D₂O? Authors wrote (Line 136) "In 2D NOESY NMR spectroscopy, signals of nuclear overhauser effect (NOE) correlation were detected between the glycogen of M- γ -CD and the amide of RG, implying a partial entrapment of RG in the cavity of M- γ -CD (Fig. 1d and s2b)." This comment is very generic.

Amide protons of RG are exchangeable protons and cannot show any signal in D₂O. Typically the protons of the included compound show NOE correlation with the H-3, H-5 protons of CD. Noesy spectra (Fig 1d) should be appropriately discussed.

The characterization in the solid-state suggests the formation of ordered structure self-assembled. This behavior has been found in the case of CDs. The ordered structure may not exist in the solution.

In fact, NMR spectra do not show broad signals typically found in the case of nanosystems.

The authors compared RG@M- γ -CD to a mixture of RG and M- γ -CD. A "mixture" of these molecules in water forms the inclusion complex (see calorimetric titrations in SI), with the stability constant reported in the manuscript. On this basis, the differences found between RG@M- γ -CD and mix are not clear.

The differences in the UV-Vis spectra are not clear: UV-Vis spectrum of RG@M- γ -CD shows a shift in the band at 250 nm and this may be due to the inclusion but what is the new band at 310 nm? Please analyze the spectrum.

For all the spectra and other chemical-physical characterizations, the concentration, the solvent used etc should be reported.

In the manuscript there are some typos (Line 12 TEM, line 67 TMA, SI 2D NOesy etc)

For these reasons, I cannot recommend the publication of the manuscript in Nature Communications.

Reviewers' comments

Reviewer #1 (Remarks to the Author): expertise in cancer immunology and mouse models of colorectal cancer

In this manuscript the authors describe a method to produce mannose-modified nanoparticles which carry the tyrosine kinase inhibitor Regorafenib (RG). Nanoparticles with RG showed better internalization in the two CRC cancer cell lines CT26 and HT29 than control nanoparticles, higher apoptosis rates and reduced proliferation rates. In isolated peritoneal macrophages treatment with RG-containing nanoparticles lead to a reduced expression of pro-inflammatory cytokines. In the AOM-DSS model mice treated with RG-containing nanoparticles showed the fewest tumor numbers and the longest colon length compared with controls. In the CT26 CRC model mice treated with RG-containing nanoparticles showed dramatically reduced tumor volumes and an enhanced survival rate. This is an interesting manuscript on a very important topic. Delivering therapeutics directly to tumors and metastasis will improve therapeutic options dramatically. Although the study is informative and will be of great interest to researchers in the CRC field, I do suggest a few experiments and some controls.

Response:

We thank the reviewer for the positive comments and kind suggestions to our manuscript. In this manuscript, we have reported a CD-derived, MR-targeting TKI nano-medicine (RG@M- γ -CD CNPs) and its potentials in CRC treatment. Before the construction of RG@M- γ -CD CNPs, we have prepared a series of TKI@CD nano-medicines using host-guest chemistry, including RG@ γ -CD. In comparison with RG@M- γ -CD, RG@ γ -CD without mannose-modification have shown a suboptimal outcome combating CRC. Therefore, the RG@ γ -CD-related results have not been discussed in the manuscript. After careful consideration of your suggestion, we have supplemented the relevant results and discussion to emphasize the novelty and applicability of mannose-functionalized CNPs. Besides, we have added a series of experiments and carefully revised the manuscript based on your suggestions and respond to your comments point-by-point.

Major points:

1. In Fig. 2a, 2b and 2c the authors show that Rhodamine-containing nanoparticles got incorporated into CT26 and HT29 cells although to a lesser extent than Rhodamine-containing M- γ -CD particles. In the functional assays with Regorafenib there are no RG@ γ -CD nanoparticles but a mixture, presumably RG plus M- γ -CD (I could not find any description of the mixture throughout the whole manuscript!!). It would be of some importance to show that RG-containing particles without mannose can not reduce eg. viability to the same extent as particles with mannose.

Response:

This is a constructive suggestion. The functional assays with RG-containing particles without mannose (denoted as RG@ γ -CD) have been performed and the relevant results have been supplemented to evidence the meliority of mannose modification. The updated **Fig. 2e, 2f** has indicated that RG@ γ -CD nano-formulation has amplified RG's inhibitory effects as compared to free drug, resulting in enhanced apoptosis induction and proliferation inhibition. In comparison with RG@M- γ -CD, RG@ γ -CD has shown suboptimal suppression towards CRC cells, confirming the rational design of M- γ -CD-based CNP formulation. The composition of mixture has been added in the manuscript for better understanding.

"The CRC cells were treated with RG, M- γ -CD, mixture of RG and M- γ -CD (denoted as Mix), RG@ γ -CD and RG@M- γ -CD at equivalent concentrations of RG."

2. In the text the authors claim that within 12 h of incubation tube formation by HUVEC cells was significantly blocked by RG@M- γ -CD but I could not find the 12 h images in Fig. 2g. The authors should complement the figure or correct the text. As endothelial cells express mannose receptors it would be of some interest to show that RG-containing nanoparticles without mannose do not block tube formation. With the exception of Fig. 2g, throughout the whole Figure 2 there are no time points stated. And I could not find any in the figure legends. This makes the figure very difficult to read and understand.

Response:

We thank the reviewer for pointing out the mistake in the description of tube formation assay. The incubation in tube formation assay has been lasted for 6 h and we have corrected the description. Tube formation assay using RG@ γ -CD has been supplemented according to the comment. Within 6 h of incubation, RG@ γ -CD has also hindered tube formation and induced a slightly weaker inhibition comparing with

RG@M- γ -CD, probably owing to the negative expression of MR on HUVECs [1]. To confirm the inhibitory effect, phosphorylation of VEGFR and PDGFR has been evaluated using RG@ γ -CD CNPs. As shown in **Fig. 2j**, both CNPs (RG@ γ -CD and RG@M- γ -CD) have exhibited stronger inhibition on phosphorylation of growth factor receptors than free drug, suggesting that CD-derived nano-formulations facilitate the functionalization of RG in HUVECs. According to the reviewer's comment, we have carefully supplemented the description of time points for the experimental results presented in **Fig. 2**. The figure legend has been modified and expanded to convey more experimental information.

[1] Groger M, Holnthoner W, Maurer D, et al., Dermal microvascular endothelial cells express the 180-kDa macrophage mannose receptor in situ and in vitro. **Journal of Immunology**, 2000, 165(10): 5428-5434.

Figure 2. In vitro anti-tumor effect of RG@M- γ -CD CNPs. Cell internalization of CNPs. (a and b) Representative CLSM images of the CT26 and HT29 cells cultured with Rho@M- γ -CD and Rho@ γ -CD CNPs (equivalent Rho, 5.0 μ M) for 2 h. Blue fluorescence indicated nuclear staining with DAPI; green fluorescence indicated β -actin staining with FITC-antibody; red fluorescence presented the internalization of CNPs. Scale bar represented 20 μ m. (c) FACS analysis of fluorescence intensities in CT26 and HT29 cells (2×10^4 cells/sample, n=3) induced by internalized Rho@M- γ -CD and Rho@ γ -CD (equivalent Rho, 5.0 μ M and 10.0 μ M). (d) In vitro cytotoxicity of various formulations (equivalent RG, 0.5-10.0 μ M) towards CT26 and HT29 cells with 12 h of treatment. Proliferation inhibition and apoptosis induction on CT26 and HT29

cells treated with RG@M- γ -CD. (e) Representative proliferation rate for treatment groups. Cells were treated with various formulations (equivalent RG, 2.0 μ M) for 6 h and (f) proliferation rates were examined by BrdU assay (n=3). (g) Representative apoptosis rates for treatment groups. Cells were exposed with various formulations (equivalent RG, 2.0 μ M) for 6 h and (h) apoptosis rates were evaluated by Annexin V-PI assay (n=3). (i) Representative images of in vitro tube formation by HUVECs treated with different formulations (equivalent RG, 2.0 μ M) for 6 h. The numbers of micro-tube in wells were counted (n=3). (j) Western blot analysis of expression and phosphorylation of VEGFR-2 and PDGFR- β in HUVECs under different treatments (equivalent RG, 2.0 μ M, 6 h). The data were expressed as means \pm s.e.m.*P < 0.05, **P < 0.01.

3. In Fig. 3 the authors show the effects of nanoparticles on isolated peritoneal macrophages. In Fig. 3a it would be interesting to show that macrophages possess mannose receptors. In Fig.3f I do not understand why the authors measured CD206 in IL-4-treated macrophages. Is the mannose receptor CD206 in peritoneal macrophages only expressed after IL-4 stimulation? Is the mannose receptor on M1 macrophages missing? The qPCR analysis are a good hint towards anti-inflammatory function of RG@M- γ -CD but the authors should show some protein data like FACS analysis, ELISA or immunofluorescence to validate the PCR data.

Response:

As the reviewer suggested, the cell internalization using peritoneal macrophages has been re-performed and the mannose receptor on the macrophages has been stained with fluorescent antibody (primary antibody, abcam, ab64693; secondary antibody, abcam, ab7086). From the updated **Fig. 3a**, it can be indicated that Rho@M- γ -CD CNPs have exerted a more efficient internalization comparing with Rho@ γ -CD CNPs owing to the specific binding between mannose group and mannose receptor.

As the reviewer's comment, peritoneal macrophages constitutively express mannose receptor on their surfaces. With the treatment of IL-4, the expression and activity of CD206 on macrophages has been proven to be potently enhanced and the treated macrophages have been polarized into M2 phenotype [1-3], which makes CD206 become an M2 macrophage marker in both the mouse and the human [4]. The colonic macrophages have shown similar polarization profile. Once stimulated with IL-4 or educated by CRC cells, they have been activated into M2 phenotype and their CD206 expression has also been up-regulated [5, 6]. Therefore, we have used CD206 as a marker to identify the M2 activation in this manuscript.

In order to validate the anti-inflammation effect of RG@M- γ -CD in protein level, the inflammatory cytokines secreted by macrophages have been quantified using ELISA assay. Peritoneal macrophages are seeded into 6-well plates (1×10^6 cell/well) and pre-treated with LPS (10 ng/mL). Activated macrophages are then treated with PBS, RG, M- γ -CD, mix, RG@ γ -CD and RG@M- γ -CD (equivalent RG, 2.0 μ M) for 6 h. Then the supernatants are collected and the inflammatory cytokines (IL-1 β , IL-6, TNF- α and TGF- β 1) are determined by commercial ELISA kits (Neobioscience) according to the manufacturer's instructions. Protein expression levels measured by ELISA have shown similar profiles as compared to the PCR data, confirming that RG@M- γ -CD treatment has reduced the pro-inflammatory cytokines. Moreover, the

reduction of TGF- β 1 in protein level has found to be more evident than that in mRNA level. Besides, the cytokine expression of VEGF-B, PDGF-A and MMP9 by polarized macrophages has also been examined using ELISA and the result has shown similar tendency with the qPCR data. Protein levels of the secreted cytokines have been expressed as pg/mL and presented as average \pm SD (**Fig. s31, s32**). These results have further strengthened the conclusion that RG@M- γ -CD effectively exerts regulatory effects towards macrophages.

- [1] Rivollier A, He J, Kole A, Valatas V, Kelsall BL, Inflammation switches the differentiation program of Ly6chi monocytes from antiinflammatory macrophages to inflammatory dendritic cells in the colon. *The Journal of Experimental Medicine*, 2012, 209(1): 139-155.
- [2] Stein M, Keshav S, Harris N, Gordon S, Interleukin 4 potently enhances murine macrophage mannose receptor activity: a marker of alternative immunologic macrophage activation. *The Journal of Experimental Medicine*, 1992, 176(1): 287-292.
- [3] Wang J, Xu LZ, Xiang Z, Ren Y, Zheng XF, Zhao QY, Zhou QZ, Zhou YF, Xu L, Wang YP, Microcystin-LR ameliorates pulmonary fibrosis via modulating CD206(+) M2-like macrophage polarization. *Cell Death & Disease*, 2020, 11(2): 136.
- [4] Murray PJ, Allen JE, Biswas SK, et al., Macrophage activation and polarization: nomenclature and experimental guidelines. *Immunity*, 2014, 41 (2): 339-340.
- [5] Isidro RA, Appleyard CB, Colonic macrophage polarization in homeostasis, inflammation, and cancer. *American Journal of Physiology-Gastrointestinal and Liver Physiology*, 2016, 311(1): 59-73.
- [6] Bain CC, Mowat AM, Intestinal macrophages specialised adaptation to a unique environment. *European Journal of Immunology*, 2011, 41(9): 2470-2525.

Figure 3. In vitro regulatory effect of RG@M- γ -CD upon macrophage. Cell internalization of CNPs by macrophages. (a) Representative CLSM images of the peritoneal macrophages cultured with Rho@M- γ -CD and Rho@ γ -CD CNPs (equivalent Rho, 5.0 μ M) for 2 h. Blue fluorescence indicated nuclear staining with DAPI; green fluorescence indicated mannose receptors staining with FITC-antibody; red fluorescence presented the internalization of CNPs. Scale bar presented 20 μ m. (b) FACS analysis of fluorescence intensities in peritoneal macrophages (2×10^4 cells/sample, $n=3$) induced by internalized Rho@M- γ -CD and Rho@ γ -CD (equivalent Rho, 5.0 μ M and 10.0 μ M). (c) Schematic illustration of peritoneal macrophages isolation, stimulation, and evaluation of anti-inflammation or M2 polarization. (d) mRNA expression levels of pro-inflammatory cytokines by peritoneal macrophages treated with various formulations (equivalent RG, 2.0 μ M, $n=3$) for 4 h. (e) Western blot analysis of p-p65, p65 and abca1 expressions in macrophages under different treatments (equivalent RG, 2.0 μ M) for 4 h. (f) mRNA expression levels of M2-associated markers in macrophages under different treatments (equivalent RG, 2.0 μ M, $n=3$) for 4 h. (g) mRNA expression levels of VEGF-B, PDGF-A and MMP-9 in macrophages under different treatments (equivalent RG, 2.0 μ M, $n=3$) for 4 h. Data

were expressed as means \pm s.e.m, *P < 0.05, **P < 0.01.

4. In Fig. 4 the authors switch to the AOM/DSS mouse model of CRC. In the in vitro experiments the authors used Rho@ γ -CD and Rho@M- γ -CD nanoparticle but in Fig. 4a only free Rho and no Rho@ γ -CD. Why is only free Rhodamine used? In Fig. 4d and Fig.4e it seems that mice treated with Mix (whatever this may be) were doing best of all mice. They had the highest body weight and the lowest disease activity index. And in Fig. 4f, although the Mix in the in vitro assays is not able to reduce inflammatory cytokines efficiently or to polarize macrophages towards an anti-tumoral phenotype, mice treated with Mix had the same colon length as mice treated with RG@M- γ -CD. At the same time mice fed with Mix had the same tumor numbers and the same tumor volumes compared with mice fed with RG. Obviously the Mix dampens inflammation in this tumor model. But reduced inflammation is not leading to reduced tumorigenesis. This is a discrepancy the authors should explain and discuss.

Response:

When investigating the pharmacokinetic properties via oral administration, we have monitored the plasma concentration and tested the bio-distribution for free Rho, Rho@ γ -CD and Rho@M- γ -CD in C57 mice. The results of free Rho and Rho@M- γ -CD have been presented in **Fig. 4a** for comparison to highlight the M- γ -CD-based CNP formulation improving the pharmacokinetics of free drug. According to your suggestion, we have supplemented the result of Rho@ γ -CD as another control which has been shown in the updated **Fig. 4a**. Meanwhile, relevant description and discussion have been added.

Fig. 4d-4j has presented the therapeutic effect of different treatments towards CAC model. The systematic outcomes has been evaluated in combination with survival profile. Therefore, we have supplemented the survival curve in **Fig. s34**. It can be found that the mice treated with RG@M- γ -CD displays the longest survival time comparing with other treatment groups. Although the Mix group (physical mixture of equivalent RG and M- γ -CD) has higher average of body weight at day 22, it shows a shorter median survival time as compared to RG@M- γ -CD group. **Fig. 4e** presents the colitis pathology for the treatment groups, which indicates that treatment with Mix lowers the DAI of colitis. Coupled with the statistics of colon length (**Fig. 4f**), it can be speculated that the Mix treatment mitigates the colonic inflammation induced by DSS, which is further confirmed by the result of reduced pro-inflammatory cytokines in **Fig. 5e**.

Despite alleviating the colitis, Mix treatment shows faintish suppression on colonic tumorigenesis. From **Fig. 4g-4j**, the statistics shows that, although Mix

treatment reduces the tumor number to a certain extent, the highest reduction of tumor number is obtained by RG@M- γ -CD treatment. Besides, RG@M- γ -CD treatment also significantly inhibits the tumor growth, while Mix group shows similar tumor volume with RG group. Together, the results indicate that RG@M- γ -CD nano-medicine exerts an efficient therapeutic function towards CAC by synergistically mitigating inflammation and suppressing tumorigenesis. In contrast, although the mixture of RG and M- γ -CD alleviates the colitis (which is attributed to the anti-inflammation effect of cyclodextrin), it fails to improve the anti-tumor effect of RG. As the reviewer's comment, reduced inflammation is not leading to reduced tumorigenesis. According to the suggestion, the relevant explanation has been supplemented in **Discussion** Section.

Figure 4. Therapeutic efficacy of RG@M- γ -CD in CAC model. (a) Representative ex vivo fluorescence imaging and mean fluorescence intensity of intestinal tissues for C57 mice orally given with Rho, Rho@ γ -CD and Rho@M- γ -CD CNPs at different time points post administration (equivalent Rho, 10 μ g/g, n=3). (b) Representative ex vivo fluorescence imaging of main organs at 2 h, 6 h, 12 h and 24 h post administration.

administration with Rho@M- γ -CD CNPs. (c) Schematic illustration of the CAC mouse model establishment based on AOM-DSS method. Changes of (d) body weight and (e) disease activity index (DAI) for different treatment groups during the in vivo anti-CAC experiment (equivalent RG, 10 μ g/g, n=10). (f) Representative images of colons, and colon lengths from different treatment groups (n=5). (g) Representative colons with tumor loading from different treatment groups. (h) Colon tumor numbers, (i) tumor volumes, (j) and average tumor sizes from the treatment groups (n=5). Data were expressed as means \pm s.e.m, *P < 0.05, **P < 0.01, ***P < 0.001.

5. In Fig. 5 the authors performed gene expression analysis in tumors from RG@M- γ -CD-treated mice and PBS-treated mice as controls. What is the expression status of the microenvironment? What is the expression status of treated tumor cells? The authors should isolate the cells, sort them and then perform gene expression analysis on the different cell types of the TEM and isolated carcinoma cells.

Response:

We thank for the reviewer's important suggestions. In our experimental design of gene expression analysis, we have referred to some published reports [1-3] and the experiment has been conducted using fresh tumor tissues without sorting. The data would be more convincing if the gene expression analysis are performed on different cell types. To show the regulatory effects of RG@M- γ -CD on tumor cells and microenvironment, we have re-analyzed the differential genes by Gene Ontology enrichment analysis (GOEA). In the updated **Fig. 5c**, we have classified the GO annotation and modified the description according to the correlation between differential genes and TME mediation or tumor inhibition ($-\log_{10}(p \text{ value})$, 27-15). Coupled with the overall GOEA in **Fig. 5b**, the regulatory profile can be demonstrated for RG@M- γ -CD against CAC.

[1] Yu GC, Yu S, Saha ML, Zhou J, et al., A discrete organoplatinum(II) metallacage as a multimodality theranostic platform for cancer photochemotherapy. *Nature Communications*, 2018, 9: 4335.

[2] Zheng DW, Chen Y, Li ZH, Xu L, et al., Optically-controlled bacterial metabolite for cancer therapy. *Nature Communications*, 2018, 9: 1680.

[3] Zimmer S, Grebe A, Bakke SS, et al., Cyclodextrin promotes atherosclerosis regression via macrophage reprogramming. *Science Translational Medicine*, 2016, 8(333): 33ra50.

In Fig. 5d the images are not very good. Especially the quality of the Ki67 and the CD31 stainings are very poor and need to be improved. And a small image in high magnification for every image would help the reader to see the differences between the groups more clearly. In Fig. 5f the FACS analysis of the TAMs is not very convincing. Treating mice with RG@M- γ -CD nanoparticles lead to a decrease of the mannose receptor in TAMs compared with control macrophages. In the same group CD80-positive activated macrophages decreased from 3.92% (control) to 3.24%. In the text the authors claim that treatment with RG@M- γ -CD nanoparticles lead to a potent deactivation on TAMs. This is not evidenced by the data.

Response:

According to the comment, the immunohistochemical staining assay has been

re-performed to improve the quality of Ki67 and the CD31 staining. The paraffin-embedded colon tumor sections from the treatment groups have been stained with antibodies against Ki67 and CD31 to indicate the proliferation of tumor cells and formation of micro-vessels. The images in high resolution have been captured by Nanozoomer 2.0RS system. Small images in high magnification for every image have been shown in **Fig. s40**. In order to demonstrate the RG@M- γ -CD treatment deactivating the TAMs, FACS analysis has been re-performed for CD11b⁺ CD80⁺ subset. The data have indicated that CD80⁺ TAMs of RG@M- γ -CD group moderately decreased from 3.92 \pm 1.42% (control) to 3.24 \pm 0.75%. In the text, we have corrected the description as below "In comparison with control, RG@M- γ -CD treatment decreased both M1 and M2 TAM infiltration (**Fig. 5f-5h**). Particularly, M2 cells (CD11b⁺ CD206⁺ TAMs) were reduced by 66.2 \pm 5.3%. Treatment with RG, mix or RG@ γ -CD also significantly decreased M2 population, implying that the deactivation was a result of multi-kinase inhibition."

In Fig. 5i the MMP9 expression of all groups are more or less the same and I doubt that there are differences in the mRNA levels of PDGF-A between the different groups.
Response:

The significance in **Fig. 5i** have indicated the differences of mRNA levels between the treatment groups and control group. During the revision, the treatment with RG@ γ -CD has been conducted and the mRNA quantification has been re-performed. In comparison with the untreated control, treatment with RG, Mix, RG@ γ -CD and RG@M- γ -CD significantly decreases the mRNA levels of VEGF-B, PDGF-A and MMP9, particularly RG@M- γ -CD treatment displays the strongest suppression on the production of M2-related tumor-supportive factors. The statistical significance of all treatment groups is calculated and shown in **Fig. 5i** to indicate the differences of expression level.

In Fig. 5j the quality of the images need to be improved. Although the authors say that "the positive areas in extracellular depots were dramatically decreased compared with other groups" I could not see any difference between the Mix group and the RG@M- γ -CD.

Response:

According to your comment, we have re-performed the immunohistochemical staining assay and the sections have been scanned with Nanozoomer 2.0RS for higher quality. Representative images have been updated in **Fig. 5j**. It can be seen that

treatment with RG@M- γ -CD CNPs results in the significant reduction of VEGF-positive area as compared with other groups.

Although the authors claim that the mechanism by which RG@M- γ -CD targets malignant processes are the macrophages in the TME, the results they present are largely phenomenological. To really show the mechanism via macrophages the authors should perform therapeutic experiments with mice lacking macrophages e.g. knockout mice.

Response:

This study is interdisciplinary and is based on the prior achievements by researchers of oncology, medicine, cancer therapy and drug delivery. The therapeutic targets and mechanism we used have been confirmed and validated in the clinical and experimental studies. TAMs within TME have been proven to exert various tumor-promoting functions (including pro-angiogenic, pro-invasive, pro-metastatic, mitogenic, apoptosis-resistant and immune-suppressive) and have been regarded as an important therapeutic target for anti-cancer therapy [1-3]. For CRC, macrophage infiltration in tumor tissues is markedly increased and plays pivotal roles in CRC-related inflammation and tumor-supportive TME by producing various cytokines, chemokine and growth factors. Besides, increased TAM density is usually associated with metastasis and poor prognosis [4-6]. To show the therapeutic outcome by targeting TAMs, knockout or gene-deficient models have been established and used for the investigation of CRC treatment. Naofumi et al. used TNF-Rp55-deficient mice to prevent macrophages accumulating in colon and identified that reducing macrophage infiltration markedly suppressed the progression of colitis-associated colon cancer [7]. Liu et al. used Crhr1-deficient mice to explore the role of macrophages in the development of AOM-DSS CAC. They concluded that Crhr1-deficiency resulted in decreased macrophage infiltration. Consequently, the production of IL-1 β , IL-6 and TNF- α was attenuated, which reduced tumorigenesis during early CAC [8]. Aiming at the established therapy target of TAMs, we herein design the new host-guest nano-medicine and investigate its potentials in CRC treatment. Particularly, Mannose-modified cyclodextrin plays a dual role that not only serving as a CRC-targeting nanocarrier for TKI, but also exerting a regulatory effect on inflammation. In our study, we have designed substantial *in vitro* and *in vivo* experiments to confirm the therapeutic mechanism of RG@M- γ -CD and validate the implement of the new nano-medicine in CRC therapy.

- [1] Hanahan D, Weinberg RA, Hallmarks of cancer: the next generation. *Cell*, 2011, 144(5): 646-674.
- [2] Komohara Y, Fujiwara Y, Ohnishi K, Takeya M, Tumor-associated macrophages: potential therapeutic targets for anti-cancer therapy. *Advanced Drug Delivery Reviews*, 2016, 99: 180-185.
- [3] Qian BZ, Pollard JW, Macrophage diversity enhances tumor progression and metastasis. *Cell*, 2010, 141(1): 39-51.
- [4] Galon J, Costes A, Sanchez-Cabo F, et al., Type, density, and location of immune cells within human colorectal tumors predict clinical outcome. *Science*, 2006, 313(5795): 1960-1964.
- [5] Isidro RA, Appleyard CB, Colonic macrophage polarization in homeostasis, inflammation, and cancer. *American Journal of Physiology-Gastrointestinal and Liver Physiology*. 2016, 311(1): 59-73.
- [6] Kang JC, Chen JS, Lee CH, Chang JJ, Shieh YS, Intratumoral macrophage counts correlate with tumor progression in colorectal cancer. *Journal of Surgical Oncology*, 2010, 102(3): 242-248.
- [7] Popivanova BK, Kitamura K, Wu Y, et al., Blocking TNF-alpha in mice reduces colorectal carcinogenesis associated with chronic colitis. *Journal of Clinical Investigation*, 2008, 118(2): 560-570.
- [8] Liu YX, Fang XJ, Yuan J, et al., The role of corticotropin-releasing hormone receptor 1 in the development of colitis-associated cancer in mouse model. *Endocrine-Related Cancer*, 2014, 21(4): 639-651.

Figure 5. In vivo anti-CAC mechanism of RG@M- γ -CD CNPs. (a) Heat map of differential expression (DE) genes involved in TME mediation and tumor inhibition (RG@M- γ -CD Vs. control, n=3). (b) GOEA of differential expression DE genes (fold change >1.5, P<0.05) visualized as GO network, where nodes with different colors indicated the clusters based on connected GO terms. The clusters were annotated manually by a shared general term. (c) Histograms of DE genes associated with tumor inhibition (left part) and TME (right part) mediation based on GO annotation (-log₁₀ (p value), 27-15). (d) Representative images of Tumor, Ki67 and CD31 staining sections of CAC tumor tissues from different treatment groups (20 \times , scale bar, 50 μ m). (e) mRNA expression levels of inflammatory cytokines (IL-1 β , IL-6, TNF- α and TGF- β 1) in CAC tumors from different treatment groups, quantified by q-PCR (n=3). (f) Representative

contour plots of CD206⁺ and CD80⁺ subsets gated on the CD11b⁺ cell set after different treatments. (g) FACs analysis of TAM (M1, M2) polarization (n=3). (h) Representative fluorescence intensities of CD206⁺ and CD80⁺ subsets from different treatment groups, and medians of fluorescence intensity of CD206⁺ and CD80⁺ subsets (n=3). (i) mRNA expressions of M2-related tumor-supportive cytokines (VEGF-B, PDGF-A, and MMP9) in CAC tumor tissues from different treatment groups (n=3). (j) VEGF staining sections of CAC tumor tissues from different groups (20 \times , scale bar, 50 μ m). Data were expressed as means \pm s.e.m, *P < 0.05, **P < 0.01, ***P < 0.001.

6. In Figure 6 the authors switch to a transplantation model. This means that the therapeutic agent RG is no longer fed but applied systemically. In Fig. 6d it looks as if the PBS-treated control mouse shows the least uptake of RGD of all five groups. Are the authors sure that CT26 colon carcinoma cells absorb the PET tracer? Please provide better PET images or remove the images from the manuscript.

Response:

As described in **Method** section, Balb/c mice are subcutaneously transplanted with CT26 tumors at inguinal region and the tumors are allowed to grow to $\sim 100 \text{ mm}^3$. The tumor bearing mice are then treated with RG@M- γ -CD CNPs and other control formulations via intravenous injection. To visualize tumor inhibition, the PET/CT imaging has been conducted on U-SPECT-II/CT scanning apparatus using ^{18}F -RDG as indicator. As the reviewer's comment, the untreated control group shows the least uptake of ^{18}F -RDG of all five groups. Such phenomenon is attributed to the malignant progression of CT26 tumor. With CT26 tumor progression, the vasculature is structurally and functionally abnormal, blood vessels is leaky, tortuous, dilated, and saccular. Besides, intratumoral pressure compresses intra-tumor blood and lymphatic vessels, which significantly decreases the uptake of ^{18}F -RDG. According to the reviewer's suggestion, we have updated the representative PET/CT images with high resolution (shown in **Fig. 6d** and **s37**).

Please use higher magnification or better images in Fig. 6h. The stainings are really hard to see. The authors claim that “the potent TAM deactivation of RG@M- γ -CD CNP has significant implication...”. This conclusion is not supported by the data the authors present. To really show that macrophages are the main target of the treatment the authors should perform in vivo treatment in mice deficient of macrophages.

Response:

According to the suggestion, the representative immunohistochemistry result for the CT26 model has been updated for better visualization. We thank the reviewer's suggestion for improving the medical preciseness of our manuscript. After careful consideration, we feel that the treatment investigation on macrophage-deficient mice is beyond the scope of this study. In this study, we firstly have synthesized a CRC-targeting carrier M- γ -CD and constructed a host-guest nano-medicine RG@M- γ -CD with channel architecture. Substantial characterization has been performed to expatiate the physicochemical properties of RG@M- γ -CD and analyze

the mechanism underlying the host-guest construction. Secondly, by designing the in vitro and in vivo experiments, we have thoroughly investigated the performances and advantages of RG@M- γ -CD in drug delivery and cancer therapy. Particularly, the biological superiority of M- γ -CD and its synergistic effect with RG have been studied and emphasized. Such host-guest strategy that integrates the CD's chemical virtue with its biological merits can provide a new design philosophy for drug delivery. Thirdly, aiming at the established therapeutic targets of CRC, we have validated the implement of RG@M- γ -CD in AOM-DSS and CT26 models, which has provided a comprehensive and meticulous demonstration for RG@M- γ -CD in CRC treatment. In our opinion, in vivo study using deficient or knockout mice is more important for discovery and confirmation of new therapeutic targets, which is beyond the scope of this study. According to your comment, we have removed the inappropriate description and discussion for better understanding.

Figure 6. Therapeutic efficacy of RG@M-γ-CD in CT26 CRC model. (a) Schematic illustration of CT26 subcutaneous tumor model and treatment via tail intravenous injection (equivalent RG, 10 μg/g, n=7). (b) The pharmacokinetics and (c) bio-distribution of RG, RG@γ-CD and RG@M-γ-CD CNPs (equivalent RG, 10 μg/g) determined by HPLC within 24 h post injection (n=3). (d) Representative PET/CT imaging results and photos of mice CT26 tumor-bearing mice in different treatment groups at 21 day of treatment. (e) The tumor volume changes of CT26 tumor-bearing mice in different treatment groups. (f) Representative photoacoustic

imaging of the mice in different treatment groups and the degree of blood oxygen saturation (HbO₂) within tumors at 21 day of treatment (n=3). (g) Kaplan-Meier survival curves for treatment groups (n=7). (h) CD31, Ki67 and TUNEL staining sections of tumor tissues from different groups (20×, scale bar, 50μm). (i) Flow cytometry analysis of TAM polarization in different treatment groups (n=3). Data were expressed as means ± s.e.m. *P < 0.05, **P < 0.01, ***P < 0.001.

7. Throughout the whole manuscript the authors claim that RG@M- γ -CD reprograms the TME but the authors never show the composition of the TME in neither of the two tumor models they use. It would be of great interest to show the composition of the TME in both models.

Response:

This is a constructive suggestion. We have separated the colon lamina propria cells in AOM-DSS models, and analyzed the subsets of T cells. The results showed similar proportion of CD4⁺ / CD8⁺ T cells in each group, and decreased F4/80⁺ percentage in mix, RG@ γ -CD and RG@M- γ -CD groups was observed. However, since we have not analyzed all of the immune cells in TME due to lacking of massive samples, reagents and specialized works, we could not simply conclude that RG@M- γ -CD had impacts only on macrophages and few effect on other immune cells. In this study, we have emphasized the regulations of RG@M- γ -CD on specific aspects of TME, including production of inflammatory cytokines, expression of tumor-supportive factors, progression of tumor angiogenesis and activation of tumor-associated macrophages. These aspects have been confirmed to play important roles in CRC progression and have become therapeutic targets to CRC. Focusing on these aspects, we have proven that RG@M- γ -CD can suppress the activation of malignant cells, endothelia cells and macrophages. By in vivo studies on CRC models, we demonstrated that RG@M- γ -CD could modulate key TME components of CRC, such as angiogenesis, inflammation, tumor-supportive TAMs, exerting synergistic effects to inhibit tumor growth.

Minor points

I know there is a limit of characters for this manuscript, but I think the discussion is too short. Half of the discussion deals with the physical properties of the nanoparticles and only very few words are said to the functional results of the in vivo experiments.

Response:

This is an important suggestion. In our revision, we have extended the discussion on the functional results of the in vivo experiments, including therapeutic rationale, treatment outcomes and application potentials of RG@M- γ -CD nano-medicine.

The authors use a Mix they never specify (or at least I could not find a word about the composition of the Mix).

Response:

We are sorry for the carelessness. In this study, the physical mixture of equivalent RG and M- γ -CD (corresponding to RG@M- γ -CD) is defined as Mix. We have supplemented the description to specify the composition of Mix.

In the figures and in the figure legends there is no word about the time points used. This makes the figures hard to read. I know there is a limit of characters for the figure legends but the legends used are too short. They do not help to understand the figures.

Response:

According to the suggestion, the figure legends for all figures have been modified and extended to provide the experimental information, such as time points, drug dosages, statistic analysis.

Reviewer #2 (Remarks to the Author): expertise in cyclodextrine-based nanomaterials and drug delivery

The manuscript reports the synthesis of a new gamma-CD mono-functionalized with mannose as a carrier of regorafenib. The manuscript is in the field of cyclodextrin inclusion complexes and reports several experiments in vitro and in vivo. The main problem of the manuscript is the very lacking characterization of the cyclodextrin derivative M- γ -CD and some doubt on the identity of the proposed nanosystem. Chemical-physical characterization and full analysis of data are the first essential step in this kind of study. I think that RG@M- γ -CD in solution is not a nanosystem but is the inclusion complexes of RG and M- γ -CD. Regardless, this aspect may not reduce interest in the results. The title should be changed.

Response:

We thank the reviewer for the constructive suggestions to our manuscript. Based on these suggestions, we have firstly supplemented a series of physicochemical characterization for the new host molecule M- γ -CD and the inclusion complex RG@M- γ -CD, including NMR, FTIR, UV-Vis and MS. Secondly, meticulous data analysis has been performed to demonstrate the structures of target molecules. Thirdly, we have supplemented the detailed description to explain the formation of CD-derived CNPs. Besides, our the morphological and diffraction studies have confirmed that the inclusion complexes can assemble into nanosystems with channel architecture (CNPs) which stay stable in solution during the concerned period of time.

The figures in **Manuscript** and **Supporting Information** have been updated for better understanding. Meanwhile, the relevant assignments and descriptions have been added to provide accurate structural information for M- γ -CD and RG@M- γ -CD, as well as some control molecules used in our study (RG@ γ -CD, Rhodamine@ γ -CD and Rhodamine@M- γ -CD). Furthermore, we have also provided the synthetic methods for RG@M- γ -CD inclusion complex and RG@M- γ -CD nanosystem. We have revised the manuscript according to your suggestions and respond to your comments point-by-point.

Main points

Synthesis: The authors should demonstrate the identity of the new product, C-4 functionalization of Man and C-6 functionalization of CD in the derivative. The authors should add references to the synthetic approach.

Response:

This is an important suggestion to identify the structure of synthesized M- γ -CD. According to the suggestion, we have performed the NMR assay (^1H NMR, ^{13}C NMR, HSQC and HMBC) using DMSO- d_6 as solvent and fully identified the protons/carbons. In ^1H and ^{13}C spectra, the co-existence of peaks from γ -CD and Mannose has been detected, meanwhile the signal of carbonyl C generated from conjugation has also been confirmed at 154.22 ppm. The result has indicates that the Mannose is conjugated with CDI- γ -CD through our synthetic approach. By analyzing the 2D HMBC spectrum, we have found that the carbonyl C on CDI arises correlations with C-1 of Mannose and C-6 of γ -CD, indicating the reaction between C-1 of Mannose and C-6 of γ -CD. The NMR spectra for M- γ -CD have been updated in **Supporting Information (Fig. s6-s9)** and the full assignments of $^1\text{H}/^{13}\text{C}$ resonances have been summarized in **Table s3**. Moreover, MS detection has been re-performed for M- γ -CD using ESI-ToF-MS with syringe pump to maintain the integrity of CD derivative. MS spectrum of M- γ -CD has displayed a strong signal at m/z 1502.4606, relative to Mw [M- γ -CD] (**Fig. 1b, s5**). Besides, signals corresponding to adducts of [M- γ -CD]+Na and [M- γ -CD]+K have also been detected. Together, these results have confirmed the synthesis of M- γ -CD.

According to the analysis, we have modified the **Fig. 1a** to present the accurate structural information of M- γ -CD. In the revision, the detailed synthetic approach and relevant references have been provided in **Methods** section.

"Synthesis of M- γ -CD. The synthesis of M- γ -CD is based on previous work using 1,1'-carbonyldiimidazole (CDI) as a cross linker to conjugate the Mannose and CD^[1-3]. γ -CD (0.65 g, 0.50 mmol) and CDI (0.70 g, 4.30 mmol) were dissolved in DMSO (10.0 mL). The mixture was stirred at room temperature for 3 h and was then precipitated with cold diethyl ether. The obtained CDI-activated γ -CD was filtered, and stored at 4 °C for further use. Next, Mannose (0.60 g, 3.30 mmol) was dissolved in 10 mL DMSO. The CDI-activated γ -CD in 5.0 mL DMSO and 0.3 mL triethylamine (Et3N) were added drop wise to the Mannose solution over 2.5 h with continuous stirring, followed by reaction for 5 h. After the reaction, the mixture was dialyzed in water for 24 h and freeze-dried to produce Mannose- γ -CD (M- γ -CD)."

- [1] Huang HL, Tang GP, Wang QQ, et al., Two novel non-viral gene delivery vectors: low molecular weight polyethylenimine cross-linked by (2-hydroxypropyl)-beta-cyclodextrin or (2-hydroxypropyl)-gamma-cyclodextrin. *Chemical Communications*, 2006, 22: 2382-2384.
- [2] Varshosaz J, Hassanzadeh F, Aliabadi HS, et al., Novel worm-like amphiphilic micelles of folate-targeted cyclodextrin/retinoic acid for delivery of doxorubicin in KG-1 cells. *Colloid and Polymer Science*, 2014, 292(10): 2647-2662.
- [3] SherjeAP, Surve A, Shende P, CDI cross-linked-cyclodextrin nanosponges of paliperidone: synthesis and physicochemical characterization. *Journal of Materials Science: Materials in Medicine*, 2019, 30(6): 74.

Characterization in solution M- γ -CD: A full characterization with NMR of the new derivative M- γ -CD is needed. The few NMR spectra (^1H NMR and ^{13}C NMR) reported were not assigned. Man and CD signals overlap but 2D spectra (COSY; TOCSY, HSQC, HMBC etc) can identify all the protons and the carbons. In fig 1sa small signals between 3.5-2.0 ppm are also evident. What are they? Spectra should be fully assigned, integrated and used to support the proposed structure of the derivative. Furthermore, Mannose forms two anomers and this should be evident in the spectra. In the SI, NMR spectra in D_2O are reported while in the manuscript the authors wrote "d6 DMSO". The spectra in SI are not well readable.

Response:

In the revision, we have supplemented the full NMR characterization of M- γ -CD according to the reviewer's suggestion. ^1H NMR, ^{13}C NMR, HSQC and HMBC spectra have been presented in **Fig. s6-s9**. Fully assignment of NMR shifts has been listed in **Table s3**.

Figure s6. ^1H NMR spectrum (500 MHz, DMSO-D_6 , room temperature) of M- γ -CD

Figure s7. ^{13}C NMR spectrum (500 MHz, DMSO- D_6 , room temperature) of M- γ -CD

Figure s8. HMBC spectrum (500 MHz, DMSO- D_6 , room temperature) of M- γ -CD

Figure s9. HMQC spectrum (500 MHz, DMSO- D_6 , room temperature) of M- γ -CD

D (+)-Mannose (purity, 99%) has been used analyzed for synthesis. We have performed NMR spectroscopy for this mannose and the signal of anomers has not been detected in the spectra (**Fig. s1, s2**).

Figure s1. ^1H NMR spectrum (500 MHz, DMSO-D6, room temperature) of Mannose

Figure s2. ^{13}C NMR spectrum (500MHz, DMSO-D6, room temperature) of Mannose

Before characterization, M- γ -CD has been purified by dialysis to remove residual solvent and unreacted molecules. The feeble signal in 3.5-2.0 ppm has been vanished in the updated ^1H NMR spectrum. To improve the quality of the presented NMR spectra, all raw data have been processed using MestReNova software, and NMR spectra have been reproduced with 600 dpi for better visualization. The relevant descriptions have been updated in the **Main Text** and **Supporting Information** to make the structure of M- γ -CD clear. The solvent used in NMR assay is DMSO-d6 and we have modified the description of experimental conditions according to the reviewer's comment.

A careful analysis of MS spectra is needed. The reported analysis of MS spectra is very approximate. The MS analysis show a main peak at 1511 m/z assigned to M- γ -CD+K (line129). The expected isotopic molecular mass of the molecule shown in Figure 1 is 1502 and so we can expect in MS spectrum peaks at 1502+H, 1502+Na, 1502+K m/z (not 1511). Furthermore, the peak at 758.22 cannot be the double charge of M- γ -CD (line 170). M₍₂₎- γ -CD cited and used in SI was not described and characterized in the manuscript. A full characterization should be added.

Response:

This is a pertinent suggestion. We have re-performed MS analysis for M- γ -CD and all inclusion complexes used in this study. Under the guidance of technician, M- γ -CD and RG@ γ -CD, Rho@ γ -CD, RG@M- γ -CD, Rho@M- γ -CD inclusion complexes have been tested by ESI time-of-flight mass spectrometry (AB Triple TOF 5600^{plus} System). The detailed experimental condition has been described in **Method** section.

"Mass Spectrometry (MS). ESI-ToF-MS detection was conducted using AB Triple TOF 5600^{plus} Mass Spectrometry System (AB SCIEX, Framingham, USA) with syringe pump (flow rate, 20 L/min⁻¹) in positive ion mode. The source voltage was +5.0 kV and the source temperature was normal atmospheric temperature. The pressure of Gas 1 (air) and Gas 2 (air) were set to 25 psi, and the pressure of Curtain Gas (N₂) was set to 25 psi. Declustering potential (DP) and collision energy (CE) was 60 V and 10 V, respectively. Period Cycle Time was 275 ms and the scan range ranged from 200 m/z to 3000 m/z . Exact mass calibration was performed automatically before each analysis employing the Automated Calibration Delivery System. Data acquisition and analysis were performed by Peak View software (version 1.2, AB Sciex, USA)."

Since inclusion complexes are non-covalent molecular systems, syringe pump with flow rate of 20L/min⁻¹ has been applied to avoid damage to inclusion complexes. As shown in **Fig. 1b** and **s5**, the MS of M- γ -CD shows characteristic peaks at m/z 1502.4606, 1525.4503 and 1541.5243, corresponding to [M- γ -CD], [M- γ -CD]+Na and [M- γ -CD]+K, respectively. This result has confirmed the formation mono-substituted CD derivative M- γ -CD. For inclusion complexes (**Fig. 1c, s10**), RG@M- γ -CD yields characteristic peak at m/z 1965.6228, relative to [RG@M- γ -CD] +H-H₂O. In the mass spectrum of Rho@M- γ -CD (**Fig. s26**), the characteristic peak is located at m/z 1928.6642 which corresponds to [Rho@M- γ -CD]-Cl+H-H₂O. The MS analysis of RG@M- γ -CD and Rho@M- γ -CD has confirmed 1:1 host-guest complexation. Spectra

in **Fig. 1b, 1c** have been updated and the relevant explanation has been modified.

Figure s5. ESI-TOF-MS spectrum of M- γ -CD

Figure s10. ESI-TOF-MS spectrum of RG@M- γ -CD

Figure s26. ESI-TOF-MS spectrum of Rho@M- γ -CD

RG@M- γ -CD: Line 135 Authors: "The co-existence of peaks related to the protons of M- γ -CD and RG in ^1H NMR spectrum of RG@M- γ -CD revealed the formation of interlocked molecule." The coexistence of peaks is not evidence of inclusion or interaction, it is the natural sum of the proton resonances of the two molecules. Other experiments should be carried out to investigate the interaction. Spectrum should be fully assigned and integrated.

Response:

According to the reviewer's comment, the inappropriate description has been modified and full assignment of NMR shifts has been analyzed. Firstly, we have fully assigned the NMR shifts for host and guest molecules (**Fig. s6-s9, s11, s12, Table s3 and s4**). Secondly, NMR shifts of RG@M- γ -CD inclusion complex have been assigned according to spectra and previous analysis (**Fig. s13, s14 and Table s5**).

Figure s11. ^1H NMR spectrum (500 MHz, DMSO- D_6 , room temperature) of RG

Figure s12. ^{13}C NMR spectrum (500 MHz, DMSO- D_6 , room temperature) of RG

Table s4. ^{13}C NMR chemical shifts, δ (ppm), and ^1H NMR chemical shifts, δ (ppm), of C-H in RG in DMSO-D6

No.	Chemical Shift δ (ppm)	HMQC δ (ppm)	Group
1	153.03	/	
2	109.46	7.435(1H)	
3	165.98	/	
4	114.64	7.190(1H)	
5	150.98	8.536(1H)	
7	164.21	/	
8	/	8.796(1H)	-NH-
9	26.51	2.806(3H)	
11	148.62		
12	109.69	7.343(1H)	
13	154.23	/	
14	125.41	/	
15	123.11	8.175(1H)	
16	117.60	7.080(1H)	
17	/	9.529(1H)	-NH-
18	152.66	/	
19	/	8.743(1H)	-NH-
20	139.51	/	
21	117.12	8.135(1H)	
22	127.30	/	
23	126.45	/	
24	123.46	7.631(1H)	
25	132.60	7.633(1H)	
26	123.04	/	

Figure s13. ^1H NMR spectrum (500 MHz, DMSO-D₆, room temperature) of RG@M- γ -CD

Figure s14. ^{13}C NMR spectrum (500 MHz, DMSO-D₆, room temperature) of RG@M- γ -CD

Table s5. ^{13}C NMR chemical shifts, δ (ppm), and ^1H NMR chemical shifts, δ (ppm), of C-H in RG@M- γ -CD in DMSO-D₆

No.	Chemical Shift δ (ppm)	HMQC δ (ppm)	Group
1	102.19	4.887(1H)	
2	73.42	3.325(1H)	

3	73.10	3.588(1H)	
4	81.43	3.359-3.370(1H)	
5	72.68	3.533(1H)	
6	60.48	3.625(2H)	
7	154.22	/	-O-CO-O-
8	/	5.758(1H)	-OH
9	/	5.781(1H)	-OH
10	/	4.531(1H)	-OH
1'	94.50	4.884-4.891(1H)	
2'	71.93	3.524-3.543(1H)	
3'	71.10	3.578-3.598(1H)	
4'	67.89	3.370(1H)	
5'	73.70	3.543(1H)	
6'	62.05	3.317(2H)	
1"	153.03	/	
2"	109.46	7.434(1H)	
3"	165.98	/	
4"	114.64	7.190(1H)	
5"	150.98	8.536(1H)	
7"	164.21	/	
8"	/	8.796(1H)	-NH-
9"	26.51	2.806(3H)	
11"	148.62		
12"	109.69	7.343(1H)	
13"	154.23	/	
14"	125.41	/	
15"	123.11	8.175(1H)	
16"	117.60	7.082(1H)	
17"	/	8.738(1H)	-NH-

18"	152.66	/	
19"	/	9.522(1H)	-NH-
20"	139.51	/	
21"	117.12	8.145(1H)	
22"	127.30	/	
23"	126.45	/	
24"	123.46	7.661(1H)	
25"	132.60	7.663(1H)	
26"	123.04	/	

In the spectra of fig 3s (pictures have low quality and are unreadable), there are peaks in the aliphatic region that are not present in the spectra in fig 1sa and are not due to RG protons. What are they? Is the spectrum in D2O? Authors wrote (Line 136) "In 2D NOESY NMR spectroscopy, signals of nuclear overhauser effect (NOE) correlation were detected between the glycogen of M- γ -CD and the amide of RG, implying a partial entrapment of RG in the cavity of M- γ -CD (Fig. 1d and s2b)." This comment is very generic. Amide protons of RG are exchangeable protons and cannot show any signal in D2O. Typically the protons of the included compound show NOE correlation with the H-3, H-5 protons of CD. NOESY spectra (Fig 1d) should be appropriately discussed.

Response:

According to the reviewer's suggestion, we have updated the 2D NOESY NMR spectrum of RG@M- γ -CD in **Fig. s15** for better presentation. RG@M- γ -CD has been dissolved in DMSO-D6 and examined with NMR spectroscopy. Based on previous assignment and 2D spectrum in **Fig. s15**, correlation signals in RG@M- γ -CD have been detected between amide protons of RG (NH-17" and NH-19") and glycogen of M- γ -CD (OH-8, OH-9, OH-10, CH-3 and CH-4). We believe that steric correlation can reflect the complexation between host and guest molecules.

Figure s15. 2D-NOESY spectrum (500 MHz, DMSO-D6, room temperature) of RG@M- γ -CD

The characterization in the solid-state suggests the formation of ordered structure self-assembled. This behavior has been found in the case of CDs. The ordered structure may not exist in the solution. In fact, NMR spectra do not show broad signals typically found in the case of nanosystems. The authors compared RG@M- γ -CD to a mixture of RG and M- γ -CD. A "mixture" of these molecules in water forms the inclusion complex (see calorimetric titrations in SI), with the stability constant reported in the manuscript. On this basis, the differences found between RG@M- γ -CD and mix are not clear.

Response:

According to the previous reports^[1-3], the CD inclusion complexes can be linearly packed on top of each other to assemble into channel-type supramolecular architectures. The construction of supramolecular architectures is driven by (1) the hydrophobic interactions between CDs and guest molecules; (2) hydrogen bonding between adjacent CDs; (3) stereo-hindrance effects. Noteworthy, the CD-derived nano-channel structures can stably exist in the aqueous solution and can proceed secondary self-organization, assembling into nano or micron systems^[4,5]. As non-covalent systems, these CD molecular assemblies have been reported to dissociate in non-equilibrium conditions, such as high temperature, high pH value and high proportion of organic phase^[1,6].

In our study, the construction of RG@M- γ -CD channel-type nano-particles (CNPs) has been divided into three steps to insure the rigid assembling (1) complexation of RG and M- γ -CD; (2) nano-suspension of RG@M- γ -CD; (3) multilevel assembly of RG@M- γ -CD. RG and M- γ -CD are dissolved and then form inclusion complex in ethanol solution (70%). The use of organic solvent with water miscibility allows the contact and complexation between the host and guest molecules. Afterwards, the inclusion molecules spontaneously to self-assembly, which leads to the nano-suspension of RG@M- γ -CD. The nano-suspension in this step has also been regarded as the formation of crystal cores^[7], inducing free RG@M- γ -CD to assemble into ordered structures. In the third step, the experimental conditions are carefully controlled to allow multilevel assembly of RG@M- γ -CD and avoid flocculation. Then the RG@M- γ -CD CNPs is collected the by filtration. The obtained CNPs show high dispersibility and stability in water, meanwhile we find that they undergo dissociation in lipophilic environment and exert the sustained drug release.

Physicochemical characterization is carried out for RG@M- γ -CD in different states. For molecular level, RG@M- γ -CD inclusion complex in aqueous solution is

characterized with NMR and MS to analyze the structure and molecular weight. After assembly into CNPs, RG@M- γ -CD nanosystem is characterized with FTIR, UV-Vis, XRD and TG-DSC. Furthermore, the CNPs are also evaluated with DLS, TEM and AFM to reveal the morphology of RG@M- γ -CD CNPs.

The isothermal titration calorimetry (ITC) has originally been conducted to investigate the inclusion ability of M- γ -CD towards RG. After careful consideration on the reviewer's comments, we find there are certain deficiencies in the experimental design of ITC. (1) The result of ITC actually reflects not only inclusion effect between the two target molecules but also other forms of interaction. (2) The experimental conditions of TIC (including temperature, stirring speed, concentration and reaction volume) are quite different from the conditions of complexation between RG and M- γ -CD. Therefore, we have decided to remove the ITC characterization to avoid inaccurate deduction.

The construction of RG@M- γ -CD inclusion complex and RG@M- γ -CD CNP has been expatiated in the revised manuscript. The detailed description indicates that RG@M- γ -CD inclusion complexes and CNPs are formed by specific experimental procedure rather than mixing the two molecules in water. In addition to the physical and morphological characterization, we believe that the existence of RG@M- γ -CD CNP in solution has been proven.

"Preparation of RG@M- γ -CD CNPs. Co-crystallization method was adapted to prepare the RG@M- γ -CD inclusion complex in ethyl alcohol-water mix solution. RG (0.50 g, \sim 1.0 mmol) and M- γ -CD (1.50 g, \sim 1.0 mmol) were dissolved in 30.0 mL of alcohol water solvent (70%, v/v) in a round bottom flask. The mixture solution was stirred at 75 °C with reflux until the solids dissolving completely. Then the solution was kept for 75 °C for 12 h with continuous stirring to allow the host-guest complexation. After the formation of RG@M- γ -CD inclusion complexes, the solution was cooled slowly to room temperature, leaving 4-6 h for nano-suspension of RG@M- γ -CD. Then the solution was ultrasonically stirred at low speed for 2-4 h, the temperature and speed were carefully controlled to allow multilevel assembly of RG@M- γ -CD and avoid flocculation. The RG@M- γ -CD CNPs was collected the by filtration."

[1] He YF, Fu P, Shen XH, Gao HC, Cyclodextrin-based aggregates and characterization by microscopy. *Micron*, 2008, 39(5): 495-516.

[2] Wu AH, Shen XH, He YK, Investigation on gamma-cyclodextrin nanotube induced by N,N'

- diphenylbenzidine molecule. *Journal of Colloid and Interface Science*, 2006, 297(2): 525-533.
- [3] Celebioglu A, Ipek S, Durgun E, Uyar T, Selective and Efficient Removal of Volatile Organic Compounds by Channel-type γ -Cyclodextrin. Assembly through Inclusion Complexation. *Industrial & Engineering Chemistry Research*, 2017, 56(25): 7345-7354.
- [4] Bonini M, Rossi S, Karlsson G, Almgren M, et al., Self-assembly of beta-cyclodextrin in water. Part 1. Cryo-TEM and dynamic and static light scattering. *Langmuir*, 2006, 22: 1478-1484.
- [5] Connors KA, The stability of cyclodextrin complexes in solution. *Chemical Reviews*, 1997, 97(5): 1325-1357.
- [6] Gonzalez-Gaitano G, Rodriguez P, Isasi JR, et al., The aggregation of cyclodextrins as studied by photon correlation spectroscopy. *Journal of Inclusion Phenomena Macrocyclic Chemistry*, 2002, 44(4): 101-105.
- [7] Choisnard L, Geze A, Putaux JL, Wong YS, et al., Nanoparticles of beta-cyclodextrin esters obtained by self-assembling of biotransesterified beta-cyclodextrins. *Biomacromolecules*, 2006, 7(2): 515-520.

The differences in the UV-Vis spectra are not clear: UV-Vis spectrum of RG@M- γ -CD shows a shift in the band at 250 nm and this may be due to the inclusion but what is the new band at 310 nm? Please analyze the spectrum.

Response:

In the revision, UV-Vis spectrophotometer has been calibrated and UV-Vis spectroscopy has been re-performed to confirm the absorption characteristic of RG@M- γ -CD. Free RG, M- γ -CD and RG@ γ -CD at equivalent RG (10 mg) have also been tested as controls. As shown in updated **Fig. 1j**, the absorbance peak of RG is located at 250 nm with extinction coefficient of $2.4 \times 10^4 \text{ L mol}^{-1} \text{ cm}^{-1}$. In comparison RG@M- γ -CD CNPs display characteristic absorbance at 260 nm and the extinction coefficient is calculated to be $2.0 \times 10^4 \text{ L mol}^{-1} \text{ cm}^{-1}$. Similar hypochromatic shift and decreased extinction coefficient are also observed in the spectrum of RG@ γ -CD, which may be caused by the shielding effect of CDs. We have repeated the scanning while no evident absorbance band has been found at 310 nm. Since characterization has been supplemented and refined, **Fig. 1** has been modified in **Main Text** and complementary data have been added in **Supporting Information**.

Figure 1. Synthesis and chemical characterization of RG@M- γ -CD. (a) Synthetic procedure of RG@M- γ -CD, including activation of γ -CD by CDI (upper), functionalization of γ -CD with mannose (middle) and construction of inclusion complex, RG@M- γ -CD (lower). (b and c) ESI-TOF-MS analysis on M- γ -CD and RG@M- γ -CD. (d) Partial magnification of 2D NOESY spectrum of RG@M- γ -CD. (e) XRD patterns of RG, M- γ -CD, none-channel type RG@M- γ -CD, channel-type RG@M- γ -CD, and stimulated channel-type γ -CD using Material Studio. Schematic illustration of γ -CD stacking with channel-type architecture. Tracing observation of the assembly process by TEM and AFM. (f) Nano-rods with a size of 20-50 nm; (g) nano-bricks with a size of 50-100nm, and (h) nano-spheres with a size of 100-300nm. (i) DLS results of the RG@M- γ -CD CNPs. (j) UV-vis spectra in methyl alcohol of

free RG, M- γ -CD, RG@ γ -CD and RG@M- γ -CD. **(k)** Controlled release profiles of RG@M- γ -CD CNPs under different conditions. Data were expressed as means \pm s.e.m (n = 3).

For all the spectra and other chemical-physical characterizations, the concentration, the solvent used etc should be reported. In the manuscript there are some typos (Line 12 TEM, line 67 TMA, SI 2D NOESY etc).

Response:

This is a constructive suggestion for improving our manuscript. In the revision, we have supplemented the detailed experimental conditions of synthesis and characterizations, as well as in vitro and in vivo studies.

"NMR. The NMR spectra including ^1H , ^{13}C , HMQC, HMBC and NOESY were recorded on a Bruker Avance DMX-500 spectrometer (500 MHz, Billerica). M- γ -CD (15 mg), RG@ γ -CD (18 mg), RG@M- γ -CD (20 mg), Rho@ γ -CD (18 mg) and Rho@M- γ -CD (20 mg) were separately dissolved in 600 μL of DMSO- d_6 and samples were examined in room temperature. The NMR spectra were analyzed with MestReNova software."

"Mass Spectrometry (MS). ESI-ToF-MS detection was conducted using AB Triple TOF 5600plus Mass Spectrometry System (AB SCIEX, Framingham, USA) with syringe pump (flow rate, 20 L/min-1) in positive ion mode. The source voltage was +5.0 kV and the source temperature was normal atmospheric temperature. The pressure of Gas 1 (air) and Gas 2 (air) were set to 25 psi, and the pressure of Curtain Gas (N_2) was set to 25 psi. Declustering potential (DP) and collision energy (CE) was 60 V and 10 V, respectively. Period Cycle Time was 275 ms and the scan range ranged from 200 m/z to 3000 m/z. Exact mass calibration was performed automatically before each analysis employing the Automated Calibration Delivery System. Data acquisition and analysis were performed by Peak View software (version 1.2, AB Sciex, USA)."

"UV-Vis spectrophotometer. RG, RG@ γ -CD, RG@M- γ -CD (equivalent RG, 10 mg) and M- γ -CD were separately solubilized in 25 mL methanol. Then 1 mL of the prepared solutions was diluted to 25 mL with distilled water. After stand for 1 h, the solutions were added into the quartz cell for scanning. The path length of the quartz cell was 1 cm. The absorbance spectra of the samples were recorded by TU-1901 dual-beam UV-vis spectrophotometer (Beijing Pu Analysis General Co., Ltd., China) between 200 and 400 nm."

"X-ray diffraction (XRD). The X-Ray diffraction patterns for RG (5 mg), unordered M- γ -CD (10 mg), none-channel type RG@M- γ -CD (15 mg) and RG@M- γ -CD CNPs (15 mg) were separately measured on Rigaku D/Max-2550PC diffractometer using a rotating-anode Cu-target X-ray ($\lambda = 1.5406 \text{ \AA}$) generator

operated at 40 kV and 250 mA. The range of scanning was from 3.0° to 40.0° (2θ) with an increasing step size of 0.02° and count time of 0.5s-2 s."

"Differential scanning calorimetry (DSC). RG (2 mg), M- γ -CD (5 mg), RG@M- γ -CD CNPs (8 mg) were separately placed in aluminum pan, the DSC and transition glass (TG) analysis was conducted on a TA DSC Q100 differential scanning calorimeter (TA, USA). The heating was carried out at rate of $10^{\circ}\text{C}/\text{min}$ under a N_2 flow of $50\text{ cm}^3/\text{min}$, and the scanning was ranged from 30-380 $^{\circ}\text{C}$."

"Transmission Electron Microscopy (TEM). In order to observe the multiple self-assembly, RG@M- γ -CD-containing solutions were sampled at various time points and were diluted with water. Then the TEM samples (10 μL) were prepared by dropping diluted aqueous solutions onto 400-mesh carbon-coated copper grids with the excess solvent immediately evaporated. The morphologies of CNPs were observed using TEM system (HITACHI-7066, Japan) operating at an accelerating voltage of 120 kV without any staining."

"Atomic Force Microscopy (AFM). A Veeco MultimodeV AFM (Veeco, USA) was used to observe the 3-D morphology of CNPs during self-assembly. RG@M- γ -CD-containing aqueous solutions were sampled at various time points and were diluted with water (10%, v/v). After dispersion by using ultrasonic, the solution (about 10 μL) was adsorbed onto a piece of freshly cleaved mica sheet and dried at 25°C for 12 h before imaging. The imaging was conducted in a tapping mode with a scan rate of 0.5 Hz at room temperature."

"Dynamic light scattering (DLS). The DLS measurement was conducted on Zetasizer Nano ZS series, Malvern Instruments, UK) at 25°C . The measurements were carried out with scattering angle of 173° , and apparent hydrodynamic diameter (D_h) was presented as intensity."

In the revision, we have checked the abbreviations and modified the typos one by one for better understanding. We appreciate the reviewer for these important comments. According to the comments and suggestion, we have thoroughly, carefully performed characterization and analyzed the results to demonstrate the host-guest nano-medicine RG@M- γ -CD CNPs. Through in vitro and in vivo studies, we have meticulously investigated the therapeutic rationale, treatment outcomes and application potentials of RG@M- γ -CD for CRC treatment, which shows meaningful perspective for CD-based drug delivery system. By this revision, we believe we have fully addressed your comments to our best and the quality of manuscript has been improved.

REVIEWER COMMENTS

Reviewer #2 (Remarks to the Author):

The authors supplemented the manuscript with new material. The full assignment of the spectra is still lacking.

Formula reported in SI (Fig s5) is different from that of the manuscript (Fig 1). How many mannose units have been added to CD?

Why did the authors use DMSO-d6? The product is water-soluble. Is the product stable in the water?

¹³C NMR spectrum shows many peaks. Are they byproducts?

Usually, the chemical shift of H-6 of the glucose functionalized is different from the unfunctionalized unit. In this spectrum, this is not observed.

Moreover, DMSO can significantly modify the affinity of the guest for the cavity. NOesy spectra of fig S15 show cross peak only with the OH groups. This data does not support the inclusion in the cavity. In the spectrum of Figure s15 Phase should be corrected.

For this reason, I do not recommend publication of the manuscript in the present form

Reviewer #4 (Remarks to the Author): to replace original Reviewer #1

I was asked to take over as reviewer for the manuscript entitled "Host-guest nano-medicine for colorectal cancer therapy" for Nature Communications regarding the (tumor)immunology part.

I agree with my predecessor, that this is an interesting manuscript on a very important topic. The study demonstrates the effect of a newly designed compound for cancer therapy in the mouse model. There is a clear effect on the tumor development and the authors succeed also in demonstrating effects on the involved cell types.

While I also agree with reviewer 1, that the immunological analysis is rather basic, it demonstrates the points the authors want to make nevertheless. Therefore, I would accept the mechanistic claims, if the materials are exploited to the fullest, which means in this case, we need a proper immune histochemic or fluorescence analysis. See details below.

I am satisfied with the response regarding figure 2. The bar graph 2i should be dots (like 4h).

I am satisfied with the response regarding figure 3. Yet, as it is too obvious, I have to ask an additional question: 3d, f and g show downregulations of all the tested markers. Isn't there anything under any condition that remains unchanged? Too be more specific, can you show, that you are not just killing the cells but really change their inflammatory phenotype? What is the short term (4h) or long term (?) effect of your compound regarding viability of the macrophages? How stable is the phenotype? Additionally, I could not find the polarization protocol in the M&M and the polarization itself is missing in the legend for s31/s32.

I am satisfied with the response regarding figure 4 although the figure makes me wonder about the AOM/DSS-protocol. It is stated "After 7 days post injection, the mice were given 2% (w/v) DSS in drinking water for 5 days and then followed by ordinary water for 5 days. The cycles were repeated thrice to induce chronic inflammation in colon." That should make 47 days. What happened in the rest of the 105 days from the figure? It is further stated, "After 21-day treatment

followed by 14-day symptom observation, the treated mice were sacrificed". These 14-days are missing in the figure. For Fig 4f, I would advise showing the dots as in 4h or 4i. Furthermore, I do not get the meaning of Figure 4g. Should we see less and smaller tumors? That would be very rough considering the size of the pictures and the erratic appearance of the tumors. Fig 4j seems redundant.

Figure 5: I agree with reviewer 1, that a bulk expression analysis is not state-of-the-art for the purpose of demonstrating effects on specific cell types. Yet, the analysis shows a clear effect on the inflammatory state of the tissue. Perhaps I overlooked it, but I could not find a description which tissue was actually used. That samples from the AOM/DSS model were used should be part of the figure legend and how the tissue was prepared (whole colon or specifically dissected tumors) should be described in the M&M. As there are many cell types in the respective tissue that can add to the observed expression signature, including the cytokine expression from the qPCR in Fig 5e, the specific analysis of macrophages via flow cytometry is the most important part of that figure. As I can see only CD11b (which is not specific for macrophages), CD80 and CD206, I would like to know if this was the complete panel or if there was a pre-gating already and what controls has been used to confirm the specificity of the staining (FMO?). A proper macrophage staining with only eight markers is already difficult, let alone just three. Also, I couldn't find cytometry in the M&M (including the used antibodies). Fig. 5h: Was that already gated on CD11b high? For the sake of the message of this experiment, I would advise exploiting the material as good as possible on a single cell level. As the experiment is done and the mice are dead, improved expression analysis and cytometry are impossible, at least use the FFPE material and perform proper immune staining to confirm the FACS or qPCR data (triple staining for cell characterization is a standard method). For 5d, I agree with reviewer 1 about the quality of the staining. The higher magnification in fig s40 does not improve the figure. TUNEL looks OK and seems to support the authors claim. Ki67 is surprisingly weak and with exception of RG@M- γ -CD, there is no trend visible. For CD31 there is no trend at all. Huge problem apart from the staining is the selection of the regions shown. Some of the regions show mostly stroma, others crypts or tumor tissue, some show mild dysplasia, others full adenoma. With figures like this, also the non-histology expert should be able to see clear differences, what is not the case here. On a minor note, the scale bars are very hard to see in print. Same for Fig 5j, there is no clear trend, the huge background or unspecific staining, respectively, does not help in that manner. On a minor note, VEGF should be written somewhere in the figure.

Fig 6: I am satisfied with the response regarding Fig 6a-g, but I agree with reviewer 1, that the main mechanism, the effect on macrophages, could be demonstrated much better. As for Fig. 5, I would not ask for additional experiments, but would expect the authors to exploit the present material to the fullest, with proper FFPE stainings for macrophages to demonstrate the effect of the compound on the TME. Regarding proper staining, I share the opinion of reviewer 1, Fig 6h (or s40b) is not very helpful. I do not see a trend for CD31 neither for Ki67, which looks rather blurry. The authors could consider using CD34 as angiogenesis marker. TUNEL, again, looks OK, yet it appears that brightness or exposure time of control and RG@M- γ -CD are rather different. Additionally, please use dots instead of bars (6f and 6i) and describe the FACS from Fig 6i properly in the M&M including the used antibodies and gating strategy.

Reviewer #2

The authors supplemented the manuscript with new material. The full assignment of the spectra is still lacking.

We thank the reviewer for the detailed evaluation and constructive suggestion. In the revision, we have supplemented the structural information and chemical shifts in the spectra, and have complemented the full assignments for all compounds. Moreover, we have performed NMR characterization for M- γ -CD in D₂O to test the stability. Besides, according to your comment, the NOE spectrum has been re-analyzed to expatiate the correlation between host and guest. For better understanding, some misleading or inaccurate presentations have also been modified in **Figure 1** and **Supporting Information**, and relevant descriptions have been revised. The updated **Figure 1** is shown as blow. With these revisions, we believe that the clear and detailed information for M- γ -CD and RG@M- γ -CD has been provided. The point by point response has been presented blow.

Figure 1. Synthesis and chemical characterization of RG@M- γ -CD. (a) Synthetic procedure of RG@M- γ -CD, including activation of γ -CD by CDI (upper), functionalization of γ -CD with mannose (middle) and construction of inclusion complex, RG@M- γ -CD (lower). (b and c) ESI-TOF-MS analysis on M- γ -CD and RG@M- γ -CD. (d) Partial magnification of 2D NOESY spectrum of RG@M- γ -CD. (e) XRD patterns of RG, M- γ -CD, none-channel type RG@M- γ -CD, channel-type RG@M- γ -CD, and stimulated channel-type γ -CD using Material Studio. Schematic illustration of γ -CD stacking with channel-type architecture. Tracing observation of the assembly process by TEM and AFM. (f) Nano-rods with a size of 20-50 nm; (g) nano-bricks with a size of 50-100nm, and (h) nano-spheres with a size of 100-300nm. (i) Histogram of average diameter. (j) Absorption spectra. (k) Cumulative released RG, %.

(i) DLS results of the RG@M- γ -CD CNPs. (j) UV-vis spectra in methyl alcohol of free RG, M- γ -CD, RG@ γ -CD and RG@M- γ -CD. (k) Controlled release profiles of RG@M- γ -CD CNPs under different conditions. Data were expressed as means \pm s.e.m (n = 3).

Comment 1

Formula reported in SI (Fig s5) is different from that of the manuscript (Fig 1). How many mannose units have been added to CD?

Response

Figure s5 displays the ESI-TOF-MS result of M- γ -CD, and **Figure 1b** is the partial magnification of **Figure s5** in the range from m/z 1440 to m/z 1580. The main peak located at m/z 1502.4606 is relative to Mw [M- γ -CD], confirming that one mannose unit has been added to CD (mole ratio, 1:1). According to the comment, we have checked the two figures, and have found that **Figure 1b** and **Figure 1c** are too close to make a misleading presentation. Therefore, we have modified **Figure 1b** and **Figure 1c** for better demonstration. Besides, all formulas in manuscript and SI have been modified for consistency.

Comment 2

Why did the authors use DMSO-D₆? The product is water-soluble. Is the product stable in the water?

Response

For all the ingredients and products in this study, DMSO-D₆ has been used as solvent to conduct the NMR characterization. Since Regorafenib is almost insoluble in water, we have chosen DMSO-D₆ to dissolve Regorafenib and perform NMR characterization. For the consistency of characterization, other compounds have also been examined with NMR spectroscopy in DMSO-D₆.

As the reviewer's comment, the product has a high solubility in water. In order to characterize the stability of M- γ -CD in water, we have monitored M- γ -CD in D₂O using NMR spectroscopy within 5 days. The spectra and magnification images are shown in **Figure R1**. Along with the time prolonging, the locations and intensities of chemical shifts stay unchanged, which confirms the stability of M- γ -CD in water.

Figure R1. ¹H NMR spectra of M- γ -CD at day1 and day 5 (500 MHz, D₂O, room temperature) and magnification images in the range from 3.40 ppm to 4.00 ppm.

Comment 3

^{13}C NMR spectrum shows many peaks. Are they byproducts?

Response

According to the comment, we have carefully re-examined the ^{13}C NMR spectrum of M- γ -CD. All peaks have been confirmed to be derived from M- γ -CD. For better understanding, we have supplemented the structural information and chemical shifts in the ^{13}C NMR spectrum (**Figure s6** and **Figure s7**). The revised figures are shown as blow.

Figure s6. ^{13}C NMR spectrum (500 MHz, DMSO- D_6 , room temperature) of M- γ -CD.

Figure s7. ¹³C NMR spectrum (500 MHz, DMSO-D₆, room temperature) of M-γ-CD
Table s3. ¹³C NMR chemical shifts, δ (ppm), and ¹H NMR chemical shifts, δ (ppm), of C-H in M-γ-CD in DMSO-D₆

No.	Chemical Shift δ (ppm)	HMQC δ(ppm)	group
1	102.19	4.884-4.891(1H)	-CH-
2	73.42	3.317-3.333(1H)	-CH-
3	73.10	3.578-3.598(1H)	-CH-
4	81.43	3.360-3.370(1H)	-CH-
5	72.68	3.524-3.543(1H)	-CH-
6	60.48	3.621-3.629(2H)	-CH ₂ -
10	154.22	/	-O-CO-O-
8	/	5.752-5.764(1H)	-OH
9	/	5.780-5.783(1H)	-OH
11	/	4.520-4.543(1H)	-OH
1'	94.50	4.884-4.891(1H)	-CH-
2'	71.93	3.524-3.543(1H)	-CH-
3'	71.10	3.578-3.598(1H)	-CH-
4'	67.89	3.360-3.370(1H)	-CH-
5'	73.70	3.524-3.543(1H)	-CH-

6'	62.05	3.621-3.629(2H)	-CH ₂ -
----	-------	-----------------	--------------------

Comment 4

Usually, the chemical shift of H-6 of the glucose functionalized is different from the unfunctionalized unit. In this spectrum, this is not observed.

Response

According to the comment, we have made a spectrum comparison between M- γ -CD and γ -CD in DMSO-D₆ solvent (**Figure R2A, B**). Although H-6 chemical shift of the functionalized glucose shows negligible variations when compared with that of the unfunctionalized one, the profile of H-6 is changed as indicated in the magnification image. To further confirm the difference, we have also conducted NMR characterization for M- γ -CD and γ -CD in D₂O. Spectrum comparison indicates that the main peak of H-6 shifts from 3.802 to 3.795 ppm after functionalization. Besides, the shape of multiplet peak of H-6 on M- γ -CD is different from that of γ -CD (**Figure R2C, D**). The comparison can provide a supporting material to evidence the substitution on C-6.

Figure R2. Comparison of ^1H NMR spectra between $\gamma\text{-CD}$ and $\text{M-}\gamma\text{-CD}$ (500 MHz, room temperature). (A and B) Magnification images of H-6 on $\gamma\text{-CD}$ and $\text{M-}\gamma\text{-CD}$ (DMSO-D_6). (C and D) Magnification images of H-6 on $\gamma\text{-CD}$ and $\text{M-}\gamma\text{-CD}$ (D_2O).

Comment 5

Moreover, DMSO can significantly modify the affinity of the guest for the cavity. NOesy spectra of Fig. s15 show cross peak only with the OH groups. This data does not support the inclusion in the cavity. In the spectrum of Fig. s15 Phase should be corrected.

Response

We appreciate the constructive suggestion from reviewer. For NMR characterization, the test is immediately performed when the DMSO-D₆ solution of RG@M- γ -CD is prepared to diminish the influence of solvent on inclusion complex. We have meticulously re-analyzed the raw NOE spectrum (**Figure s15**) using MestReNova and the additional correlation signals have been found between RG and M- γ -CD. The benzene structure of RG shows correlations with the glycogen of M- γ -CD. Together, in 2D NOE spectrum, multiple correlation signals are detected between RG and M- γ -CD, (1) amide protons of RG (NH-17" and NH-19") and glycogen of M- γ -CD (CH-3, CH-4, OH-8, OH-9 and OH-11); (2) benzene of RG (CH-2", CH-24" and CH-25") and glycogen of M- γ -CD (CH-3 and CH-4). Therefore, we believe that the host-guest complexation is achieved for RG and M- γ -CD, confirming the formation of inclusion complex. The updated NOE spectrum and magnification image are shown as blow.

Figure R3. (A) NOE spectrum and (B) magnification image for RG@M- γ -CD (500 MHz, DMSO-D₆, room temperature).

Reviewer #4 (Remarks to the Author): to replace original Reviewer #1

I was asked to take over as reviewer for the manuscript entitled "Host-guest nano-medicine for colorectal cancer therapy" for Nature Communications regarding the (tumor) immunology part.

I agree with my predecessor, that this is an interesting manuscript on a very important topic. The study demonstrates the effect of a newly designed compound for cancer therapy in the mouse model. There is a clear effect on the tumor development and the authors succeed also in demonstrating effects on the involved cell types. While I also agree with reviewer 1, that the immunological analysis is rather basic, it demonstrates the points the authors want to make nevertheless. Therefore, I would accept the mechanistic claims, if the materials are exploited to the fullest, which means in this case, we need a proper immune histochemic or fluorescence analysis. See details below.

We are grateful to you for your positive comments and constructive suggestions on our work. During the revision, we have performed immunofluorescence using the FFPE materials to characterize the TAMs and confirm the FACS data. We have also updated the results of immunohistochemistry. First, the staining images of TUNEL and Ki-67 have been captured in tumor regions for each group to show the different inhibitory effects of involved formulations and highlight the superiority of RG@M- γ -CD nano-medicine. Second, according to your valuable suggestion, the anti-angiogenesis effect has been evaluated by CD34 staining immunohistochemistry. The updated staining images have showed the micro-vessels in tumor regions clearly, and the anti-angiogenic trend can be demonstrated with the representative images. Besides, we have made the figure modification and supplemented the detailed protocols for FACS, AOM-DSS modeling and other experiments. With these revisions based on your comments, we believe that the quality of immunological analysis has been improved. The point by point response has been presented below.

Comment 1

I am satisfied with the response regarding figure 2. The bar graph 2i should be dots (like 4h).

Response

Thank you for the valuable suggestion. The bar graph 2i has been remade with data dots (n=3), and the updated **Figure 2** has been shown below.

Figure 2. In vitro anti-tumor effect of RG@M-γ-CD CNPs. Cell internalization of CNPs. (a and b) Representative CLSM images of the CT26 and HT29 cells cultured with Rho@M-γ-CD and Rho@γ-CD CNPs (equivalent Rho, 5.0 μM) for 2 h. Blue

fluorescence indicated nuclear staining with DAPI; green fluorescence indicated β -actin staining with FITC-antibody; red fluorescence presented the internalization of CNPs. Scale bar, 20 μm . (c) FACS analysis and statistical result of intracellular fluorescence intensities induced by internalized Rho@M- γ -CD and Rho@ γ -CD (equivalent Rho, 5.0 μM and 10.0 μM) in CT26 and HT29 cells (2×10^4 cells/sample, $n=3$). (d) In vitro cytotoxicity of different formulations (equivalent RG, 0.5-10.0 μM) towards CT26 and HT29 cells with 12 h of treatment. Proliferation inhibition and apoptosis induction of different formulations (equivalent RG, 2.0 μM) was evaluated in CT26 and HT29 cells by BrdU assay and Annexin V-PI assay, respectively. (e) Representative FACS analysis and (f) statistical result of proliferation rate for treatment groups ($n=3$). (g) Representative FACS analysis and (h) statistical result of apoptosis rates for treatment groups ($n=3$). (i) Representative images of in vitro tube formation by HUVECs treated with different formulations (equivalent RG, 2.0 μM) for 6 h. The numbers of micro-tube in wells were counted ($n=3$). The data were expressed as means \pm s.e.m. * $P < 0.05$, ** $P < 0.01$. (j) Western blot analysis of phosphorylation of VEGFR-2 and PDGFR- β in HUVECs treated with different formulations (equivalent RG, 2.0 μM , 6 h).

Comment 2

I am satisfied with the response regarding figure 3. Yet, as it is too obvious, I have to ask an additional question: 3d, f and g show downregulations of all the tested markers. Isn't there anything under any condition that remains unchanged? To be more specific, can you show, that you are not just killing the cells but really change their inflammatory phenotype? What is the short term (4h) or long term (?) effect of your compound regarding viability of the macrophages? How stable is the phenotype? Additionally, I could not find the polarization protocol in the M&M and the polarization itself is missing in the legend for s31/s32.

Response

Thank you very much for this constructive comment. In our experiment design, the cytotoxicity of CNPs has been firstly evaluated with the target cells, including CRC cells, peritoneal macrophages and HUVECs. According to your suggestion, we have supplemented the cytotoxicity evaluation within 4 h and 12 h (shown in **Figure s29** and **Table s10**). The results indicate that RG@ γ -CD and RG@M- γ -CD induce negligible cell death ($8.3\pm 3.6\%$) upon peritoneal macrophages within 4 h, which is consistent with the MTT result of CRC cells. After 12 h, the cell viability drops in a dosage-dependent manner, and the IC₅₀ values for RG@ γ -CD and RG@M- γ -CD are 2.38 μ M and 1.76 μ M, respectively. In order to avoid cell death during in vitro phenotype investigation, the RG content of different formulations is controlled at 2.0 μ M and the treatment is lasted for 4 h. Specifically, peritoneal macrophages seeded in 6-well plates (1×10^6 cells/well) are treated with different formulations for 4 h, then incubated with LPS (10 ng/mL), IL-4 (40 ng/mL) or tumor supernate for 2 h. Then the cells and culture media are collected for quantitative real-time PCR and ELISA assay, respectively. For culture media, the inflammatory cytokines (IL-6, TNF- α and TGF- β 1) and M2-associated markers (VEGF-B, PDGF-A and MMP-9) are quantified by mouse ELISA kit (Neo bioscience) according to the manufacturer's instructions. For collected cells, total RNA is isolated with RNAiso Plus (Takara) and single-strand cDNA is synthesized by using reverse transcription kit (Tyobo). Quantitative real-time PCR is performed on CFX-Touch real-time PCR system (Bio-Rad) using SYBR

Green reagent (Roche). The mRNA level for each target gene is normalized to the corresponding GAPDH level, which is obtained by $2^{-\Delta\Delta CT}$ method (the primers we used have been listed in **Supporting Information**).

Table s11 Sequence of primers used for RT-PCR

mRNA	Primers (5'-3')
IL-1 β	F: TGTGAAATTGCCACCTTTTGA
	R: TGTCCCTCATCCTGGAAGGTC
IL-6	F: CTGCAAGAGACTTCCATCCAG
	R: AGTGGTATAGACAGGTCTGTTGG
TNF- α	F: GAACTGGCAGAAGAGGGCACT
	R: AGGGTCTGGGCCATAGAACT
TGF- β 1	F: GGAGAGCCCTGGATACCAAC
	R: CAACCCAGGTCCCTTCCTAAA
Arg-1	F: AGAGCTGACAGCAACCCTGT
	R: GGATCCAGAAGGTGATGGAA
IL-10	F: GCTGGACAACATACTGCTAACC
	R: ATTTCCGATAAGGCTTGGCAA
CD206	F: CTCTGTTTCAGCTATTGGACGC
	R: TGGCACTCCCAAACATAATTTGA
VEGF-B	F: GCCAGACAGGGTTGCCATAC
	R: GGAGTGGGATGGATGATGTCAG
PDGF- α	F: TGGCTCGAAGTCAGATCCACA
	R: TTCTCGGGCACATGGTTAATG
MMP-9	F: GGACCCGAAGCGGACATTG
	R: CGTCGTCGAAATGGGCATCT
GAPDH	F: ACCCAGAAGACTGTGGATGG
	R: CTTGCTCAGTGTCTTGCTG

According to your comment, we have evaluated the stability of phenotype using ELISA. Peritoneal macrophages are treated with RG@M- γ -CD for 4 h (RG content, 2.0 μ M). Then they are incubated with LPS (or IL-4)-containing DMEM medium for 12 h. Afterwards, the medium is collected and the phenotype-related factors are quantified using mouse ELISA kit. In comparison with the result of 2 h, the quantification result of 12 h indicates that the expression levels of phenotype-related factors are further increased. For control group, the M1-related factors and M2-related factors are increased by 2.8~4.0-fold and 3.2~4.4-fold, respectively, compared with 2 h result. With RG@M- γ -CD treatment, the inflammatory cytokines and M2-related factors are only increased by 1.6~2.2-fold and 2.0~2.4-fold, respectively, confirming

the stable inhibition of RG@M- γ -CD on PM polarization (**Figure s32**). The MTT and ELISA results have been supplemented in **Supporting Information**, and corresponding protocol has been added in **Method** Section. Besides, we have also modified the **Figure 3c** to illustrate the experimental process regarding peritoneal macrophages, the updated **Figure 3** has been shown blow.

Figure s29 Cytotoxicity test in HUVECs and peritoneal macrophages treated with RG, RG@M- γ -CD and RG@M- γ -CD for 4 h and 12 h. Concentrations of equivalent RG ranged from 0.5 μM to 10 μM . Data were expressed as means \pm s.e.m (n=5).

Figure s32. Evaluation of the stability of phenotype regulation by RG@M- γ -CD. PMs were treated with RG@M- γ -CD for 4 h (RG content, 2.0 μM), then were incubated with LPS or IL-4 for 12 h. The medium was collected and the phenotype-related cytokines were quantified by ELISA (n=3). The comparison was then made between 12 h result and 2 h result to evaluate the stability of phenotype regulation by RG@M- γ -CD.

Figure 3. In vitro regulatory effect of RG@M- γ -CD upon macrophage. Cell internalization of CNPs by macrophages. (a) Representative CLSM images of the peritoneal macrophages cultured with Rho@M- γ -CD and Rho@ γ -CD CNPs (equivalent Rho, 5.0 μ M) for 2 h. Blue fluorescence indicated nuclear staining with DAPI; green fluorescence indicated mannose receptors staining with FITC-antibody; red fluorescence presented the internalization of CNPs. Scale bar, 20 μ m. (b) FACS analysis and statistical result of intracellular fluorescence intensities induced by internalized Rho@M- γ -CD and Rho@ γ -CD (equivalent Rho, 5.0 μ M and 10.0 μ M) in peritoneal macrophages (2×10^4 cells/sample, n=3). (c) Schematic illustration of peritoneal macrophages isolation, stimulation and phenotype evaluation. Peritoneal macrophages were treated with different formulations for 4 h, followed by incubation with LPS, IL-4 or tumor supernate for 2 h. Then the cells and culture media were collected for phenotype evaluation. (d) mRNA expression levels of pro-inflammatory cytokines by peritoneal macrophages treated with different formulations (equivalent RG, 2.0 μ M, n=3). (e) Western blot analysis of p-p65, p65 and abca1 expressions in macrophages under different treatments (equivalent RG, 2.0 μ M, 4 h). (f) mRNA

expression levels of M2-associated markers by macrophages under different treatments (equivalent RG, 2.0 μ M, n=3). (g) mRNA expression levels of VEGF-B, PDGF- α and MMP-9 by macrophages under different treatments (equivalent RG, 2.0 μ M, n=3). Data were expressed as means \pm s.e.m, *P < 0.05, **P < 0.01.

Comment 3

I am satisfied with the response regarding figure 4 although the figure makes me wonder about the AOM/DSS-protocol. It is stated "After 7 days post injection, the mice were given 2% (w/v) DSS in drinking water for 5 days and then followed by ordinary water for 5 days. The cycles were repeated thrice to induce chronic inflammation in colon." That should make 47 days. What happened in the rest of the 105 days from the figure? It is further stated, "After 21-day treatment followed by 14-day symptom observation, the treated mice were sacrificed". These 14-days are missing in the figure. For Fig 4f, I would advise showing the dots as in 4h or 4i. Furthermore, I do not get the meaning of Figure 4g. Should we see less and smaller tumors? That would be very rough considering the size of the pictures and the erratic appearance of the tumors. Fig 4j seems redundant.

Response

We thank you for pointing out the inappropriate illustration in **Figure 4c**. In order to demonstrate the modeling of AOM-DSS, we have modified **Figure 4c** by making clear marks corresponding to different stages and supplementing the scheme of 14-day observation. The modeling of AOM-DSS requires about 100 days, including mutagenesis, colonitis and carcinogenesis. Specifically, the C57BL/6 mice were intraperitoneally injected with AOM for mutagenesis. After 7 days post injection, the mice were given with 2% DSS in drinking water for 5 days, followed by regular water for 5 days. The cycle was totally repeated four times to induce colonitis. For the next 50 days, the model mice were fostered to allow the development of CAC. At 105 day post-AOM injection, 3 mice were sacrificed, their colons were excised and analyzed to confirm the establishment of CAC model. CAC mice were randomly divided into six groups, orally administrated with PBS, free RG, M- γ -CD, mixture, RG@ γ -CD and RG@M- γ -CD, respectively. During treatment, the changes of body weight, survival rate and DAI were recorded every 2 days. After 21-day treatment followed by 14-day symptom observation, the treated mice were sacrificed and the colons were surgically excised. Correspondingly, the modeling protocol has been refined and supplemented in **Method** Section.

According to your suggestion, we have substituted histogram with dot diagram for **Figure 4f**, and made marks for **Figure 4g** to indicate the tumors in each treatment group. In **Figure 4g**, two representative images of tumor-loading colon tissues are organized for each treatment group to show the therapeutic effects. As your comment, some tumors in the images are erratic and small. For better visualization, we have improved the image resolution and used red arrows to indicate tumor areas. **Figure 4j** can be obtained by **Figure 4h** and **4i**, which seems redundant. In some research reports regarding AOM-DSS CAC model, the subfigure of average tumor volume are presented for better understanding [1]. For this reason, we may keep **Figure 4j**.

[1] Bijun Cui, Shen Lu, Lihua Lai, Yiwei Xie, Jia He, Yue Xue, Peng Xiao, Ting Pan, Luoquan Chen, Yang Liu, Xuetao Cao, Qingqing Wang. **Protective function of interleukin 27 in colitis-associated cancer via suppression of inflammatory cytokines in intestinal epithelial cells.** *OncoImmunology*, 2017, 6: e1268309.

Figure 4. Therapeutic efficacy of RG@M- γ -CD in CAC model. Bio-distribution of free Rho, Rho@ γ -CD and Rho@M- γ -CD CNPs was evaluated with ex vivo imaging in C57 mice via oral administration. **(a)** Representative ex vivo fluorescence images and mean fluorescence intensities within intestinal tissues at different time points post administration (equivalent Rho, 10 μ g/g, n=3). **(b)** Representative ex vivo fluorescence imaging of main organs at 2 h, 6 h, 12 h and 24 h post administration of Rho@M- γ -CD CNPs. **(c)** Schematic illustration of the CAC mouse model establishment based on AOM-DSS method. Changes of **(d)** body weight and **(e)** disease activity index for different treatment groups during CAC modeling and anti-cancer experiment (equivalent RG, 10 μ g/g, n=10). **(f)** Representative images of colons, and colon lengths from different treatment groups (n=5). **(g)** Representative colons with tumor loading from different treatment groups. The red arrows indicated tumor areas. **(h)** Colon tumor numbers, **(i)** tumor volumes, **(j)** and average tumor sizes from treatment groups (n=5). Data were expressed as means \pm s.e.m, *P < 0.05, **P < 0.01, ***P < 0.001.

Comment 4

Figure 5: I agree with reviewer 1, that a bulk expression analysis is not state-of-the-art for the purpose of demonstrating effects on specific cell types. Yet, the analysis shows a clear effect on the inflammatory state of the tissue. Perhaps I overlooked it, but I could not find a description which tissue was actually used. That samples from the AOM/DSS model were used should be part of the figure legend and how the tissue was prepared (whole colon or specifically dissected tumors) should be described in the M&M. As there are many cell types in the respective tissue that can add to the observed expression signature, including the cytokine expression from the qPCR in Fig 5e, the specific analysis of macrophages via flow cytometry is the most important part of that figure. As I can see only CD11b (which is not specific for macrophages), CD80 and CD206, I would like to know if this was the complete panel or if there was a pre-gating already and what controls has been used to confirm the specificity of the staining (FMO?). A proper macrophage staining with only eight markers is already difficult, let alone just three. Also, I couldn't find cytometry in the M&M (including the used antibodies). Fig. 5h: Was that already gated on CD11b high? For the sake of the message of this experiment, I would advise exploiting the material as good as possible on a single cell level. As the experiment is done and the mice are dead, improved expression analysis and cytometry are impossible, at least use the FFPE material and perform proper immune staining to confirm the FACS or qPCR data (triple staining for cell characterization is a standard method).

Response

We thank for the reviewer's important comment. For gene expression analysis, small pieces of colon tissue with tumors was used to perform the microarray. We have added the description in the M&M section. For flow cytometry, colon lamina propria cells from CAC mice were separated and analyzed of the subsets of macrophages. As you mentioned in the letter, intestinal macrophages are very heterogeneous. In a healthy colon, monocytes attracted to the intestine gradually differentiate into tissue-resident macrophages by losing Ly6C expression, up-regulating the expression of macrophage markers, such as CX3CR1, F4/80, CD64 and CD11c. In CAC,

macrophages are more heterogeneous. During our experiment, we tried some strategy to identify TAMs, but the staining was quite unstable probably owing to the different inhibitory effects of nano-formulations. We feel sorry for not identifying macrophages accurately by staining more markers. We used F4/80, CD11b to identify mature macrophages, and used CD206, CD80 to identify M2-type, M1-type macrophages respectively. Cells stained with only one fluorochrome-conjugated antibody (the same fluorophore as the sample) were analyzed individually to do the fluorescence compensation, and isotype controls were used as negative controls. In **Figure 5f**, we have performed F4/80⁺ CD11b⁺ pre-gating, and then the CD206⁺ and CD80⁺ subpopulations have been analyzed to characterize the TAMs. In **Figure 5h**, we have re-analyzed the fluorescence intensities with F4/80⁺CD11b⁺ pre-gating. The FACS results have been updated in **Figure 5** and corresponding protocols have been supplemented in **Method** section (and **Supporting Information**).

Figure 5. (f) Representative contour plots of CD206⁺ and CD80⁺ subsets gated on the F4/80⁺ CD11b⁺ cell set for different treatments. (g) Statistical analysis of TAM (M1, M2) polarization (n=3).

Flow cytometry. The colons were cut into pieces, followed by EDTA incubation. Then the incubation was digested by collagenase IV (300 U/mL, Worthington) in 5% FBS PBS solution at 37 °C for 1 h. After digestion, the cell suspension was filtered to get single-cell suspension containing lamina propria cells. Single-cell suspensions were then stained and subsequently analyzed with NovoCyteTM flow cytometer (ACEA). The antibodies used were listed in **Supporting Information**. Flow cytometry analysis was done with the FlowJo software.

Table s12. Antibodies used in FACS analysis for CAC model

FACS	
APC anti-mouse F4/80	Biolegend, 123116
PerCP anti-mouse/human CD11b	Biolegend, 101229
FITC anti-mouse CD206	Biolegend, 141703
PE anti-mouse CD80	Biolegend, 104707
APC/Cyanine7 anti-mouse CD45	Biolegend, 103116

We thank the reviewer for the valuable suggestion regarding exploiting the FFPE material to perform cell characterization. According to your suggestion, we have performed the immunofluorescence staining using anti-F4/80, anti-CD206 and anti-CD80 antibodies to indicate the intratumoral TAMs for different groups. As shown in **Figure s37**, the levels of intratumoral TAM infiltration were markedly decreased in CNP groups, particularly RG@M- γ -CD CNPs showed the most potent deactivation effect.

Figure s37. Immunofluorescence analysis on colon tumor sections by F4/80, CD206 and CD80 staining. Representative images of tumor areas from treatment groups were captured using inverted fluorescence microscopy. Scale bar, 50 μ m.

Comment 5

For 5d, I agree with reviewer 1 about the quality of the staining. The higher magnification in fig s40 does not improve the figure. TUNEL looks OK and seems to support the authors claim. Ki67 is surprisingly weak and with exception of RG@M- γ -CD, there is no trend visible. For CD31 there is no trend at all. Huge problem apart from the staining is the selection of the regions shown. Some of the regions show mostly stroma, others crypts or tumor tissue, some show mild dysplasia, others full adenoma. With figures like this, also the non-histology expert should be able to see clear differences, what is not the case here. On a minor note, the scale bars are very hard to see in print. Same for Fig 5j, there is no clear trend, the huge background or unspecific staining, respectively, does not help in that manner. On a minor note, VEGF should be written somewhere in the figure.

Response

This is an important comment to improve the quality of the staining. First, we have consulted a histopathological expert to distinguish the tumor regions. Owing to the different inhibitory effects on CAC, the staining images from treatment groups show some differences in tumor size and shape. We have shown the representative images of colon tumors from each group ($2.4 \times$ magnification, **Figure R1**) as references [1,2]. Then the representative staining images in high magnification have been carefully re-selected for each group to show the differences of apoptosis, proliferation and angiogenesis within tumor regions.

[1] Hongxia Zhang, Zhisheng Xu, Hen Lin, Mi Li, Tian Xia, Kaisa Cui, Suyun Wang, Youjun Li, Hongbing Shu, Yanyi Wang. **TRIM27 mediates STAT3 activation at retromer-positive structures to promote colitis and colitis-associated carcinogenesis.** Nature Communications, 2018, 9: 3441.

[2] Xinyang Song, Hanchao Gao, Yingying Lin, Yikun Yao, Shu Zhu, Jingjing Wang, Yan Liu, Xiaomin Yao, Guangxun Meng, Nan Shen, Yufang Shi, Yoichiro Iwakura, Youcun Qian. **Alterations in the microbiota drive interleukin-17C production from intestinal epithelial cells to promote tumorigenesis.** Immunity, 2014, 40: 140.

Figure R1 Representative images of immunohistochemical staining of colon tumors from treatment groups ($2.4 \times$ magnification)

Second, the staining sections of TUNEL and Ki67 for each group have been scanned in consistent condition to present the trend of apoptosis or proliferation within tumor regions. Meanwhile, according to your suggestion, we have re-performed the immunohistochemical staining using CD34 as marker to investigate the tumor angiogenesis. Thanks to your advice, the microvessels within tumor regions have been clearly stained using anti-CD34 anti-body (Abcam, ab81289). The staining sections have been scanned and the representative images have been updated in **Figure 5d**. Besides, we have also bolded the scale bars for clear indication. As compared to the controls, there was a significant decrease in angiogenesis and survival rate in RG@M- γ -CD group, confirming the inhibitory effects of RG@M- γ -CD CNPs. The VEGF staining sections were re-selected to minimize the appearance of unspecific staining. The updated images have been shown in **Figure 5j**. With these modifications, the **Figure 5** has been updated and the figure legend has been amended (shown as below).

Figure 5d Representative images of Tunel, Ki67 and CD34 staining sections of CAC tumor tissues from different treatment groups (scale bar, 50 μ m)

Figure 5j Representative images of VEGF staining sections of CAC tumor tissues from different treatment groups (scale bar, 50 μ m)

Figure 5. In vivo anti-CAC mechanism of RG@M- γ -CD CNPs. (a) Heat map of differential expression (DE) genes involved in TME mediation and tumor inhibition (RG@M- γ -CD VS. control, n=3). (b) GOEA of differential expression DE genes (fold change >1.5, P<0.05) visualized as GO network, where nodes with different colors indicated the clusters based on connected GO terms. The clusters were annotated manually by a shared general term. (c) Histograms of DE genes associated with tumor inhibition (left part) and TME (right part) mediation based on GO annotation (-log₁₀ (p value), 27-15). (d) Representative images of Tunel, Ki67 and CD34 staining sections of CAC tumor tissues from different treatment groups (scale bar, 50 μ m). (e) mRNA

expression levels of inflammatory cytokines (IL-1 β , IL-6, TNF- α and TGF- β 1) in CAC tumors from different treatment groups, quantified by PCR (n=3). (f) Representative contour plots of CD206⁺ and CD80⁺ subsets gated on the F4/80⁺ CD11b⁺ cell set for different treatments. (g) Statistical analysis of TAM (M1, M2) polarization (n=3). (h) Representative fluorescence intensities and median values of CD206⁺ and CD80⁺ subsets gated on the F4/80⁺ CD11b⁺ cell set for different treatment groups (n=3). (i) mRNA expressions of M2-related tumor-supportive cytokines (VEGF-B, PDGF- α , and MMP-9) in CAC tumor tissues from different treatment groups (n=3). Data were expressed as means \pm s.e.m, *P < 0.05, **P < 0.01, ***P < 0.001. (j) Representative images of VEGF staining sections of CAC tumor tissues from different groups (scale bar, 50 μ m).

Comment 6

Fig 6: I am satisfied with the response regarding Fig 6a-g, but I agree with reviewer 1, that the main mechanism, the effect on macrophages, could be demonstrated much better. As for Fig. 5, I would not ask for additional experiments, but would expect the authors to exploit the present material to the fullest, with proper FFPE stainings for macrophages to demonstrate the effect of the compound on the TME. Regarding proper staining, I share the opinion of reviewer 1, Fig 6h (or s40b) is not very helpful. I do not see a trend for CD31 neither for Ki67, which looks rather blurry. The authors could consider using CD34 as angiogenesis marker. TUNEL, again, looks OK, yet it appears that brightness or exposure time of control and RG@M- γ -CD are rather different. Additionally, please use dots instead of bars (6f and 6i) and describe the FACS from Fig 6i properly in the M&M including the used antibodies and gating strategy.

Response

Thank you very much for your constructive suggestions regarding the CT26 CRC model. First, in order to improve the quality of immunohistochemical sections, we have re-performed the Ki67 and CD34 staining using the FFPEs as you suggested. Then the sections have been scanned carefully with same parameters (brightness, gain level and exposure time). All representative images have been organized in 600 dpi. We hope that the updated staining images can demonstrate the inhibitory effect of RG@M- γ -CD on malignancy.

Figure 6h Representative images of CD34, Ki67 and TUNEL staining sections of CT26 tumor tissues from different treatment groups (scale bar, 50 μ m)

Second, we have supplemented the immunofluorescence staining to demonstrate the effect of RG@M- γ -CD on the macrophages within TME for CT26 CRC model. According to your suggestion, triple staining method has been used for cell characterization, specifically DAPI for nucleus, red for F4/80 and green for CD206. The co-localization of green and red fluorescence indicates TAMs. In consistent with FACS result, immunofluorescence analysis confirms that both RG@ γ -CD and RG@M- γ -CD CNPs effectively inhibit the TAM activation, and in comparison, RG@M- γ -CD exhibits more potent inhibition effect. Representative images have been organized and shown in **Figure s40**.

Figure s40. Immunofluorescence analysis on CT26 tumor sections by F4/80 and

CD206 staining. Representative images of tumor areas from treatment groups were captured using inverted fluorescence microscopy. Scale bar, 50 μm .

Third, according to your suggestion, we have substituted the histograms with dot diagrams for **Figure 6f and 6i**. The updated **Figure 6** has been shown below. FACS of TAM population for CT26 CRC model has been re-processed. CD206⁺ F4/80⁺ subset gated on CD45⁺ cell set within tumor tissues from different treatment groups has been analyzed to characterize the regulatory effect of RG@M- γ -CD CNPs on TAMs [1]. As shown in **Figure s39**, treatment with RG, Mix, RG@ γ -CD or RG@M- γ -CD significantly decreased the intratumoral TAM population (CD206⁺ F4/80⁺ subset gated on CD45⁺ cell set) compared with control, and ~5.0-fold decrease of CD206⁺ F4/80⁺ subpopulation was observed in RG@M- γ -CD group. We have supplemented the gating strategy in **Method** section and all the used anti-bodies have been listed in **Supporting Information**.

[1] Yanxian Feng, Ruoyu Mu, Zhenzhen Wang, Panfei Xing, Junfeng Zhang, Lei Dong, Chunming Wang. A toll-like receptor agonist mimicking microbial signal to generate tumor-suppressive macrophages. *Nature Communications*, 2019, 10: 2272.

Figure s39. Representative contour plots of CD206⁺ F4/80⁺ subsets gated on CD45⁺

cell set in CT26 tumor tissues after different treatments.

Table s12. Antibodies for immunohistochemistry, immunofluorescence and FACS

Immunohistochemistry	
TUNEL	Roche, 39081800
Anti-Ki67	Abcam, ab16667
Anti-mouse CD34	Abcam, ab81289
Anti-mouse VEGF	Abcam, ab46154
Immunofluorescence	
DAPI	Abcam, ab104139
Anti-mouse F4/80	Cell Signaling Technology, 70076
Anti-mouse CD206	Santa Cruz, sc-58986
Anti-mouse CD80	Santa Cruz, sc-376012
FACS	
APC anti-mouse F4/80	Biolegend, 123116
PerPC anti-mouse/human CD11b	Biolegend, 101229
FITC anti-mouse CD206	Biolegend, 141703
PE anti-mouse CD80	Biolegend, 104707
APC/Cyanine anti-mouse CD45	Biolegend, 103116

Figure 6. Therapeutic efficacy of RG@M-γ-CD in CT26 CRC model. (a) Schematic illustration of CT26 subcutaneous tumor model and treatment via tail intravenous injection (equivalent RG, 10 μg/g, n=7). (b) The pharmacokinetics and (c) bio-distribution of RG, RG@γ-CD and RG@M-γ-CD CNPs (equivalent RG, 10 μg/g) determined by HPLC within 24 h post injection (n=3). (d) Representative PET/CT imaging results and photos of mice CT26 tumor-bearing mice in different treatment groups at 21 day of treatment. (e) The tumor volume changes of CT26 tumor-bearing mice in different treatment groups. (f) Representative photoacoustic

imaging of the mice in different treatment groups and the degree of blood oxygen saturation (HbO₂) within tumors at 21 day of treatment (n=3). (g) Kaplan-Meier survival curves for treatment groups (n=7). (h) CD34, Ki67 and TUNEL staining sections of tumor tissues from different groups (scale bar, 50μm). (i) Flow cytometry analysis of TAM population (CD206⁺ F4/80⁺ subset gated on CD45⁺ cell set) in different treatment groups using (n=3). Data were expressed as means ± s.e.m. *P < 0.05, **P < 0.01, ***P < 0.001.

REVIEWER COMMENTS

Reviewer #2 (Remarks to the Author):

The authors very slightly improved the manuscript in the section of the chemical characterization of the new mannose derivative.

Characterization is still very lacking. The assignment of the spectra added is very superficial. Authors report the characterization of a new compound with a new synthetic route. They should show the identity first. Especially since in the biological data the differences between RG@M-CD and RG@CD are often minimal, smaller than the experimental error.

In Table s3: -CH- and OH.... Which -CH- and OH? A deep assignment should be provided to support the synthetic route. It is strange that, in the ¹H spectra, the signals of mannose and CD are always overlapped, and that in Fig R2 CD and M-CD show the same. Of course, the shift from 3.802 to 3.795 is not evidence of functionalization (Fig R2C). 2D NMR spectra can quickly provide a full detailed assignment of each peak.

The NOESY spectra should be well phased. I am surprised that the author did not correct.

For this reason, I cannot recommend the publication of the manuscript

Reviewer #4 (Remarks to the Author):

Comments to the author:

The authors clearly improved the data and therefore the manuscript. While there is no doubt, that the respective compounds have a clear effect on the tumor development, the specific effect on macrophages is still not convincingly presented.

Comment 1: everything is fine.

Comment 2: The authors performed further analyses regarding viability and stability of the cells or the phenotype, respectively.

I expressed my concern, that the compound might not repolarize the macrophages from pro- to anti-inflammatory but might just damage the cells or even kills them, respectively. Reason for my concern was the fact that every single marker expression that is shown in Fig 3 actually goes down. In the response to my concerns, the authors demonstrate a very strong pro-apoptotic effect of the compound on peritoneal macrophages. After 12 h treatment with 2 μ M, half of the cells are dead. So it is not surprising that in Fig s32 the cytokine production after LPS treatment is down. It is also not surprising that expression of all the tested mRNA in all the tested condition goes down if one does just intoxicate the cells. Additionally, 12 h is not really long term considering a treatment of mice or even patients. Therefore, what I can see in these data is that the anti-inflammatory effect as well as the anti-tumor-promoting effect on macrophages is actually an anti-macrophage effect. Therefore I have to ask these questions:

- Is there any marker that does NOT go down after treatment with the compound?
- How toxic is the compound regarding other immune cells or cancer cells?
- Is there a dose, for which the compound exerts an immunomodulatory effect without killing half of the cells after 12 hours?
- On a minor note: how do the authors explain the increase of VEGF, PDGF and MMP9 under il4 conditions after 12 h in Fig s32, while the mRNA expression of these markers goes down in Fig. 3?

Comment 3: everything is fine. Yet I have to comment on the colitis-associated colon cancer: while AOM/DSS is in some way a model for human CAC development, what the mouse gets at the end is not cancer. What one gets are tumors, more specifically adenomas. This is shown nicely in Fig. R1, where we see six adenoma and not one example of cancer. The cited publication of Song et al. is naming their model correctly as tumorigenesis (and not carcinogenesis). For the sake of

accuracy, I would like to ask the authors to check the text for this and consider changing the respective expressions.

Comment 4: The authors clearly improved the FACS plots as well as the histological analysis. The authors state that Fig. 5f is gated on F4/80 and CD11b double positives. Yet there are still CD11b negatives in the dot plots. So I assume the pre-gating was on F4/80 alone. But what are then the shown percentages? CD206+CD11b+ out of F4/80+? Or CD80+CD11b+ out of F4/80+, respectively? With the given CD80 staining, this strategy makes not much sense, as there are no CD80 negative or positive populations. Here, only the MFI analysis as seen in Fig5h is useful. But also for CD206 I would recommend to define macrophages as F4/80+CD11b+ and then check how many of these are CD206 positive. On a minor note, isotype controls to define the positive fractions are outdated for many years and when analyzing single cells out of a tissue (mechanical manipulation, enzymatic digestion) a live/dead staining is highly recommended. Regarding the fluorescence tissue staining, this looks nice, but of course, the authors need to count the cells and deliver a proper statistic. One needs to see how many macrophages are there and how many of them are CD206 positive. For example in the RG@M- γ -CD group in Fig.s37 there are clearly less CD206, but there also way less macrophages in general. For the CD80 staining in the respective group this is not the case. The authors could also consider moving this figure then out of the supplementals to the main text as these are really important data to accept the claim of the effect on macrophages.

Comment 5: The authors clearly improved the immunohistochemistry. The staining is better, the chosen regions are consistent now, the technical conditions appear equal.

Comment 6: As above, the immunohistochemistry is fine and the immunofluorescence needs counting.

What is this huge shadow in the DAPI staining of M- γ -CD?

For Fig. s39: what I see here is again not a reduction of CD206 expression among the macrophages but a reduction of macrophages as such. What might be a good thing as they are almost all CD206 positive and therefore most probably tumor promoting, yet mechanistically it is something different.

New comment: As it has me confused while reading the paper again, the authors have to make very clear in the manuscript what their compound actually should do. What is the hypothesis on the mode of action? It should decrease the number of M1 macrophages in the colitis, where there are only M1 macrophages and it should decrease the number of M2 macrophages in the TME, where there are only M2 macrophages. Therefore, it should actually just kill all macrophages. If the authors think otherwise, this has to be properly stated and discussed in the manuscript. And in regard to the mode of action: how or when should the compound be given to a patient and with what purpose? In case of colitis to fight inflammation and prevent tumorigenesis or in case of cancer to help the immune system to fight the tumors, but then the anti-inflammatory effect might be a problem. I would like to ask the authors to discuss the potential use case, as this did not become clear to me.

Reviewer #2 (Remarks to the Author):

The authors very slightly improved the manuscript in the section of the chemical characterization of the new mannose derivative. Characterization is still very lacking. The assignment of the spectra added is very superficial.

Response: This is an important comment for our manuscript. According to your comment, we have carefully re-analyzed the NMR data of M- γ -CD and other compounds.

In this revision, we have performed detailed assignment for each compound, all chemical shifts in ^1H and ^{13}C NMR spectra have been assigned through in-depth analysis of 1D and 2D NMR spectra.

Correspondingly, we have labeled the proton numbers and carbon numbers in the spectra based on the illustration of structural formula. For consistency, the protons and carbons in M- γ -CD moiety were labeled with H_i and C_i (i , numbers), respectively. The protons and carbons from Mannose moiety were labeled with H_i' and C_i' , respectively. As for the guest molecules (RG and Rho), their protons and carbons were labeled with H_i'' and C_i'' , respectively.

For better understanding, the assignment tables have been reproduced to present the correspondences of chemical shifts to hydrogen and carbon atoms. With the full assignments, the section of chemical characterization has been updated in **Supporting Information**.

Authors report the characterization of a new compound with a new synthetic route. They should show the identity first. Especially since in the biological data the differences between RG@M-CD and RG@CD are often minimal, smaller than the experimental error. In Table s3: -CH- and OH.... Which -CH- and OH? A deep assignment should be provided to support the synthetic route.

Response: We are sorry that the proton and carbon numbers were not properly listed in the assignment tables during last revision, which led to the unclear assignment. According to your comment, a deep assignment of NMR spectra of M- γ -CD has been performed using 2D HSQC spectrum.

In the ^1H spectrum, M- γ -CD showed overlapped signals of mannose and CD, owing to the structural similarity between mannose and the glucose units of CD. In comparison, the ^{13}C NMR spectrum of M- γ -CD showed distinguishable signals for mannose and CD. Through comprehensive analysis of 1D NMR spectra and 2D

HSQC spectrum, we have made the detailed assignment, and the coupling constants J (Hz) of C-H protons have also been provided. Besides, a newly-generated signal at 154.22 ppm was observed in ^{13}C NMR spectrum, which was assigned to the newly-formed carbonyl (C10) introduced by CDI. In HMBC spectrum, the C10 signal has showed correlations with H6 of CD and H1' of mannose, implying the conjugation between C6 of γ -CD and C1' of mannose.

Figure s7. ^1H NMR spectrum of M- γ -CD (500 MHz, DMSO- D_6 , room temperature)

Figure s8. ^{13}C NMR spectrum of M- γ -CD (500 MHz, DMSO- D_6 , room temperature)

Figure s9. HSQC spectrum of M- γ -CD (500 MHz, DMSO- D_6 , room temperature)

Figure s10. HMBC spectrum of M- γ -CD (500 MHz, DMSO-D₆, room temperature)

¹H NMR (500 MHz, DMSO) δ 5.781 (d, J = 1.7 Hz, 1H), 5.758 (d, J = 6.4 Hz, 1H), 4.887 (d, J = 3.5 Hz, 1H), 4.884 (s, 1H), 4.531 (t, J = 5.6 Hz, 1H), 3.625 (d, J = 4.3 Hz, 2H), 3.620 (s, 2H), 3.588 (d, J = 9.6 Hz, 1H), 3.578 (s, 1H), 3.543 (s, 1H), 3.533 (d, J = 9.8 Hz, 1H), 3.370-3.333 (m, 1H), 3.370-3.359 (m, 1H), 3.333 (s, 1H), 3.325 (d, J = 8.1 Hz, 1H). ¹³C NMR (126 MHz, DMSO) δ 154.22, 102.19, 94.50, 81.43, 73.70, 73.42, 73.10, 72.68, 71.93, 71.10, 67.89, 62.05, 60.48.

Table s3. ¹³C NMR and ¹H NMR chemical shifts of M- γ -CD in DMSO-D₆

Carbon No.	¹³ C δ (ppm)	HSQC ¹ H δ (ppm)	J value
C1	102.19	4.887 (H1)	d, J = 3.5 Hz
C2	73.10	3.325 (H2)	d, J = 8.1 Hz
C3	73.42	3.588 (H3)	d, J = 9.6 Hz
C4	81.43	3.370-3.333 (H4)	m
C5	72.68	3.533 (H5)	d, J = 9.8 Hz
C6	60.48	3.625 (H6)	d, J = 4.3 Hz
C10	154.22	/	/
C1'	94.50	4.884 (H1')	s
C2'	71.93	3.333 (H2')	s
C3'	71.10	3.578 (H3')	s
C4'	67.89	3.370-3.359 (H4')	m
C5'	73.70	3.543 (H5')	s
C6'	62.05	3.620 (H6')	s
Oxygen No.		¹ H δ (ppm)	J value

O8	/	5.758 (H8)	d, $J = 6.4$ Hz
O9	/	5.781 (H9)	d, $J = 1.7$ Hz
O11	/	4.531 (H11)	t, $J = 5.6$ Hz

It is strange that, in the ^1H spectra, the signals of mannose and CD are always overlapped, and that in Fig R2 CD and M-CD show the same. Of course, the shift from 3.802 to 3.795 is not evidence of functionalization (Fig R2C). 2D NMR spectra can quickly provide a full detailed assignment of each peak.

Response: We agree with the reviewer that the signals of mannose and CD exhibited overlaps in the ^1H NMR spectrum of M- γ -CD.

This phenomenon is attributed to the structural property of M- γ -CD. Mannose and CD's glucose units had same structures. After conjugation, the protons of glucose structures from mannose and CD were restrained in a similar chemical circumstance. This made the signals of mannose moiety and CD overlapped in ^1H NMR spectrum.

According to your suggestion, we have performed the detailed assignment using the 2D NMR spectrum (**Fig. R1**). The heteronuclear single quantum coherence (HSQC) spectrum was carefully phased, and the chemical shifts in the ^1H NMR spectrum was assigned based on the correlations in the HSQC spectrum. It can be indicated that the chemical shifts of mannose moiety and CD were not completely overlapping, and the signals of corresponding protons from mannose and CD's glucose units were separated to a small extent.

Figure R1. HSQC spectrum of M- γ -CD (500 MHz, DMSO-D₆, room temperature)

In HMBC spectrum, a newly-generated signal at 154.22 ppm (C10) showed correlation signals with H6 of CD and H1' of mannose, which indicated that a

CDI-derived carbonyl was formed during the substitution reaction and the conjugation was taken place between C6 of γ -CD and C1' of mannose (**Fig. R2**).

Figure R2. HMBC spectrum of M- γ -CD (500 MHz, DMSO-D₆, room temperature)

Furthermore, we have investigated the mixture of mannose and CD at molar ratio for reaction using NMR spectroscopy. Interestingly, without conjugation, their signals in the ¹H NMR spectrum were quite distinguishable, reflecting the independent chemical circumstances of mannose and CD. When mannose was conjugated to γ -CD, the signals of glucose units of M- γ -CD in the ¹H NMR spectrum were prone to convergence (**Fig. R3**).

Figure R3. Comparison of ^1H NMR spectra between the mixture of mannose and γ -CD at molar ratio for reaction (below) and M- γ -CD (above) The red frames indicated the differences in the spectra (500 MHz, DMSO-D6, room temperature)

In comparison with γ -CD, M- γ -CD exhibited markedly different physical property. As shown in **Fig. R4**, γ -CD was in powdered form, while M- γ -CD displayed a fleecy feature. Besides, the water-solubility of M- γ -CD at 25 °C was significantly enhanced compared with γ -CD. The equilibrium solubility of M- γ -CD was 669.3 ± 16.0 mg/mL, while the equilibrium solubility of γ -CD was 257.4 ± 12.4 mg/mL.

Figure R4. Images of M- γ -CD and γ -CD in solid state. Equilibrium solubility of M- γ -CD and γ -CD at 25 °C

The NOESY spectra should be well phased. I am surprised that the author did not correct.

Response: We are sorry for our carelessness that the NOESY spectra were not well phased during last revision. This is quite important suggestion and only the phased NOESY spectra could provide the accurate information on correlation.

According to your comment, the NOESY spectra in **Fig. s16** and **Fig. 1d** have been carefully phased using MestReNova software. Then the correlation between RG and M- γ -CD has been investigated based on detailed assignment and signals in the NOESY spectrum. In the partial magnification of 2D NOESY spectrum, signals of NOE correlation were detected between (1) amide protons of RG (H17'' and H19'') and glycogen of M- γ -CD (H1 (H1')-H4 (H4')); (2) benzene protons of RG (H2'', H24'' and H25'') and glycogen of M- γ -CD (H3 (H3')), reflecting the complexation between host and guest molecules. For better understanding, we have supplemented the illustration of RG@M- γ -CD with proton and carbon numbers in the partial magnification of 2D NOESY spectrum. The illustration was consistent with the structural formula in the NMR characterization section of **Supporting Information**.

Figure s16. 2D-NOESY spectrum (500 MHz, DMSO-D₆, room temperature) of RG@M- γ -CD

Figure s1d. Partial magnification of 2D NOESY spectrum of RG@M- γ -CD. The red lines indicated the correlations between RG and M- γ -CD.

Reviewer #4 (Remarks to the Author):

Comments to the author

The authors clearly improved the data and therefore the manuscript. While there is no doubt, that the respective compounds have a clear effect on the tumor development, the specific effect on macrophages is still not convincingly presented.

Comment 1: everything is fine.

Response: We appreciate the favorable reply you have shown to us. Thank you very much for your positive comment.

Comment 2: The authors performed further analyses regarding viability and stability of the cells or the phenotype, respectively.

I expressed my concern, that the compound might not repolarize the macrophages from pro- to anti-inflammatory but might just damage the cells or even kills them, respectively. Reason for my concern was the fact that every single marker expression that is shown in Fig 3 actually goes down. In the response to my concerns, the authors demonstrate a very strong pro-apoptotic effect of the compound on peritoneal macrophages. After 12 h treatment with 2 μ M, half of the cells are dead. So it is not surprising that in Fig s32 the cytokine production after LPS treatment is down. It is also not surprising that expression of all the tested mRNA in all the tested condition goes down if one does just intoxicate the cells. Additionally, 12 h is not really long term considering a treatment of mice or even patients. Therefore, what I can see in these data is that the anti-inflammatory effect as well as the anti-tumor-promoting effect on macrophages is actually an anti-macrophage effect.

Therefore I have to ask these questions:

- Is there any marker that does NOT go down after treatment with the compound?
- How toxic is the compound regarding other immune cells or cancer cells?
- Is there a dose, for which the compound exerts an immunomodulatory effect without killing half of the cells after 12 hours?
- On a minor note: how do the authors explain the increase of VEGF, PDGF and MMP9 under il4 conditions after 12 h in Fig s32, while the mRNA expression of these markers goes down in Fig. 3?

Response: Thank you for raising these important questions. According to your comments and questions, we have amended the experimental design and supplemented *in vitro* experiments regarding macrophages to investigate the regulatory effect of RG@M- γ -CD.

First, the treatment upon macrophages has been prolonged to 24 h to explore the long-term cytotoxic effect of RG@M- γ -CD. Besides, we have tested the cytotoxicity of RG@M- γ -CD regarding other immune cells and colon cancer cells including primary dendritic cells, splenocytes, SW480 and RKO cells.

Second, the dosage of RG@M- γ -CD has been carefully screened according to the cytotoxicity result regarding macrophages. q-PCR and ELISA assays have been re-performed using the appropriate dosage to investigate the regulatory effect of RG@M- γ -CD.

Third, the expression levels of other macrophage makers have also been examined to demonstrate that RG@M- γ -CD can regulate macrophages without killing them. For the sake of consistency and accuracy, the *in vitro* experiments have been re-performed using the same batch of peritoneal macrophages. Corresponding results have been updated in **Figure 3** and **Supporting Information**.

C57BL/6 mice were intraperitoneally injected with thioglycollate broth. Peritoneal macrophages (PMs) were surgically isolated from the peritoneal cavity of treated mice. The isolated cells were incubated in complete DMEM medium. After 2 h incubation, non-adherent cells were removed to obtain PMs. Bone marrow-derived cells were extracted from marrow cavities of femurs. Then the cells were incubated with 10 ng mL⁻¹ GM-CSF and 5 ng mL⁻¹ IL-4 for 7 days for the differentiation of dendritic cells (DCs). The spleens were dispersed with 70- μ m cell strainer. Then the splenocytes were isolated using mouse lymphocyte separation medium.

These primary cells were seeded into 96-well plates, and treated with RG, RG@ γ -CD and RG@M- γ -CD for 4 h, 12 h, and 24 h. The cytotoxicity regarding PMs was evaluated by MTT assay. For DCs and splenocytes, the cytotoxicity of RG@M- γ -CD was assessed by CCK-8 assay. As shown in **Fig. s33** and **Table s11**, within 24 h treatment, RG@M- γ -CD at low concentrations (< 0.5 μ M) induced negligible cytotoxic effects on these immune cells. At high concentrations, RG@M- γ -CD brought dose-dependent cell death. Compared with free RG, M- γ -CD nano-formulation magnified the inhibitory effects of RG owing to efficient cell internalization and sustained drug release. Similar phenomenon has also been reported in previous papers regarding drug delivery of RG,^[1, 2] that RG nanomedicine with high drug contents can inhibit the viability of macrophages, meanwhile it influences the polarization of M2 TAMs when at appropriate dosage.^[2] **Fig. s32** indicated the cytotoxicity of RG@M- γ -CD regarding other colon cancer cells. Within 12 h treatment, RG@M- γ -CD induced significantly higher lethality upon colon cancer cells as compared to free RG. The IC50 values for SW480 and RKO cells were supplemented in **Table s10**.

[1] Zhao PF, et al. Dual-targeting biomimetic delivery for anti-glioma activity via remodeling the tumor microenvironment and directing macrophagemediated immunotherapy. *Chem Sci* **9**, 2674-2689 (2018).

[2] Zhao PF, et al. Dual-Targeting to Cancer Cells and M2 Macrophages via Biomimetic Delivery of Mannosylated Albumin Nanoparticles for Drug-Resistant Cancer Therapy.

Figure s33. Cytotoxicity test in HUVECs, PMs, DCs and splenocytes treated with RG, RG@M- γ -CD and RG@M- γ -CD for 4 h, 12 h and 24 h. Concentrations of equivalent RG ranged from 0.5 μ M to 10 μ M. Data were expressed as means \pm s.e.m (n=5).

Table s11. IC₅₀ values for RG, RG@ γ -CD and RG@M- γ -CD in target cells

IC ₅₀ , μ M	HUVECs		PMs		DCs		splenocytes	
	12 h	24 h	12 h	24 h	12 h	24 h	12 h	24 h
RG	12 h	24 h	12 h	24 h	12 h	24 h	12 h	24 h
	4.66	2.54	9.22	4.68	/	9.24	/	/
RG@γ-CD	12 h	24 h	12 h	24 h	12 h	24 h	12 h	24 h
	3.75	2.62	5.08	4.20	/	8.14	/	9.65
RG@M-γ-CD	12 h	24 h	12 h	24 h	12 h	24 h	12 h	24 h
	3.48	1.76	4.34	4.06	/	7.60	9.34	8.50

Figure s32. Cytotoxicity test in CT26, HT29, SW480 and RKO cells treated with RG, RG@ γ -CD and RG@M- γ -CD for 4 h, 8 h and 12 h. Concentrations of equivalent RG ranged from 0.5 μ M to 10 μ M. Data were expressed as means \pm s.e.m (n=5).

Table s10. IC₅₀ values for RG, RG@ γ -CD and RG@M- γ -CD in CRC cells

IC ₅₀ , μ M	CT26		HT29		SW480		RKO	
	8 h	12 h	8 h	12 h	8 h	12 h	8 h	12 h
RG	9.52	8.20	7.65	6.84	8.44	8.08	8.42	7.28
RG@γ-CD	6.65	5.76	5.66	4.58	7.20	5.74	7.35	5.05
RG@M-γ-CD	5.86	3.04	4.43	2.63	5.15	4.20	5.82	4.45

During the initial *in vitro* experiments and last revision, we have noticed the cytotoxicity of RG@M- γ -CD regarding PMs. Since 12 h treatment of RG@M- γ -CD at 2 μ M could induce obvious cell death, we have controlled the drug content at 2 μ M and the treatment was only lasted for 4 h. Afterwards, the drug-containing media were removed and the cells were washed to terminate the treatment. Then the cells were stimulated with LPS- or IL-4-containing media to allow polarization. After 2 h or 12 h stimulation with LPS or IL-4, the expression of markers was measured, which could

reflect the regulatory effect of RG@M- γ -CD and the phenotype stability.

We are sorry that we omitted the "treatment termination and cell wash" procedure in the description of experiment method, which led to the misunderstanding that the treatment were lasted for 12 h.

After carefully reading your comment, we have decided to amend the experimental design regarding PMs. PMs were incubated with various formulations for 4 h (short term), 12 h and 24 h (long term). Based on the MTT result, the RG content was controlled below 0.5 μ M. After treatment, the cells were washed and were given with LPS (or IL-4) stimulation.

The effective dose of RG@M- γ -CD was determined using q-PCR assay. Based on MTT result, we used 0.2 μ M and 0.5 μ M of RG@M- γ -CD to test the regulatory effects. With 24 h treatment followed by 2 h stimulation, macrophage polarization was found to be regulated in 0.5 μ M RG@M- γ -CD group (**Fig. R1**). The PMs treated with RG@M- γ -CD kept high viability, meanwhile their phenotype-related factors were significantly down-regulated. Treatment with 0.2 μ M RG@M- γ -CD only showed negligible deactivation effect on macrophages, and the regulation was not significant. The mRNA level of CD11b and caspase-3 was examined as reference markers. We found that there was no significant difference for CD11b and caspase-3 levels among the treatment groups, providing supporting information that RG@M- γ -CD could regulate the polarization of PMs without killing them.

Figure R1. Peritoneal macrophages were treated with different formulations for 24 h. Then the cells were washed, followed by incubation with LPS or IL-4 for 2 h. Afterwards, the total mRNA of the treated cells was extracted for q-PCR. The expression levels of IL-1 β , IL-6, TNF- α , Arg-1, IL-10 and CD206 were quantified for phenotype evaluation. The expression levels of CD11b and caspase-3 were quantified as reference markers. (A) mRNA expression levels of pro-inflammatory cytokines by

peritoneal macrophages treated with different formulations (equivalent RG, 0.5 μ M, n=3). (B) mRNA expression levels of Arg-1, IL-10 and CD206 by macrophages under different treatments (equivalent RG, 2.0 μ M, n=3). Data were expressed as means \pm s.e.m, *P < 0.05, **P < 0.01, ***p < 0.001.

In order to confirm the q-PCR result and test the phenotype stability, 4 h, 12 h and 24 h treatments at 0.5 μ M were conducted on PMs, followed by 6 h stimulation with LPS or IL-4. Then ELISA assay was performed to quantify the protein levels of phenotype-related factors including IL-6, TNF- α , VEGF-B, and MMP-9. ELISA results demonstrated that the inflammatory cytokines and tumor-promoting factors were significantly down-regulated with RG@M- γ -CD treatment for 12 h and 24 h, indicating the deactivation effects on macrophages (**Fig. s36**). Moreover, such effect showed a time-dependent manner, the decline tendency of phenotype-related markers became more remarkable along with the treatment prolonging, suggesting the stability of the regulation effect of RG@M- γ -CD.

Figure s36. The regulatory effect of RG@M- γ -CD on the polarization of peritoneal macrophages. PMs were subjected to the treatments for 4 h, 12 h and 24 h (equivalent RG, 0.5 μ M), then were incubated with LPS or IL-4 for 6 h to induce polarization. The protein expression levels of phenotype-related cytokines were determined by

ELISA, including pro-inflammation cytokines (IL-6 and TNF- α) and M2-related factors (VEGF-B and MMP-9). n=3, *p < 0.05, **p < 0.01.

Comment 3: everything is fine. Yet I have to comment on the colitis-associated colon cancer: while AOM/DSS is in some way a model for human CAC development, what the mouse gets at the end is not cancer. What one gets are tumors, more specifically adenomas. This is shown nicely in Fig. R1, where we see six adenoma and not one example of cancer. The cited publication of Song et al. is naming their model correctly as tumorigenesis (and not carcinogenesis). For the sake of accuracy, I would like to ask the authors to check the text for this and consider changing the respective expressions.

Response: We appreciate this important suggestion you have given to us. We have carefully read the published paper by Song et al., and made the correction throughout the manuscript. According to your comments and the description of AOM-DSS model in the published paper, we have revised the model name into "CAC mouse model". Besides, we have corrected the "cancer" or "carcinogenesis" into "tumor" or "tumorigenesis". For example, "CAC progression" has been corrected into "tumor development".

Comment 4: The authors clearly improved the FACS plots as well as the histological analysis.

The authors state that Fig. 5f is gated on F4/80 and CD11b double positives. Yet there are still CD11b negatives in the dot plots. So I assume the pre-gating was on F4/80 alone. But what are then the shown percentages? CD206+CD11b+ out of F4/80+? Or CD80+CD11b+ out of F4/80+, respectively? With the given CD80 staining, this strategy makes not much sense, as there are no CD80 negative or positive populations. Here, only the MFI analysis as seen in Fig5h is useful. But also for CD206 I would recommend to define macrophages as F4/80+CD11b+ and then check how many of these are CD206 positive. On a minor note, isotype controls to define the positive fractions are outdated for many years and when analyzing single cells out of a tissue (mechanical manipulation, enzymatic digestion) a live/dead staining is highly recommended.

Regarding the fluorescence tissue staining, this looks nice, but of course, the authors need to count the cells and deliver a proper statistic. One needs to see how many macrophages are there and how many of them are CD206 positive. For example in the RG@M-y-CD group in Fig.s37 there are clearly less CD206, but there also way less macrophages in general. For the CD80 staining in the respective group this is not the case. The authors could also consider moving this figure then out of the supplementals to the main text as these are really important data to accept the claim of the effect on macrophages.

Response: We thank for the reviewer's important comment. Fig. 5f was gated on F4/80 alone, and the shown percentages should be CD206⁺ CD11b⁺ and CD80⁺ CD11b⁺ out of F4/80⁺, respectively. We are sorry for this mistake. As resident macrophages of the colon lamina propria constitutively express CD206,^[1] there are also no obvious CD206 negative or positive populations. So according to your advice, we have firstly define macrophages as F4/80⁺ CD11b⁺, and then showed the MFI analysis for both CD80 and CD206 (**Fig. s42**). As you mentioned, for flow cytometry, a live/dead staining is required when analyzing single cells out of a tissue. We are sorry for not doing this staining, though we only used FSC and SSC to exclude debris.

[1] Rivollier A, He J, Kole A, Valatas V, Kelsall BL. Inflammation switches the differentiation program of Ly6C(hi) monocytes from antiinflammatory macrophages to inflammatory dendritic cells in the colon. *Journal of Experimental Medicine* **209**, 139-155 (2012).

Figure s42. FACS analysis using colon lamina propria cells of tumor regions for CAC mouse model. (a) Statistical analysis of F4/80⁺ CD11b⁺ population to indicate the macrophage subset for different treatment groups. (b) Representative fluorescence intensities and (c) median values of CD206⁺ and CD80⁺ subsets in macrophage subset for different treatment groups. Data were expressed as points with means \pm s.e.m, *P < 0.05, **P < 0.01 (n=3).

We appreciate your constructive suggestion regarding the immunofluorescence result. In order to make the immunofluorescence more comprehensible, we have carefully counted the macrophages and the CD206 (CD80) positive cells in the images, then performed the statistic analysis (**Fig. s41**). The result indicated that RG@M- γ -CD nano-medicine exerted deactivation mechanisms on macrophages, by which the nanomedicine suppressed the polarization of protumorigenic TAMs via signaling blockade, and attenuated the production of pro-inflammatory cytokines via M- γ -CD functioning.

Counting analysis of M2 TAMs

Counting analysis of M1 TAMs

Figure s41. Counting analysis of intratumoral TAMs based on immunofluorescence data. Above: Cell count of F4/80⁺ cells in 150 $\mu\text{m} \times 150 \mu\text{m}$ regions of immunofluorescence images from the treatment groups, and the percentage of CD206⁺ cells in F4/80⁺ cells (M2). Below: Cell count of F4/80⁺ cells in 150 $\mu\text{m} \times 150 \mu\text{m}$ regions of immunofluorescence images from the treatment groups, and the

percentage of CD80⁺ cells in F4/80⁺ cells (M1). Data were expressed as points with means \pm s.e.m, *P < 0.05, **P < 0.01 (n=3).

According to your suggestion, we have moved the immunofluorescence to the main text to illustrate the regulation of RG@M- γ -CD on macrophages, and have updated the **Fig. 5**. Related statistic analysis has been supplemented in **Supporting Information (Fig. s41)**. As your comment, in comparison with control group, the amount of macrophages in RG@ γ -CD and groups showed a tendency of reduction without statistic significance, probably owing to the broad-spectrum of kinase inhibition of RG-derived nano-medicine which might influence monocyte differentiation and macrophage amount. With the supplement of immunofluorescence, we have moved the result of FACS to **Supporting Information**.

Figure 5. *In vivo* anti-tumor mechanism of RG@M- γ -CD CNPs in CAC mouse model. (a) Heat map of differential expression (DE) genes involved in TME mediation and tumor inhibition (RG@M- γ -CD VS. control, n=3). (b) GOEA of differential expression DE genes (fold change >1.5, P<0.05) visualized as GO network, where nodes with different colors indicated the clusters based on connected GO terms. The clusters were annotated manually by a shared general term. (c) Histograms of DE genes associated with tumor inhibition (left part) and TME (right part) mediation based on GO annotation (-log₁₀ (p value), 27-15). (d) Representative images of Tumor, Ki67 and CD34

CD34 staining sections of CAC tumor tissues from different treatment groups. Scale bar, 50 μ m. (e) mRNA expression levels of inflammatory cytokines (IL-1 β , IL-6 and TNF- α) and M2-related tumor-supportive cytokines (VEGF-B, PDGF- α , and MMP-9) in CAC tumor tissues from different treatment groups (n=3). Data were expressed as means \pm s.e.m, *P < 0.05, **P < 0.01, ***P < 0.001. (f) Immunofluorescence analysis on colon tumor sections from treatment groups by CD11b, CD80 and CD206 staining. Representative images of tumor areas from each group were imaged using inverted fluorescence microscopy. Scale bar, 50 μ m. (g) Representative images of VEGF staining sections of CAC tumor tissues from different groups. Scale bar, 50 μ m.

Comment 5: The authors clearly improved the immunohistochemistry. The staining is better, the chosen regions are consistent now, the technical conditions appear equal.

Response: We appreciate your favorable reply, and thank you very much for your positive comment on our revision of immunohistochemistry.

Comment 6: As above, immunohistochemistry is fine and the immunofluorescence needs counting.

What is this huge shadow in the DAPI staining of M- γ -CD?

For Fig. s39: what I see here is again not a reduction of CD206 expression among the macrophages but a reduction of macrophages as such. What might be a good thing as they are almost all CD206 positive and therefore most probably tumor promoting, yet mechanistically it is something different.

Response: Thank you very much for your valuable comment. We have supplemented the counting result along with the immunofluorescence images (**Fig. s46**). The shadow in M- γ -CD group and the reduction of F4/80⁺ was caused by the autofocus of fluorescence microscopy which led to regional darkening in the images. With the technician's assistance, we have reproduced the immunofluorescence images using the microscopy imaging system, and the updated immunofluorescence result has been shown in **Fig. s45**.

According to your suggestion, we have counted the positive cells in 150 μm \times 150 μm regions and performed the statistic analysis (**Fig. s46**). The result indicated that treatment with RG@ γ -CD or RG@M- γ -CD brought a slight reduction of intratumoral infiltration of macrophages. This phenomenon was probably caused by the kinase inhibition of RG, by which RG@M- γ -CD mediated the differentiation of monocytes and affected the amount of macrophages in tumor region.

Figure s45. Immunofluorescence analysis on CT26 tumor sections by F4/80 and CD206 staining. Representative images of tumor areas from treatment groups were captured using inverted fluorescence microscopy. Scale bar, 50 μm .

Figure s46. Counting analysis of intratumoral TAMs based on immunofluorescence data. Cell count of F4/80⁺ cells in 150 μm × 150 μm regions of immunofluorescence images from the treatment groups, and the percentage of CD206⁺ cells in F4/80⁺ cells. Data were expressed as points with means ± s.e.m, *P < 0.05, **P < 0.01, ***P < 0.001 (n=3).

New comment: As it has me confused while reading the paper again, the authors have to make very clear in the manuscript what their compound actually should do. What is the hypothesis on the mode of action? It should decrease the number of M1 macrophages in the colitis, where there are only M1 macrophages and it should decrease the number of M2 macrophages in the TME, where there are only M2 macrophages. Therefore, it should actually just kill all macrophages. If the authors think otherwise, this has to be properly stated and discussed in the manuscript. And in regard to the mode of action: how or when should the compound be given to a patient and with what purpose? In case of colitis to fight inflammation and prevent tumorigenesis or in case of cancer to help the immune system to fight the tumors, but then the anti-inflammatory effect might be a problem. I would like to ask the authors to discuss the potential use case, as this did not become clear to me.

Response: This is an important comment for us to improve the scientificity of the manuscript. According to your comment, we have supplemented comprehensive discussion on the mode of action in the **Discussion** section. The supplement contained following aspects: the link between inflammation and CRC, the role of macrophages in inflammation and CRC, the mechanism of RG, the therapeutic effect of RG@M- γ -CD nano-medicine, and the mode of action of RG@M- γ -CD.

It has been well-established that the most fraction of CRC has been linked to intestinal physiological factors including environmental or food-borne mutagens, intestinal commensals or pathogens, as well as chronic intestinal inflammation. Especially CAC, a subtype of CRC, is directly associated with chronic inflammation which initiates genomic instability and mutations in oncogenes and tumor suppressor genes, leading to tumorigenesis. Although the pathogenesis of non-inflammatory CRC differs from that of CAC, clinical studies have indicated that non-inflammatory CRC tumors also display obvious inflammatory infiltration and increased pro-inflammatory cytokines within the localized microenvironment, which can promote accumulation of additional mutations and epigenetic changes during CRC development.^[1-4] Therefore, intervention strategies for targeting the CRC-related inflammation have been being evaluated as adjuvant therapy for CRC treatment.^[5,6]

Colonic macrophages are one of the most abundant inflammatory cells in the colon, whose polarization contributes to homeostasis, inflammation and CRC.^[7,8] During CRC progression, macrophages within tumor microenvironment can be activated into M1 and M2 TAMs. The activated TAMs represent the extremes of a

continuum spectrum of different phenotypic features, which are determined by the variation of the physiological/pathological conditions. For non-inflammatory CRC progression, M1 TAMs begin to accumulate within the adenoma. Following the malignant transformation that results in cancer, M2 cells become the predominant TAMs. For CAC, M1 macrophages are highly expressed in the lamina propria adjacent to neoplasm, contributing to pro-inflammatory microenvironment. Within adenomas or tumors, M2 phenotype predominates to exert the protumorigenic role. Despite that the M1 cells can antagonize the growth of established tumors, they contribute to CRC-related inflammation by producing cytokines including TNF, IL-6, and IL-1. These cytokines can mediate the activity of transcription factors NF- κ B and STAT3 which are particularly important in the development of CAC and CRC.

RG simultaneously targets several hallmarks of CRC development through broad kinase inhibition with anti-angiogenesis, anti-proliferation and anti-metastasis. Increasing data indicate that RG has an immune-modulating effect to reduce M2 infiltration in CRC by blocking TIE2 signaling.^[9,10] Despite these, RG shows poor drug properties (i.e. solubility, dissolution, permeability, circulation and distribution) and some side effects, which compromises its therapeutic outcome.

Compared with free RG, our designed RG@M- γ -CD nanomedicine exhibits superiorities combating CRC. First, owing to the tailored nano-morphology and the CRC targeting capacity, RG@M- γ -CD CNPs improved drug delivery efficiency to targeted cells and therefore elevated the bioavailability of TKI against malignancy. In comparison with free RG, RG@M- γ -CD treatment achieved enhanced anti-angiogenesis and anti-proliferation effects, and simultaneously reduced the hepatic toxicity of RG. Second, verified in both CAC and CT26 mouse models, RG@M- γ -CD CNPs reversed the tumorous microenvironment by decreasing TAM infiltration and attenuating protumorigenic factors. Such effect was attributed to the kinase inhibition of RG, which was amplified by the M- γ -CD-derived nano-formulation. Third, the non-covalent binding of RG@M- γ -CD would not lesion the bio-function of CD and the CNPs were confirmed to initiate anti-inflammation mechanism through targeting macrophages. Given that inflammation affects multiple facets of CAC and CRC development, RG@M- γ -CD CNPs could be served as an integration system that combined molecular targeted therapy with anti-inflammation adjuvant therapy.

Based on our *in vitro* and *in vivo* studies, we confirm that RG@M- γ -CD

nano-formulation exerts unique deactivation mechanisms on macrophages, by which the nanomedicine attenuates the production of pro-inflammatory cytokines through M- γ -CD functionalization, meanwhile it suppresses the polarization of protumorigenic TAMs through kinase inhibition of RG.

In regard to the mode of action, we believe that prior to treatment, the genotypic subtype and pathogenesis of CRC should be firstly characterized for patients to determine the valid targets.^[11] Since RG provides potent inhibitions on multiple CRC targets and RG@M- γ -CD nanomedicine inherits this broad-spectrum of kinase inhibition, RG@M- γ -CD can be used as monotherapy for the CRC patients with specific molecular and genetic markers. Moreover, considering the aforesaid role of inflammation in CRC, we believe that the anti-inflammation function of RG@M- γ -CD has practical significance for the CRC patients with clinically detectable colorectal inflammation, which is consistent with the opinion that anti-inflammatory reagents can be used as adjuvant strategy for prevention and treatment of CRC.

- [1] Terzic J, Grivennikov S, Karin E, Karin M. Inflammation and Colon Cancer. *Gastroenterology* **138**, 2101-U2119 (2010).
- [2] Ullman TA, Itzkowitz SH. Intestinal Inflammation and Cancer. *Gastroenterology* **140**, 1807-U1148 (2011)
- [3] Lasry A, Zinger A, Ben-Neriah Y. Inflammatory networks underlying colorectal cancer. *Nat Immunol* **17**, 230-240 (2016).
- [4] Mantovani A, Allavena P, Sica A, Balkwill F. Cancer-related inflammation. *Nature* **454**, 436-444 (2008).
- [5] Diakos CI, Charles KA, McMillan DC, Clarke SJ. Cancer-related inflammation and treatment effectiveness. *Lancet Oncol* **15**, E493-E503 (2014).
- [6] Wang D, DuBois RN. The Role of Anti-Inflammatory Drugs in Colorectal Cancer. In: *Annual Review of Medicine*, Vol **64** 131-144 (2013).
- [7] Isidro RA, Appleyard CB. Colonic macrophage polarization in homeostasis, inflammation, and cancer. *Am J Physiol-Gastroint Liver Physiol* **311**, G59-G73 (2016).
- [8] Itzkowitz SH, Yio XY. Inflammation and cancer - IV. Colorectal cancer in inflammatory bowel disease: the role of inflammation. *Am J Physiol-Gastroint Liver Physiol* **287**, G7-G17 (2004).
- [9] Arai H, et al. Molecular insight of regorafenib treatment for colorectal cancer. *Cancer Treat Rev* **81**, 101912 (2019).
- [10] Zhao PF, et al. Dual-Targeting to Cancer Cells and M2 Macrophages via Biomimetic Delivery of Mannosylated Albumin Nanoparticles for Drug-Resistant Cancer Therapy.

Adv Funct Mater **27**, 1700403 (2017).

- [11] Sinicrope FA, Okamoto K, Kasi PM, Kawakami H. Molecular Biomarkers in the Personalized Treatment of Colorectal Cancer. *Clinical Gastroenterology and Hepatology* **14**, 651-658 (2016).

REVIEWERS' COMMENTS

Reviewer #2 (Remarks to the Author):

The authors very slightly improved the part of the chemical characterization of the new mannose derivative. NMR spectra are right now. It is still strange that the functionalized glucose of the cyclodextrin (usually named A) is not different from the other glucose rings in the NMR spectra. Mannose alone is a mixture of beta and alpha anomers. This is the reason for the differences between the spectra of the mixture and the spectra of the conjugate. After the functionalization, only one of this anomer is formed. The manuscript should report, which is the anomer.

I suggest the publication of the manuscript after minor revisions

Reviewer #4 (Remarks to the Author):

The authors performed several experiments and improved their pre-existing data. While I am still concerned about the cytotoxicity of the compound and while I am not 100% convinced that the macrophage re-polarization is the key mechanism of their anti-tumor effect, the anti-tumor effect itself is indisputable and the effect on macrophages has now been laid out properly. Therefore, the authors provide now all the information potential readers will need to evaluate this subject on their own, when the manuscript is published.

Comment 2: The authors performed several experiments to demonstrate an effect on macrophages without killing them. They delivered data, which were needed to understand the mode of action of their compound in a more precise manner.

Comment 3: Everything is fine, but Fig. 4c could be adapted to the nomenclature, too.

Comment 4: The cytometry data have been improved, the immune fluorescence has been counted and demonstrate a small reduction of the overall macrophages but a further reduction of the functional markers within the macrophage populations.

Comment 6: The immunofluorescence has been counted, the microscopic picture improved.

New comment: The authors explained now the potential use case and the desired mode of action of their compound.

Reviewer #2 (Remarks to the Author):

The authors very slightly improved the part of the chemical characterization of the new mannose derivative. NMR spectra are right now. It is still strange that the functionalized glucose of the cyclodextrin (usually named A) is not different from the other glucose rings in the NMR spectra.

Response: We thank Reviewer #2 for the concern on our compounds. In order to excise the interference of H₂O in DMSO-D₆ and characterize the difference between M- γ -CD and γ -CD, we have performed the NMR characterization for γ -CD and M- γ -CD at 60 °C in DMSO-D₆. The assignment has been performed using corresponding HSQC spectra to present the comparison before and after functionalization. It has been found that the M- γ -CD showed different proton signals of H6, H3 and H2 compared with γ -CD, the splitting of H6, H3 and H2 signals varied after conjugation (**Figure R1**). This difference was induced by mannose substitution on the glucose unit of cyclodextrin.

Figure R1. Comparison of ^1H NMR spectra between M- γ -CD and γ -CD (500 MHz, DMSO- D_6 , 60 °C)

Mannose alone is a mixture of beta and alpha anomers. This is the reason for the differences between the spectra of the mixture and the spectra of the conjugate. After the functionalization, only one of this anomer is formed. The manuscript should report, which is the anomer.

Response: This is an important suggestion on our study. We have consulted the published papers on the anomerization of mannose. It has been found that the two anomers showed different chemical shifts for H1 and H4 in D₂O.^[1, 2] We have made the comparison between the published ¹H NMR spectra and our obtained ¹H NMR spectrum of M- γ -CD (in D₂O, 25 °C). As shown in **Figure R2**, the chemical shifts for H1 and H4 in M- γ -CD spectrum were similar with corresponding signals in α -mannose spectrum. Therefore, we believe that in our study, α -anomer is formed after functionalization. According to your suggestion, we have supplemented this information in the main text: "By comparing the chemical shifts of α -/ β -anomers of mannose with the chemical shifts of M- γ -CD in H₂O, it was found that the α -anomer was formed in the M- γ -CD (chemical shifts were listed in Supplementary Table 3)."

Figure R2. Comparison of ¹H NMR spectra between α -/ β -anomers of mannose M- γ -CD in D₂O.

[1] Klaus Bock and HenningThøgersen. **Nuclear Magnetic Resonance Spectroscopy in the Study of Mono- and Oligosaccharides.** Annual Reports on NMR Spectroscopy, 1983 (13): 1-57.

[2] Ami Kosaka, Misako Aida, Yukiteru Katsumoto. **Reconsidering the activation entropy for anomerization of glucose and mannose in water studied by NMR spectroscopy.** Journal of Molecular Structure, 2015 (1093): 195-200.

Reviewer #4 (Remarks to the Author):

The authors performed several experiments and improved their pre-existing data. While I am still concerned about the cytotoxicity of the compound and while I am not 100% convinced that the macrophage re-polarization is the key mechanism of their anti-tumor effect, the anti-tumor effect itself is indisputable and the effect on macrophages has now been laid out properly. Therefore, the authors provide now all the information potential readers will need to evaluate this subject on their own, when the manuscript is published.

Comment 2: The authors performed several experiments to demonstrate an effect on macrophages without killing them. They delivered data, which were needed to understand the mode of action of their compound in a more precise manner.

Response: We thank the Reviewer #4 for the positive comment and we also believe that the supplement during last revision can help to demonstrate the mode of action of RG@M- γ -CD on macrophages.

Comment 3: Everything is fine, but Fig. 4c could be adapted to the nomenclature, too.

Response: This is a very important comment on Figure 4c. The nomenclature in figures should be consistent with the description in the main text. We have corrected the corresponding presentation in Figure 4c and the updated Figure 4c has been shown as below.

Figure 4c. Schematic illustration of CAC mouse model establishment based on AOM-DSS method.

Comment 4: The cytometry data have been improved, the immune fluorescence has been counted and demonstrate a small reduction of the overall macrophages but a further reduction of the functional markers within the macrophage populations.

Response: We thank the Reviewer #4 for the positive comment. With your

professional suggestions, we have improved the scientificity of the in vivo study for publication.

Comment 6: The immunofluorescence has been counted, the microscopic picture improved.

Response: We thank the Reviewer #4 for the positive comment.

New comment: The authors explained now the potential use case and the desired mode of action of their compound.

Response: We thank the Reviewer #4 for the positive comment. According to your suggestion, we have explained the mode of action and the potential application of RG@M- γ -CD, which is important for highlighting the scientific significance of our study. Thank you again for your intensive, professional and helpful suggestions on our study.